# Compositional Risk Minimization

**Divyat Mahajan** [1 2 ★]  **Mohammad Pezeshki** [1]  **Charles Arnal** [1]  **Ioannis Mitliagkas** [2]
**Kartik Ahuja** [1 †]  **Pascal Vincent** [1 2 ★ †]

## Abstract

Compositional generalization is a crucial step towards developing data-efficient intelligent machines that generalize in human-like ways. In this work, we tackle a challenging form of distribution shift, termed *compositional shift*, where some attribute combinations are completely absent at training but present in the test distribution. This shift tests the model's ability to generalize compositionally to novel attribute combinations in discriminative tasks. We model the data with flexible additive energy distributions, where each energy term represents an attribute, and derive a simple alternative to empirical risk minimization termed *compositional risk minimization (CRM)*. We first train an additive energy classifier to predict the multiple attributes and then adjust this classifier to tackle compositional shifts. We provide an extensive theoretical analysis of CRM, where we show that our proposal extrapolates to special affine hulls of seen attribute combinations. Empirical evaluations on benchmark datasets confirms the improved robustness of CRM compared to other methods from the literature designed to tackle various forms of subpopulation shifts.

## 1. Introduction

The ability to make sense of the rich complexity of the sensory world by decomposing it into sets of elementary factors and recomposing these factors in new ways is a hallmark of human intelligence. This capability is typically grouped under the umbrella term compositionality (Fodor & Pylyshyn, 1988; Montague, 1970). Compositionality underlies both semantic understanding and the imaginative prowess of humans, enabling robust generalization and extrapolation. For instance, human language allows us to imagine situations we have never seen before, such as "a blue elephant riding a bicycle on the Moon." While most works on compositionality have focused on its generative aspect, i.e., imagination, as seen in diffusion models (Yang et al., 2023a), compositionality is equally important in discriminative tasks. In these tasks, the goal is to make predictions in novel circumstances that are best described as combinations of circumstances seen before. In this work, we dive into this less-explored realm of compositionality in discriminative tasks.

We work with multi-attribute data, where each input (e.g., an image) is associated with multiple categorical attributes, and the task is to predict an attribute or multiple attributes. During training, we observe inputs from only a subset of all possible combinations of individual attributes, and during test we will see novel combinations of attributes never seen at training. Following Liu et al. (2023), we refer to this distribution shift as *compositional shift*. Towards the goal of tackling these compositional shifts, we develop an adaptation of naive discriminative Empirical Risk Minimization (ERM) tailored for multi-attribute data under compositional shifts. We term our approach Compositional Risk Minimization (CRM). The foundations of CRM are built on additive energy distributions that are studied in generative compositionality (Liu et al., 2022a), where each energy term represents one attribute. In CRM, we first train an additive energy classifier to predict all the attributes jointly, and then we adjust this classifier for compositional shifts.

Our main contributions are as follows:

- *Theory of discriminative compositional shifts:* For the family of additive energy distributions, we prove that additive energy classifiers generalize compositionally to novel combinations of attributes represented by a special mathematical object, which we call *discrete affine hull*. Our characterization of extrapolation is sharp, i.e., we show that it is not possible to generalize beyond *discrete affine hull*. We show that the volume of *discrete affine hull* grows very fast in the number of training attribute combinations thus generalizing to many attribute combinations. The proof techniques developed in this work are very different from existing works on distribution shifts and hence may be of independent interest.

★Work done at Meta †Joint last author [1]Meta FAIR [2]Mila, Université de Montréal. Correspondence to: Divyat Mahajan <divyat.mahajan@mila.quebec>.

*Proceedings of the 42nd International Conference on Machine Learning*, Vancouver, Canada. PMLR 267, 2025. Copyright 2025 by the author(s).

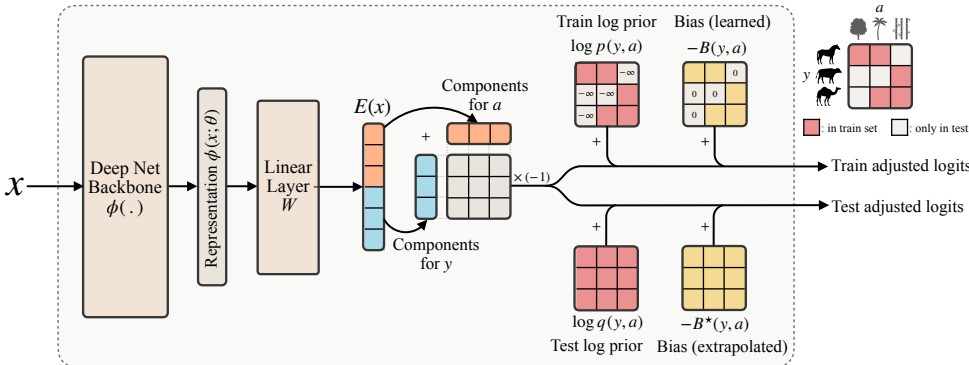

*Figure 1.* The additive energy classifier trained in CRM computes the logits for each group $z = (y, a)$ by adding the energy components of each attribute via boradcasting. For the train logits, we add the log of the prior probabilities and a learned bias $B(y, a)$ for the groups present in train data. At test time, the log prior term is replaced with the log of the test prior (if available, otherwise assumed to be uniform), and the biases for novel test groups, $B^\star(y, a)$, are extrapolated using Eq.11. Finally, we obtain $p(y, a|x)$ by applying softmax function on the adjusted logits. This adaptation from train to test is possible because of the additive energy distribution $p(x|y, a)$, which allows the model to factorize the distribution into distinct components associated with each attribute.

- *A practical method:* CRM is a simple algorithm for training classifiers, which first trains an additive energy classifier and then adjusts the trained classifier for tackling compositional shifts. We empirically validate the superiority of CRM to other methods previously proposed for addressing subpopulation shifts. Our code repository can be accessed via the link in the footnote[1].

## 2. Problem Setting

### 2.1. Generalizing under Compositional Shifts

In compositional generalization, we aim to build a classifier that performs well in new contexts that are best described as a novel combination of seen contexts. Consider an input $x$ (e.g., image), this input belongs to a group that is characterized by an attribute vector $z = (z_1, \ldots, z_m)$ (e.g., class label, background label), where $z_i$ corresponds to the value of $i^{th}$ attribute. There are $m$ attributes and each attribute $z_i$ can take $d$ possible values. So $z \in \mathcal{Z}$ with $\mathcal{Z} = \{1, \ldots, d\}^m$.

We use the Waterbirds dataset as the running example (Sagawa et al., 2019). Each image $x$ has two labeled attributes summarized in the attribute vector $z = (y, a)$, where $y$ tells the class of the bird – Waterbird (WB) or Landbird (LB), and $a$ tells the type of the background – Water (W) or Land (L). Our training distribution consists of data from three groups – $(\text{WB}, \text{W})$, $(\text{LB}, \text{L})$, $(\text{LB}, \text{W})$. Our test distribution also consists of points from the remaining group $(\text{WB}, \text{L})$ as well. We seek to build class predictors that perform well on such test distributions that contain new groups. This problem setting differs from the commonly studied problem in Sagawa et al. (2019); Kirichenko et al. (2022),

---

[1]Github: facebookresearch/compositional-risk-minimization

where we observe data from all the groups but some groups present much more data than the others.

Formally, let $p(x, z) = p(z)p(x|z)$ denote the train distribution, and $q(x, z) = q(z)q(x|z)$ the test distribution. We denote the support of each attribute component $z_i$ under training distribution as $\mathcal{Z}_i^{\text{train}}$ and the support of $z$ under training distribution as $\mathcal{Z}^{\text{train}}$. The corresponding supports for the test distribution are denoted as $\mathcal{Z}_i^{\text{test}}$ and $\mathcal{Z}^{\text{test}}$. We define the Cartesian product of marginal support under training as $\mathcal{Z}^{\times} := \mathcal{Z}_1^{\text{train}} \times \mathcal{Z}_2^{\text{train}} \times \cdots \mathcal{Z}_m^{\text{train}}$.

In this work, we study *compositional shifts* from a training distribution $p$ to a test distribution $q$, characterized by:

1. $p(x|z) = q(x|z), \forall z \in \mathcal{Z}^{\times}$.

2. $\mathcal{Z}^{\text{test}} \nsubseteq \mathcal{Z}^{\text{train}}$ but $\mathcal{Z}^{\text{test}} \subseteq \mathcal{Z}^{\times}$.

The first point states that the conditional density of inputs conditioned on attributes remains invariant from train to test, which can be understood as the data generation mechanism from attributes to the inputs remains invariant. What changes between train and test is thus due to only shifting prior probabilities of attributes from $p(z)$ to $q(z)$. The second point specifies how these differ in their support: at test we observe novel combinations of individual attributes but not a completely new individual attribute. The task of compositional generalization is then to build classifiers that are robust to such compositional distribution shifts. Also, we remark that the above notion should remind the reader of the notion of Cartesian Product Extrapolation (CPE) from Lachapelle et al. (2024). Specifically, if a model succeeds on test distributions $q(z)$ with support equal to the full Cartesian product ($\mathcal{Z}^{\text{test}} = \mathcal{Z}^{\times}$), then it is said to achieve CPE.

## 2.2. Additive Energy Distribution

We assume that $p(x|z)$ is of the form of an *additive energy distribution* (AED):

$$p(x|z) = \frac{1}{\mathbb{Z}(z)} \exp\Big( - \sum_{i=1}^{m} E_i(x, z_i) \Big) \qquad (1)$$

where $\mathbb{Z}(z) := \int \exp\Big( - \sum_{i=1}^{m} E_i(x, z_i) \Big) dx$ is the partition function that ensures that the probability density $p(x|z)$ integrates to one. Also, the support of $p(x|z)$ is assumed to be $\mathbb{R}^n$, $\forall z \in \mathcal{Z}^{\times}$.

We thus have one energy term $E_i$ associated to each attribute $z_i$. Note that we do not make assumptions on $E_i$ except $\mathbb{Z}(z) < \infty$, leaving the resulting $p(x|z)$ very flexible. This form is a natural choice to model inputs that must satisfy a *conjunction* of characteristics (such as being a natural image of a landbird *AND* having a water background), corresponding to our attributes.

Recall $z = (z_1, \ldots, z_m)$ is a vector of $m$ categorical attributes that can each take $d$ possible values. We will denote as $\sigma(z)$ the representation of this attribute vector as a concatenation of $m$ one-hot vectors, i.e.

$$\sigma(z) = [\text{onehot}(z_1), \ldots, \text{onehot}(z_m)]^{\top}$$

Thus $\sigma(z)$ will be a sparse vector of length $md$ containing $m$ ones. We also define a vector valued map $E(x) = [E_1(x, 1), \ldots, E_1(x, d), \ldots, E_m(x, 1), \ldots, E_m(x, d)]^{\top}$ where $E_i(x, z_i)$ is the energy term for $i^{th}$ attribute taking the value $z_i$. This allows us to reexpress equation 1 using a simple dot product, denoted $\langle \cdot, \cdot \rangle$:

$$p(x|z) = \frac{1}{\mathbb{Z}(z)} \exp\Big( - \langle \sigma(z), E(x) \rangle \Big), \qquad (2)$$

where $\mathbb{Z}(z) = \int \exp\Big( - \langle \sigma(z), E(x) \rangle \Big) dx$ is the partition function.

There are two lines of work that inspire the choice of additive energy distributions. Firstly, these distributions have been used to enhance compositionality in generative tasks (Du et al., 2020; 2021; Liu et al., 2021) but they have not been used in discriminative compositionality. Secondly, for readers from the causal machine learning community, it may be useful to think of additive energy distributions from the perspective of the independent mechanisms principle (Janzing & Schölkopf, 2010; Parascandolo et al., 2018). The principle states that the data distribution is composed of independent data generation modules, where the notion of independence refers to algorithmic independence and not statistical independence. In these distributions, we think of energy function of an attribute as an independent function.

This is the right juncture to contrast AEDs with distributional assumptions in recent provable approaches to compositional generalization (Dong & Ma, 2022; Wiedemer et al., 2023; 2024; Brady et al., 2023; Lachapelle et al., 2024). These works assume labeling functions or decoders that are deterministic and additive over individual features, proving generalization over the Cartesian product of feature supports (further discussion in Appendix A). While insightful, this assumption is restrictive, as each attribute combination corresponds to a single observation with limited generative interactions. In contrast, AEDs capture stochastic decoders, offering a more flexible way to model inputs as a conjunction of characteristics (see Appendix B).

# 3. Provable Compositional Generalization

Our goal is to learn a model yielding a $\hat{q}(z|x)$ that will match the test distribution $q(z|x)$ and thus allow us to predict the attributes at test time in a Bayes optimal manner. If we successfully learn the distribution $q(z|x)$, then we can straightforwardly predict the individual attributes $q(z_i|x)$, e.g., the bird class in Waterbirds dataset, by marginalizing over the rest, e.g., the background in Waterbirds dataset. Observe that $q(z|x)$ differs from the training $p(z|x)$, which can be estimated through standard ERM with cross-entropy loss. Since some attributes $z$ observed at test time are never observed at train time, the distribution learned via ERM assigns a zero probability to these attributes and thus it cannot match the test distribution $q(z|x)$.

In what follows, we first introduce a novel mathematical object termed *Discrete Affine Hull* over the set of attributes. We then describe a generative approach for classification that requires us to learn $p(x|z)$ including the partition function, which is not practical. Next, we describe a purely discriminative approach that circumvents the issue of learning $\hat{p}(x|z)$ and achieves the same extrapolation guarantees as the generative approach. We present the generative approach as it allows to understand the results more easily. Building generative models based on our theory is out of scope of this work but is an exciting future work.

## 3.1. Discrete Affine Hull

We define the *discrete affine hull* of a set of attribute vectors $\mathcal{A} = \{z^{(1)}, \ldots, z^{(k)}\}$ where $z^{(i)} \in \mathcal{Z}$, defined $\text{DAff}(\mathcal{A})$ as:

$$\Big\{ z \in \mathcal{Z} \mid \exists\, \alpha \in \mathbb{R}^k, \sigma(z) = \sum_{i=1}^{k} \alpha_i \sigma(z^{(i)}), \sum_{i=1}^{k} \alpha_i = 1 \Big\}$$

In other words, the discrete affine hull of $\mathcal{A}$ consists of all attribute vectors whose one-hot encoding lies in the (regular) affine hull of the one-hot encodings of the attribute vectors of $\mathcal{A}$. This construct helps characterize which new attribute combinations we can extrapolate to.

As an illustration, consider the Waterbirds dataset, where we observe three of four possible groups. In one-hot encoding, WB is $[1, 0]$, LB is $[0, 1]$, Water is $[1, 0]$, and Land is $[0, 1]$. We show that the missing attribute vector WB on L, represented as $[1\ 0\ 0\ 1]$, can be expressed as an affine combination of the observed vectors, meaning the discrete affine hull of three one-hot concatenated vectors contains all four possible combinations.

$$(+1)\cdot\begin{bmatrix}0\\1\\0\\1\end{bmatrix} + (-1)\cdot\begin{bmatrix}0\\1\\1\\0\end{bmatrix} + (+1)\cdot\begin{bmatrix}1\\0\\1\\0\end{bmatrix} = \begin{bmatrix}1\\0\\0\\1\end{bmatrix} \quad (3)$$

In Section D.5, we generalize this finding to a formal mathematical characterization of discrete affine hulls, providing a visualization method. In Section D.7, we show how discrete affine hulls generalize the extrapolation of additive functions studied in Dong & Ma (2022); Lachapelle et al. (2024) over discrete domains. We also show how our results lead to a sharp characterization of extrapolation of these functions. Throughout, "affine hull" refers to the discrete affine hull.

### 3.2. Extrapolation of Conditional Density

We learn a set of conditional probability densities $\hat{p}(x|z) = \frac{1}{\hat{\mathbb{Z}}(z)}\exp\left(-\langle\sigma(z),\hat{E}(x)\rangle\right), \forall z \in \mathcal{Z}^{\text{train}}$ by maximizing the likelihood over the training distribution, where $\hat{E}$ denotes the estimated energy components and $\hat{\mathbb{Z}}$ denotes the estimated partition function. Under perfect maximum likelihood maximization $\hat{p}(x|z) = p(x|z)$ for all the training groups $z \in \mathcal{Z}^{\text{train}}$. We can define $\hat{p}(x|z)$ for all $z \in \mathcal{Z}^{\times}$ beyond $\mathcal{Z}^{\text{train}}$ in a natural way as follows. For each $z \in \mathcal{Z}^{\times}$, we have estimated the energy for every individual component $z_i$ denoted $\hat{E}_i(x, z_i)$. We set $\hat{\mathbb{Z}}(z) = \int \exp\left(-\langle\sigma(z),\hat{E}(x)\rangle\right)dx$ and the density for each $z \in \mathcal{Z}^{\times}$, $\hat{p}(x|z) = \frac{1}{\hat{\mathbb{Z}}(z)}\exp\left(-\langle\sigma(z),\hat{E}(x)\rangle\right)$.

**Theorem 1.** *If the true and learned distribution ($p(\cdot|z)$ and $\hat{p}(\cdot|z)$) are AED, then $\hat{p}(\cdot|z) = p(\cdot|z), \forall z \in \mathcal{Z}^{\text{train}} \implies \hat{p}(\cdot|z') = p(\cdot|z'), \forall z' \in \mathsf{DAff}(\mathcal{Z}^{\text{train}})$.*

The result above argues that so long as the group $z'$ is in the discrete affine hull of $\mathcal{Z}^{\text{train}}$, the estimated density extrapolates to it. We provide a proof sketch ahead, with the complete proof in Appendix D.1.

*Proof sketch:* Under perfect maximum likelihood maximization $\hat{p}(x|z) = p(x|z), \forall z \in \mathcal{Z}^{\text{train}}$. Replacing these densities by their expressions and taking their $\log$ we obtain

$$\langle\sigma(z),\hat{E}(x)\rangle = \langle\sigma(z),E(x)\rangle + C(z), \forall z \in \mathcal{Z}^{\text{train}} \quad (4)$$

where $C(z) = \log\left(\mathbb{Z}(z)/\hat{\mathbb{Z}}(z)\right)$.

For any $z' \in \mathsf{DAff}(\mathcal{Z}^{\text{train}})$, by definition there exists $\alpha$ such that $\sigma(z') = \sum_{z \in \mathcal{Z}^{\text{train}}} \alpha_z\sigma(z)$. Thus $\langle\sigma(z'),\hat{E}(x)\rangle = \sum_{z \in \mathcal{Z}^{\text{train}}}\alpha_z\langle\sigma(z),\hat{E}(x)\rangle$, by linearity of the dot product. Substituting the expression for $\langle\sigma(z),\hat{E}(x)\rangle$ from equation 4, this becomes

$$\langle\sigma(z'),\hat{E}(x)\rangle = \sum_{z \in \mathcal{Z}^{\text{train}}} \alpha_z\left(\langle\sigma(z),E(x)\rangle + C(z)\right)$$
$$= \langle\sigma(z'),E(x)\rangle + \sum_{z \in \mathcal{Z}^{\text{train}}}\alpha_z C(z), \quad (5)$$

From equation 5, we can conclude that $\langle\sigma(z'),\hat{E}(x)\rangle$ estimates $\langle\sigma(z'),E(x)\rangle$ perfectly up to a constant error that does not depend on $x$. This difference of constant is absorbed by the partition function and hence the conditional densities match: $\hat{p}(x|z') = p(x|z')$.

**Classifier based on conditional density** $p(x|z)$**.** If, on data from training distribution $p$, we were able to train a good conditional density estimate $\hat{p}(x|z), \forall z \in \mathcal{Z}^{\text{train}}$, then Theorem 1 implies that $\hat{p}(x|z')$ will also be a good estimate of $p(x|z')$ for *new unseen* attributes $z' \in \mathsf{DAff}(\mathcal{Z}^{\text{train}})$. Provided $\mathcal{Z}^{\text{test}} \subseteq \mathsf{DAff}(\mathcal{Z}^{\text{train}})$, it is then straightforward to obtain a classifier that generalizes to compositionally-shifted test distribution $q$. Indeed, we have

$$q(z'|x) = \frac{q(x|z')q(z')}{\sum_{z'' \in \mathcal{Z}^{\text{test}}} q(x|z'')q(z'')}$$
$$= \frac{p(x|z')q(z')}{\sum_{z'' \in \mathcal{Z}^{\text{test}}} p(x|z'')q(z'')} \approx \frac{\hat{p}(x|z')q(z')}{\sum_{z'' \in \mathcal{Z}^{\text{test}}} \hat{p}(x|z'')q(z'')} \quad (6)$$

where we used the property of compositional shifts $q(x|z) = p(x|z)$. If we know test group prior $q(z')$ (or e.g. assume it to be uniform), we can directly use the expression in RHS to correctly compute the test group probabilities $q(z|x)$, even for groups never seen at training.

### 3.3. Extrapolation of Discriminative Model

In Section 3.2, we saw how we could, in principle, obtain a classifier that generalizes under compositional shift, by first training energy based conditional probability density models $\hat{p}(x|z)$. However learning such a model requires dealing with the problematic partition function throughout training. Indeed making a gradient step to maximize its log likelihood with respect to parameters $\theta$ involves estimating the gradient of its log partition function $\nabla_\theta \log \hat{\mathbb{Z}}(z;\theta) = \nabla_\theta \log \int \exp\left(-\langle\sigma(z),\hat{E}(x;\theta)\rangle\right)dx$ which is typically intractable. This difficulty in training energy-based models is a well known open problem. While crude stochastic approximations of this gradient might be obtained via e.g. Contrastive Divergence (Hinton, 2002) or variants of more expensive MCMC sampling, no unbiased computationally efficient solution is known in the general case.

But is it really necessary to precisely model the conditional density of high dimensional $x$, when our goal is simply to predict a few classes and attributes $z$, given $x$? We will now develop an alternative approach, *Compositional Risk Minimization* (CRM), that achieves a similar extrapolation result as Theorem 1, while being based on simple discriminative classifier training. It sidesteps the need and difficulties of explicitly modeling $p(x|z)$ and doesn't require dealing with the partition function throughout training.

Observe that if we apply Bayes rule to the AED $p(x|z)$ in equation 2, we get

$$p(z|x) = \frac{p(x|z)p(z)}{\sum_{z' \in \mathcal{Z}^{\text{train}}} p(x|z')p(z')}$$

$$= \frac{\exp\left( - \langle \sigma(z), E(x) \rangle + \log p(z) - \log \mathbb{Z}(z) \right)}{\sum_{z' \in \mathcal{Z}^{\text{train}}} \exp\left( - \langle \sigma(z'), E(x) \rangle + \log p(z') - \log \mathbb{Z}(z') \right)}$$

We thus define our *additive energy classifier* as follows. To guarantee that we can model this $p(z|x)$, we use a model with the same *form*. For each $z \in \mathcal{Z}^{\text{train}}$

$$\tilde{p}(z|x) = \frac{\exp\left( - \langle \sigma(z), \tilde{E}(x) \rangle + \log \hat{p}(z) - \tilde{B}(z) \right)}{\sum_{z' \in \mathcal{Z}^{\text{train}}} \exp\left( - \langle \sigma(z'), \tilde{E}(x) \rangle + \log \hat{p}(z') - \tilde{B}(z') \right)} \tag{7}$$

where $\hat{p}(z)$ is the empirical estimate of the prior over $z$, i.e., $p(z)$, $\tilde{E} : \mathbb{R}^n \to \mathbb{R}^{md}$ is a function to be learned, bias $\tilde{B}$ is a lookup table containing a learnable offset for each combination of attribute. Given a data point $(x, z)$, loss $\ell(z, \tilde{p}(\cdot|x)) = -\log \tilde{p}(z|x)$ measures the prediction performance of $\tilde{p}(\cdot|x)$. The risk, defined as the expected loss, corresponds to the negated conditional log-likelihood:

$$R(\tilde{p}) = \mathbb{E}_{(x,z) \sim p}\left[ \ell(z, \tilde{p}(\cdot|x)) \right] = \mathbb{E}_{(x,z) \sim p}\left[ - \log \tilde{p}(z|x) \right] \tag{8}$$

In the first step of CRM, we minimize the risk $R$.

$$\hat{E}, \hat{B} \in \underset{\tilde{E}, \tilde{B}}{\arg\min}\, R(\tilde{p}) \tag{9}$$

If the minimization is over arbitrary functions, then $\hat{p}(\cdot|x) = p(\cdot|x), \forall x \in \mathbb{R}^n$. In the second step of CRM, we compute our final predictor $\hat{q}(z|x)$ as follows. Let $\hat{q}(z)$ be an estimate of the marginal distribution over the attributes $q(z)$ with support $\hat{\mathcal{Z}}^{\text{test}}$. For each $z \in \mathcal{Z}^{\text{test}}$

$$\hat{q}(z|x) = \frac{\exp\left( - \langle \sigma(z), \hat{E}(x) \rangle + \log \hat{q}(z) - B^\star(z) \right)}{\sum_{z' \in \hat{\mathcal{Z}}^{\text{test}}} \exp\left( - \langle \sigma(z'), \hat{E}(x) \rangle + \log \hat{q}(z') - B^\star(z') \right)} \tag{10}$$

where, $B^\star$ is the *extrapolated bias* defined as $B^\star(z) =$

$$\log \mathbb{E}_{x \sim p}\left[ \frac{\exp\left( - \langle \sigma(z), \hat{E}(x) \rangle \right)}{\sum_{\tilde{z} \in \mathcal{Z}^{\text{train}}} \exp\left( - \langle \sigma(\tilde{z}), \hat{E}(x) \rangle + \log p(\tilde{z}) - \hat{B}(\tilde{z}) \right)} \right] \tag{11}$$

where $\hat{E}, \hat{B}$ are the solutions from optimization equation 9. Note that $\hat{B}(z)$ was learned for all $z \in \mathcal{Z}^{\text{train}}$ but never for $z \in \mathcal{Z}^{\text{test}}$, hence the necessity of extrapolation $B^*$. Each of these steps is easy to operationalize. We explain the process and provide pseudocode in Section 4 .

**Theorem 2.** *Consider the setting where $p(.|z)$ follows AED $\forall z \in \mathcal{Z}^\times$, the test distribution $q$ satisfies compositional shift characterization and $\mathcal{Z}^{\text{test}} \subseteq \mathsf{DAff}(\mathcal{Z}^{\text{train}})$. If $\hat{p}(z|x) = p(z|x), \forall z \in \mathcal{Z}^{\text{train}}, \forall x \in \mathbb{R}^n$ and $\hat{q}(z) = q(z), \forall z \in \mathcal{Z}^{\text{test}}$, then the output of CRM (equation 10) matches the test distribution, i.e., $\hat{q}(z|x) = q(z|x), \forall z \in \mathcal{Z}^{\text{test}}, \forall x \in \mathbb{R}^n$.*

A complete proof is provided in Appendix D.2. Observe that $\hat{p}(\cdot|x) = p(\cdot|x)$ is a condition that even a model trained via ERM can satisfy (with sufficient capacity and data) but it cannot match the true $q(\cdot|x)$. In contrast, CRM optimally adjusts the additive-energy classifier for the compositional shifts. CRM requires the knowledge of test prior $q(z)$ but the choice of uniform distribution over all possible groups is a reasonable one to make in the absence of further knowledge. Notice how learned bias $\hat{B}(z)$ can only be fitted for $z \in \mathcal{Z}^{\text{train}}$, remaining undefined for $z' \notin \mathcal{Z}^{\text{train}}$. But we can compute the extrapolated bias $B^\star(z'), \forall z' \in \mathcal{Z}^{\text{test}}$, *based remarkably on only data from the train distribution.*

**Illustrating CRM's adaptation to test distribution.** To better convey how CRM can adapt to the Bayes optimal classifier of the test distribution, we provide an example. Consider a two-dimensional setting, where the distribution of $x \in \mathbb{R}^2$ conditioned on the attributes $z_1 \in \{-1, 1\}$ and $z_2 \in \{-1, 1\}$ is a Gaussian with mean $(z_1, z_2)$ and identity covariance. Suppose the training groups are drawn with equal probability and can take one of the following three possible values $(+1, +1), (-1, +1), (+1, -1)$. We do not observe data from the group $(-1, -1)$ during training, but at test time we draw samples from all the four groups with equal probability. First, we can show that the above distribution can be expressed as an additive energy distribution, as $E(x, (z_1, z_2)) = \frac{1}{4}\|x - (2z_1, 0)\|^2 + \frac{1}{4}\|x - (0, 2z_2)\|^2)$. For the task of classifying groups, the Bayes optimal classifier has a closed form solution, where each decision region is an intersection of two half-spaces. Figure 2 shows that CRM learns the Bayes optimal classifier on the training distribution, enables shifting from train prior to test prior to yield the Bayes optimal classifier for the test distribution, and correctly generalize to the unseen $(-1 - 1)$ group. For further details, including illustration of the failure of ERM-trained binary classifier on this problem, see Appendix E. Beyond

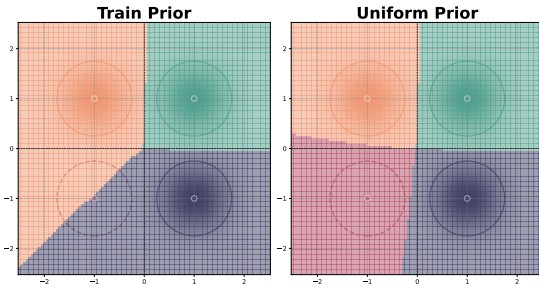

*Figure 2.* **Extrapolating to an unseen test group**. The distribution to model corresponds to a mixture of 4 Gaussians. But the $(-1, -1)$ group (pink dashed) has zero prior probability in the training distribution, i.e., is absent from the training set. As a result, discriminative training (left) learns only three decision regions and would misclassify test points from $(-1, -1)$. CRM adjusts the prior at test time (right) to a uniform distribution over all attribute combinations, enabling it to recover four decision regions and correctly generalize to the unseen $(-1, -1)$ group. The additive energy form composes the attributes' information to yield this unseen group's location. Decision regions were obtained from finite-data simulations, leading to minor imperfections.

this toy example, we highlight that the additive energy form supports modeling nearly arbitrarily complex distributions.

**Analyzing growth of Discrete Affine Hull.** In the discussion so far, we have relied on a crucial assumption that the attribute combinations in the test distribution are in the affine hull. Is this also a *necessary* condition? Can we generalize to attributes outside the affine hull? We consider the task of learning $p(\cdot|z)$ from Theorem 1 and the task of learning $q(\cdot|x)$ from Theorem 2. In Appendix D.6, we show that the restriction to affine hulls is indeed necessary.

Under the assumption of compositional shifts $\mathcal{Z}^{\mathsf{test}}$ is only restricted to be a subset of the Cartesian product set $\mathcal{Z}^{\times}$, but our results so far have required us to restrict the support further by confining it to the affine hull, i.e., $\mathcal{Z}^{\mathsf{test}} \subseteq \mathsf{DAff}(\mathcal{Z}^{\mathsf{train}}) \subseteq \mathcal{Z}^{\times}$. This leads us to a natural question. If the training groups that form $\mathcal{Z}^{\mathsf{train}}$ are drawn at random, then how many groups do we need such that the affine hull captures $\mathcal{Z}^{\times}$, i.e., $\mathsf{DAff}(\mathcal{Z}^{\mathsf{train}}) = \mathcal{Z}^{\times}$, at which point CRM can achieve Cartesian Product Extrapolation (CPE). Another way to think about this is to say, how fast does the affine hull grow and capture the Cartesian product set $\mathcal{Z}^{\times}$? Next, we answer this question.

Consider the the general setting with $m$ attributes, where each attribute takes $d$ possible values, leading to $d^m$ possible attribute combinations. Suppose we sample $s$ attribute vectors $z$ that comprise the support $\mathcal{Z}^{\mathsf{train}}$ uniformly at random (with replacement) from these $d^m$ possibilites. In the next theorem, we show that if the number of sampled attribute vectors exceeds $2c(md + d\log(d))$, then $\mathsf{DAff}(\mathcal{Z}^{\mathsf{train}})$ con-

tains all the possible $d^m$ combinations with a high probability (greater than $1 - \frac{1}{c}$), hence CRM achieves CPE. We want to emphasize this surprising finding: with almost a linear growth in $m$ and $d$, CRM generalizes to exponentially many $d^m$ groups.

**Theorem 3.** *Consider the setting where $p(.|z)$ follows AED $\forall z \in \mathcal{Z}^{\times}$, $\mathcal{Z}^{\mathsf{train}}$ comprises of $s$ attribute vectors $z$ drawn uniformly at random from $\mathcal{Z}^{\times}$, and the test distribution $q$ satisfies compositional shift characterization with $\mathcal{Z}^{\mathsf{test}} = \mathcal{Z}^{\times}$. If $s \geq 2c(md + d\log(d))$, where $d$ is sufficiently large, $\hat{p}(z|x) = p(z|x), \forall z \in \mathcal{Z}^{\mathsf{train}}, \forall x \in \mathbb{R}^n$, $\hat{q}(z) = q(z), \forall z \in \mathcal{Z}^{\times}$, then the output of CRM (equation 10) matches the test distribution, i.e., $\hat{q}(z|x) = q(z|x)$, $\forall z \in \mathcal{Z}^{\times}, \forall x \in \mathbb{R}^n$, with probability greater than $1 - \frac{1}{c}$.*

In Appendix D.3, we first present the proof for the case with $m = 2$ attributes and provide visual illustrations to assist the reader. It is then followed by the more involved proof for the general case of $m$ attributes in Appendix D.4.

### 3.4. Further Insights on CRM

**Does test distribution belong to the affine hull of train distributions?** A key implication of the AED assumption is that the energy for a novel group $z' \in \mathsf{DAff}(\mathcal{Z}^{\mathsf{train}})$ at test time can be expressed as an affine combination of the energies of the training groups, i.e., $\langle \sigma(z'), E(x) \rangle = \sum_{z \in \mathcal{Z}^{\mathsf{train}}} \alpha_z \langle \sigma(z), E(x) \rangle$ (check Lemma 1 in Appendix D for details). However, this does not imply that the conditional density $q(x|z')$ is an affine combination of the training conditional densities $\{p(x|z) \mid z \in \mathcal{Z}^{\mathsf{train}}\}$. Instead, we have the following relationship.

$$
\begin{aligned}
\log\big(q(x|z')\big) = &\sum_{z \in \mathcal{Z}^{\mathsf{train}}} \alpha_z \log p(x|z) \\
&- \log \int \exp\Big( \sum_{z \in \mathcal{Z}^{\mathsf{train}}} \alpha_z \log p(x|z) \Big) dx
\end{aligned}
\tag{12}
$$

Please check Appendix D.2 for the derivation. In contrast, prior works impose a stronger assumption that the test distribution should lie in the convex/affine hull of the train distributions (Krueger et al., 2021; Qiao & Peng, 2023; Yao et al., 2023).

**Why Additive Energy Classifier?** In Appendix D.2, for novels group $z' \in \mathsf{DAff}(\mathcal{Z}^{\mathsf{train}})$, we derive the following relationship between the test classifier for the novel group $q(z'|x)$ and the training classifiers $\{p(z|x) \mid z \in \mathcal{Z}^{\mathsf{train}}\}$.

$$
\begin{aligned}
q(z'|x) = \mathsf{Softmax}\Big( &\log q(z') + \sum_{z \in \mathcal{Z}^{\mathsf{train}}} \alpha_z \log p(z|x) \\
&- \log \mathbb{E}_{x \sim p(x)}\Big[ \exp\Big( \sum_{z \in \mathcal{Z}^{\mathsf{train}}} \alpha_z \log p(z|x) \Big) \Big] \Big)
\end{aligned}
\tag{13}
$$

This equation is central in deriving the classifier $\hat{q}(z'|x)$ (equation 10) in the second step of CRM, where we substitute $p(z|x)$ with the learned additive energy classifier $\hat{p}(z|x)$ (equation 9). While alternative methods could be used to estimate $p(z|x)$, but they would have to separately infer the affine combination weights $\alpha_z$. To address this, we adopt the additive energy classifier (equation 7), which simplifies the computation of $\hat{q}(z'|x)$. Crucially, it avoids the need to explicitly estimate the affine weights ($\alpha_z$), rather the required adjustment is absorbed into a single updated bias term $B^\star$ (equation 11).

**Why Compositional Risk?** To better understand the compositional risk formulation, note that the classical ERM objective (8) can be restated as follows,

$$R(\tilde{p}) = \sum_{z \in \mathcal{Z}^\times} p(z) R(\tilde{p}|z)$$

where $R(\tilde{p}|z) = \mathbb{E}_{x \sim p(x|z)}\big[\ell(z, \tilde{p}(\cdot|x))\big]$. Note that in the above summation $p(z)$ is zero on all groups that are not in the support of the training distribution. However, to tackle compositional shifts, we want to learn predictors that instead minimize the following *compositional risk*,

$$\begin{aligned} R_{\text{comp}}(\tilde{p}) &= \mathbb{E}_{z \sim q(z)} \mathbb{E}_{x \sim q(x|z)}\big[\ell(z, \tilde{p}(\cdot|x))\big] \\ &= \sum_{z \in \mathcal{Z}^\times} q(z) R(\tilde{p}|z) \end{aligned} \quad (14)$$

as we have $p(x|z) = q(x|z) \; \forall z \in \mathcal{Z}^\times$. In the above objective, $q(z)$ can be non-zero on groups $z$ that have zero probability under $p(z)$. Hence, the minimizer of expected risk $R(\tilde{p})$ can be different from the minimizer of compositional risk $R_q(\tilde{p})$. In Theorem 3, we show that our approach (CRM) outputs the Bayes optimal predictor and hence it provably minimizes $R_{comp}(\tilde{p})$.

## 4. Algorithm for CRM

In a nutshell, CRM consists of: a) training additive energy classifier $\hat{p}(z|x)$ (e.q. 7) by maximum likelihood (e.q.9) for trainset group prediction; b) compute extrapolated biases $B^\star$ (e.q.11); c) infer group probabilities on compositionally shifted test distribution using $\hat{q}(z|x)$ (e.q 10). Algorithm 1 provides the associated pseudo-code, where we have a basic architecture using a deep network backbone $\phi(x; \theta)$ followed by a linear mapping (matrix $W$) [2]. For the case where we have 2 attributes $z = (y, a)$ (illustrated in Figure 1), we provide a detailed algorithm and PyTorch implementation in Appendix C.

---

[2]Instead of linear, we could use separate non-linear heads to obtain the energy components for each attribute.

---

**Algorithm 1** Compositional Risk Minimization (CRM)

**Input:** Training set $\mathcal{D}^{\text{train}} = \{(x, z)\}$
**Output:** Classifier parameters $\hat{\theta}, \hat{W}, B^\star$
**Training:**

- Estimate train prior $\hat{p}(z)$ based on group counts in $\mathcal{D}^{\text{train}}$

- Compute the energy terms as $\tilde{E}(x) = \tilde{W}\phi(x; \tilde{\theta})$

- **CRM Step 1:** Train additive energy classifier $\tilde{p}$ (e.q. 7) by emprirical risk minimization: $\hat{\theta}, \hat{W}, \hat{B} \in \arg\min_{\tilde{\theta}, \tilde{W}, \tilde{B}} R(\tilde{p})$

- **CRM Step 2:** Estimate extrapolated bias $B^\star$ (e.q. 11) via an average over training examples.

**Inference on test point $x$:**

- Set $\hat{q}(z)$ as uniform prior over all groups.

- Compute test group probabilities $\hat{q}(z|x)$ via e.q. 10, using $\hat{E}$ and $B^\star$ learned during training.

---

## 5. Experiments

### 5.1. Setup

We evaluate CRM on widely recognized benchmarks for subpopulation shifts (Yang et al., 2023b), that have attributes $z = (y, a)$, where $y$ denotes the class label and $a$ denotes the spurious attribute ($y$ and $a$ are correlated). However, the standard split between train and test data mandated in these benchmarks does not actually evaluate compositional generalization capabilities, because both train and test datasets contain all the groups ($\mathcal{Z}^{\text{train}} = \mathcal{Z}^{\text{test}} = \mathcal{Z}^\times$). Therefore, we repurpose these benchmarks for compositional shifts by discarding samples from one of the groups ($z$) in the train (and validation) dataset; but we don't change the test dataset, i.e., $z \notin \mathcal{Z}^{\text{train}}$ but $z \in \mathcal{Z}^{\text{test}}$. Let us denote the data splits from the standard benchmarks as $(\mathcal{D}_{\text{train}}, \mathcal{D}_{\text{val}}, \mathcal{D}_{\text{test}})$. Then we generate multiple variants of compositional shifts $\{(\mathcal{D}_{\text{train}}^{\neg z}, \mathcal{D}_{\text{val}}^{\neg z}, \mathcal{D}_{\text{test}}) \mid z \in \mathcal{Z}^\times\}$, where $\mathcal{D}_{\text{train}}^{\neg z}$ and $\mathcal{D}_{\text{val}}^{\neg z}$ are generated by discarding samples from $\mathcal{D}_{\text{train}}$ and $\mathcal{D}_{\text{val}}$ that belong to the group $z$.

Following this procedure, we adapted Waterbirds (Wah et al., 2011), CelebA (Liu et al., 2015), MetaShift (Liang & Zou, 2022), MultiNLI (Williams et al., 2017), and CivilComments (Borkan et al., 2019) for experiments. We also experiment with the NICO++ dataset (Zhang et al., 2023), where we already have $\mathcal{Z}^{\text{train}} \subsetneq \mathcal{Z}^{\text{test}} = \mathcal{Z}^\times$ as some groups were not present in the train dataset. However, these groups are still present in the validation dataset ($\mathcal{Z}^{\text{val}} = \mathcal{Z}^\times$). Hence, the only transformation we apply to NICO++ is to drop samples from the validation dataset so that $\mathcal{Z}^{\text{train}} = \mathcal{Z}^{\text{val}}$. Note that our benchmarks cover diverse scenarios, with binary (Waterbirds, CelebA, MetaShift) and non-binary attributes (MultiNLI, CivilComments, NICO++), resulting in total groups varying from 4 to 360 (NICO++).

For baselines, we train classifiers via ERM, Group Dis-

tributionally Robust Optimization (GroupDRO) (Sagawa et al., 2019), Logit Correction (LC) (Liu et al., 2022b), supervised logit adjustment (sLA) (Tsirigotis et al., 2024), Invariant Risk Minimization (IRM) (Arjovsky et al., 2019), Risk Extrapolation (VREx) (Krueger et al., 2021), and Mixup (Zhang et al., 2017). In all cases we employ a pre-trained architecture as the representation network $\phi$, followed by a linear layer $W$ to get class predictions, and fine-tune them jointly (see Appendix F.3 for details).

For evaluation metrics, we computed the average accuracy, group-balanced accuracy, and worst-group accuracy (WGA) on the test set. Due to imbalances in group distribution, a method can obtain good average accuracy despite having bad worst-group accuracy. Therefore, WGA is more indicative of robustness to spurious correlations (Appendix F.2).

## 5.2. Results

Table 1 shows the results of our experiment, where due to space constraints, we only compare with the best performing baselines and don't report the group balanced accuracy and standard error over random seeds. Complete results with comparisons with IRM, VREx, and Mixup are provided in Appendix G.1 (Table 5). For each dataset and metric, we report the *average* performance over its various compositional shift scenarios $\{(\mathcal{D}_{\text{train}}^{\neg z}, \mathcal{D}_{\text{val}}^{\neg z}, \mathcal{D}_{\text{test}}) \mid z \in \mathcal{Z}^{\times}\}$ (detailed results for all scenarios are in Appendix G.2). In all cases, CRM either outperforms or is competitive with the baselines in terms of worst group accuracy (WGA).

Further, for Waterbirds and MultiNLI, while the logit adjustment baselines appear competitive with CRM on average, if we look more closely at the worst case compositional shift scenario, we find these baselines fare much worse than CRM. For Waterbirds, LC obtains 69.0% WGA while CRM obtains 73.0% WGA for the worst case scenario of dropping the group $(0, 1)$ (Table 6). Similarly, for MultiNLI, sLA obtains 19.7% WGA while CRM obtains 31.0% WGA for the worst case scenario of dropping the group $(0, 0)$ (Table 9).

We also report the worst group accuracy (other metrics in Table 14, Appendix G.5) for the original benchmark $(\mathcal{D}_{\text{train}}, \mathcal{D}_{\text{val}}, \mathcal{D}_{\text{train}})$, which was not transformed for compositional shifts, denoted WGA (No Groups Dropped). This can be interpreted as the "oracle" performance for that benchmark, and we can compare methods based on the performance drop in WGA due to discarding groups in compositional shifts. ERM and GroupDRO appear the most sensitive to compositional shifts, and the logit adjustment baselines also show a sharp drop for the CelebA benchmark; while CRM is more robust to compositional shifts.

**Multiple spurious attributes case.** We benchmark CRM for class label prediction with multiple spurious attributes. For this, we augment the CelebA benchmark to have three

| Dataset | Method | Average Acc | WGA | WGA (No Groups Dropped) |
|---|---|---|---|---|
| Waterbirds | ERM | 77.9 | 43.0 | 62.3 |
| | G-DRO | 77.9 | 42.3 | 87.3 |
| | LC | 88.3 | 75.5 | 88.7 |
| | sLA | 89.3 | 77.3 | 89.7 |
| | CRM | 87.1 | 78.7 | 86.0 |
| CelebA | ERM | 85.8 | 39.0 | 52.0 |
| | G-DRO | 89.2 | 67.8 | 91.0 |
| | LC | 91.1 | 57.4 | 90.0 |
| | sLA | 90.9 | 57.4 | 86.7 |
| | CRM | 91.1 | 81.8 | 89.0 |
| MetaShift | ERM | 85.7 | 60.5 | 63.0 |
| | G-DRO | 86.0 | 63.8 | 80.7 |
| | LC | 88.5 | 68.2 | 80.0 |
| | sLA | 88.4 | 63.0 | 80.0 |
| | CRM | 87.6 | 73.4 | 74.7 |
| MultiNLI | ERM | 68.4 | 7.5 | 68.0 |
| | G-DRO | 70.4 | 34.3 | 57.0 |
| | LC | 75.9 | 54.3 | 74.3 |
| | sLA | 76.4 | 55.0 | 71.7 |
| | CRM | 74.3 | 58.7 | 74.7 |
| Civil Comments | ERM | 80.4 | 55.9 | 61.0 |
| | G-DRO | 80.1 | 61.6 | 64.7 |
| | LC | 80.7 | 65.7 | 67.3 |
| | sLA | 80.6 | 65.6 | 66.3 |
| | CRM | 83.7 | 67.9 | 70.0 |
| NICO++ | ERM | 85.0 | 35.3 | 35.3 |
| | G-DRO | 84.0 | 36.7 | 33.7 |
| | LC | 85.0 | 35.3 | 35.3 |
| | sLA | 85.0 | 33.0 | 35.3 |
| | CRM | 84.7 | 40.3 | 39.0 |

*Table 1.* **Robustness under compositional shift.** We compare the proposed CRM method to baseline ERM classifier training with no group information, and to robust methods that leverage group labels: G-DRO, LC, and sLA. We report test Average Accuracy and Worst Group Accuracy (WGA) (mean over 3 random seeds), averaged as a group is dropped from training and validation sets. Last column is WGA under the dataset's standard subpopulation shift benchmark, i.e. with no group dropped. All methods have a harder time to generalize when groups are absent from training, but CRM appears consistently more robust. Full results with balanced group accuracy, standard error across random seeds, and baselines IRM, VREx, and Mixup in Appendix (Table 5).

additional attributes (total $m = 5$ attributes). We provide setup details in Appendix G.3, and Table 12 presents the associated results. We find that CRM still continues to be the superior method (as per WGA), in fact, the difference in WGA between CRM and runner up baseline increases to 28.9%, as compared to 14% for the case of CelebA 2-attribute (Table 2). Hence, the benefits with CRM become more pronounced for multiple spurious attributes settings.

**Importance of extrapolating the bias.** We conduct an ablation study for CRM where we test a variant that uses

| Method | Waterbirds | CelebA | MetaShift | MulitNLI | CivilComments | NICO++ |
|--------|-----------|--------|-----------|----------|---------------|--------|
| CRM ($\hat{B}$) | 55.7 (1.0) | 58.9 (0.4) | 58.7 (0.6) | 30.4 (2.6) | 52.4 (0.7) | 31.0 (1.0) |
| CRM | 78.7 (1.6) | 81.8 (1.2) | 73.4 0.7) | 58.7 (1.4) | 67.9 (0.5) | 40.3 (4.3) |

*Table 2.* **Importance of bias extrapolation.** We report Worst Group Accuracy, averaged as a group is dropped from training and validation (standard error based on 3 random seeds). CRM ($\hat{B}$) is an ablated version of CRM where we use the trained bias $\hat{B}$ instead of the extrapolated bias $B^\star$ mandated by our theory. The extrapolation step appears crucial for robust compositional generalization. Merely adjusting logits based on shifting group prior probabilities does not suffice.

the learned bias $\hat{B}$ (e.q. 9) instead of the extrapolated bias $B^\star$ (e.q. 11). Results are presented in Table 2. They show a significant drop in worst-group accuracy if we use the learned bias instead of the extrapolated one. Hence, our theoretically grounded bias extrapolation step is crucial to generalization under compositional shifts. In Appendix G.4 (Table 13) we conduct further ablation studies, showing the impact of different choices of the test log prior.

Further, we analyze CRM's performance with varying number of train groups, details in Appendix G.6.

# 6. Related works

Due to space constraints, we briefly describe the relevant prior works, with a detailed discussion in Appendix A.

**Compositional Generalization** Compositionality has long been seen as an essential capability (Fodor & Pylyshyn, 1988; Hinton, 1990; Plate et al., 1991; Montague, 1970) on the path to building human-level intelligence. The history of compositionality being too long to cover in detail here, we refer the reader to these surveys (Lin et al., 2023; Sinha et al., 2024). Most prior works have focused on generative aspect of compositionality, where the model needs to recombine individual distinct factors/concepts and generate the final output in the form of text (Gordon et al., 2019; Lake & Baroni, 2023) or image (Liu et al., 2022a; Wang et al., 2024). For image generation in particular, a fruitful line of work is rooted in additive energy based models (Du et al., 2020; 2021; Liu et al., 2021; Nie et al., 2021), which translates naturally to additive diffusion models (Liu et al., 2022a; Su et al., 2024). Our present work also leverages an additive energy form, but our focus is on learning classifiers robust under compositional shifts, rather than generative models.

**Domain Generalization** Generalization under subpopulation shifts, where certain groups or combinations of attributes are underrepresented in the training data, is a well-known challenge in machine learning. GroupDRO (Sagawa et al., 2019) is a prominent method that minimizes the worst-case group loss to improve robustness across groups. IRM (Arjovsky et al., 2019) encourages the model to learn invariant representations that perform well across multiple environments. Closely related to our proposed method are the logit adjustment methods, LC (Liu et al., 2022b)

and sLA (Tsirigotis et al., 2024) that use logit adjustment for group robustness. There are many other interesting approaches that were proposed, see the survey (Zhou et al., 2022) for details. The theoretical guarantees developed for these approaches (Arjovsky et al., 2019; Rosenfeld et al., 2020; Ahuja et al., 2020) require a large diversity in terms of the environments seen at the training time. In our setting, we incorporate inductive biases based on additive energy distributions that help us arrive at provable generalization with limited diversity in the environments.

# 7. Conclusion

We provide a novel approach (CRM) based on flexible additive energy models for compositionality in discriminative tasks. CRM can provably extrapolate to novel attribute combinations within the discrete affine hull of the training support, where the affine hull grows quickly with the training groups to cover the Cartesian product extension of the training support. Our empirical results demonstrate that the additive energy assumption is sufficiently flexible to yield good classifiers for high-dimensional images, and CRM is able to extrapolate to novel combinations in DAff($\mathcal{Z}^{\text{train}}$), without having to model high-dimensional $p(x|z)$ nor having to estimate their partition function. CRM is a simple and efficient algorithm that empirically proved consistently more robust to compositional shifts than approaches based on other logit-shifting schemes and GroupDRO.

**Limitations.** A key limitation of our work is the reliance on labeled attributes, which may not always be available in practice. Prior works (Pezeshki et al., 2023; Tsirigotis et al., 2024) propose to first infer spurious attributes and then use domain generalization methods, in the context of subpopulation shifts. A promising direction for future research is to extend such methods to the compositional shift setting and integrate them with our approach.

Further, based on AED, we provided compositional generalization guarantees that are applicable in both discriminative and generative tasks. However, we only offer a tractable approach using CRM for the discriminative task. Extending these ideas to the generative counterpart, e.g. by starting from equation 12, remains a promising direction for future work and could further leverage our theoretical framework.

## Acknowledgements

We thank Yash Sharma for his contribution to early exploration of compositional generalization from a generative perspective, and Vinayak Tantia for having many years ago helped shed light on the challenges posed by compositionality in discriminative training. This research was entirely funded by Meta, in the context of Meta's AI Mentorship program with Mila. Ioannis Mitliagkas in his role as Divyat Mahajan's academic advisor, acknowledges support by an NSERC Discovery grant (RGPIN-2019-06512), and a Canada CIFAR AI chair. Pascal Vincent is a CIFAR Associate Fellow in the Learning in Machines & Brains program. Divyat Mahajan acknowledges support via FRQNT doctoral scholarship (https://doi.org/10.69777/354785) for his graduate studies.

## Impact Statement

This paper presents work whose goal is to advance the field of Machine Learning. There are many potential societal consequences of our work, none which we feel must be specifically highlighted here.

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

# Appendix

## List of Contents

The content in the Appendix has been organized as follows.

# A. Further Discussion on Related Works

**Compositional Generalization.** Compositionality has long been seen as an important capability on the path to building (Fodor & Pylyshyn, 1988; Hinton, 1990; Plate et al., 1991; Montague, 1970) human-level intelligence. The history of compositionality is very long to cover in detail here, refer to these surveys (Lin et al., 2023; Sinha et al., 2024) for more detail. Compositionality is associated with many different aspects, namely systematicity, productivity, substitutivity, localism, and overgeneralization (Hupkes et al., 2020). In this work, we are primarily concerned with systematicity, which evaluates a model's capability to understand known parts or rules and combine them in new contexts. Over the years, several popular benchmarks have been proposed to evaluate this systematicity aspect of compositionality, (Lake & Baroni, 2018) proposed the SCAN dataset, (Kim & Linzen, 2020) proposed the COGS dataset. These works led to development of several insightful approaches to tackle the challenge of compositionality (Lake & Baroni, 2023; Gordon et al., 2019). Most of these works on systematicity have largely focused on generative tasks, (Liu et al., 2022a; Lake & Baroni, 2023; Gordon et al., 2019; Wang et al., 2024), i.e., where the model needs to recombine individual distinct factors/concepts and generate the final output in the form of image or text. There has been lesser work on discriminative tasks (Nikolaus et al., 2019), i.e., where the model is given an input composed of a novel combination of factors and it has to predict the underlying novel combination. In this work, our focus is to build an approach that can provably solve these discriminative tasks.

On the theoretical side, recently, there has been a growing interest to build provable approaches for compositional generalization (Wiedemer et al., 2023; 2024; Brady et al., 2023; Dong & Ma, 2022; Lachapelle et al., 2024). These works study models where the labeling function or the decoder is additive over individual features, and prove generalization guarantees over the Cartesian product of the support of individual features. The ability of a model to generalize to Cartesian products of the individual features is an important form of compositionality, which checks the model's capability to correctly predict in novel circumstances described as combination of contexts seen before. (Dong & Ma, 2022) developed results for additdynamicallyive models, i.e., labeling function is additive over individual features. While in (Wiedemer et al., 2023), the authors considered a more general model class in comparison to Dong & Ma (2022). The labeling function/decoder in (Wiedemer et al., 2023) takes the form $f(x_1, \cdots, x_n) = C(\psi_1(x_1), \cdots, \psi_n(x_n))$. However, they require a strong assumption, where the learner needs to know the function $C$ that is used to generate the data. (Lachapelle et al., 2024; Brady et al., 2023) extended the results from (Dong & Ma, 2022) to the unsupervised setting. (Lachapelle et al., 2024; Brady et al., 2023) are inspired by the success of object-centric models and show additive decoders enable generative models (autoencoders) to achieve Cartesian product extrapolation. While these works take promising and insightful first steps for provable compositional guarantees, the assumption of additive deterministic decoders (labeling functions) may come as quite restrictive. In particular a given attribute combination can then only correspond to a *unique* observation, produced by a very limited interaction between generative factors, not to a rich distribution of observations. By contrast an additive energy model can associate an almost arbitrary distribution over observations to a given set of attributes. Hence, we take inspiration independent mechanisms principle (Janzing & Schölkopf, 2010; Parascandolo et al., 2018) for our setting based on additive energy models. In the spirit of this principle, we think of each factor impacting the final distribution through an independent function, where independence is in the algorithmic sense and not the statistical sense. Based on this more realistic assumption of additive energy, our goal is to develop an approach that provably enables *zero-shot compositional generalization in discriminative tasks*, where the model needs to robustly predict never seen before factor combinations that the input is composed of. These additive energy distributions have also been used in generative compositionality (Liu et al., 2022a) but not in discriminative compositionality.

Finally, in another line of work (Schug et al., 2023), the authors consider compositionality in the task space and develop an approach that achieves provable compositional guarantees over this task space and empirically outperforms meta-learning approaches such as MAML and ANIL. Specifically, they operate in a student-teacher framework, where each task has a latent code that specifies the weights for different modules that are active for that task.

**Domain Generalization.** Generalization under subpopulation shifts, where certain groups or combinations of attributes are underrepresented in the training data, is a well-known challenge in machine learning. Group Distributionally Robust Optimization (GroupDRO) (Sagawa et al., 2019) is a prominent method that minimizes the worst-case group loss to improve robustness across groups. Invariant Risk Minimization (IRM) (Arjovsky et al., 2019) encourages the model to learn invariant representations that perform well across multiple environments. Perhaps the simplest methods are SUBG and RWG (Idrissi et al., 2022), which focus on constructing a balanced subset or reweighting examples to minimize or eliminate spurious correlations. There are many other interesting approaches that were proposed, see the survey for details (Zhou et al., 2022). The theoretical guarantees developed for these approaches (Rosenfeld et al., 2020; Arjovsky et al., 2019; Ahuja et al., 2020)

require a large diversity in terms of the environments seen at the training time. In our setting, we incorporate inductive biases based on additive energy distributions that help us arrive at provable generalization with limited diversity in the environments.

Closely related to our proposed method are the logit adjustment methods (Kang et al., 2019; Menon et al., 2020; Ren et al., 2020; Bayat et al., 2025) used in robust classification. (Kang et al., 2019) introduced Label-Distribution-Aware Margin (LDAM) loss for long-tail learning, proposing a method that adjusts the logits of a classifier based on the class frequencies in the training set to counteract bias towards majority classes. Similarly, (Menon et al., 2020) and (Ren et al., 2020) (Balanced Softmax), modify the standard softmax cross-entropy loss to account for class imbalance by shifting the logits according to the prior distribution over the classes. (Bayat et al., 2025) shifts the logits according to the held-out prediction of examples to prevent memorization towards robust generalization. Closest to our work are the Logit Correction (LC) (Liu et al., 2022b) and Supervised Logit Adjustment (sLA) (Tsirigotis et al., 2024) methods that use logit adjustment for group robustness. LC adjusts logits based on the joint distribution of environment and class label, reducing reliance on spurious features in imbalanced training sets. When environment annotations are unknown, a second network infers them. Supervised Logit Adjustment (sLA) adjusts logits according to the conditional distribution of classes given the environment. In the absence of environment annotations, Unsupervised Logit Adjustment (uLA) uses self-supervised learning (SSL) to pre-train a model for general feature representations, then derives a biased network from this pre-trained model to infer the missing environment annotations.

## B. Additive Energy Distributions versus Additive Decoder Models: A Closer Look

We will explore examples demonstrating the flexibility of additive energy distributions. Consider a set of images, each containing a distinct object that varies in shape, size, and color. We can associate an energy term with each attribute: one detecting shape, another detecting color, and a third detecting size. Together, these energy terms define the distribution of images conditioned on an object's shape, color, and size.

Now, contrast this with an additive decoder-based model. At any given pixel, it is unlikely that shape, color, and size attributes interact additively—rather, their interactions are complex and non-trivial. This example illustrates why additive energy distributions naturally model the conjunction of object characteristics, whereas additive decoders struggle with this task.

Another example from a different data modality is the CivilComments benchmark (Borkan et al., 2019), where the attributes toxic language (class label) and demographic identity (spurious attribute) interact non-trivially in text space. However, under additive energy distributions, we can model their interactions via an energy component that checks whether the language is toxic, and another energy component checks the demographic identity.

We now provide a simple example that can be modeled both by an additive decoder and additive energy distribution. Consider a two dimensional $x$, where each dimension of $x$ is controlled by a different attribute.

$$x = (x_1, x_2) = (g(z_1), h(z_2)) \tag{15}$$

Observe that we can express the above in terms of an additive decoder $x = (g(z_1), 0) + (0, h(z_2))$. We can also express the above in terms of an additive energy distribution as follows using Dirac delta functions $\delta$ as follows.

$p(x|z) = \delta(x - (g(z_1), h(z_2))) = \delta(x_1 - g(z_1))\delta(x_2 - h(z_2)) = \exp\left(\log(\delta(x_1 - g(z_1))) + \log(\delta(x_2 - g(z_2)))\right).$

The above example is meant to illustrate the point that if there are different subsets of the pixels, where each subset is controlled or impacted by a different set of attributes, then they can be modeled via an additive energy distribution or an additive decoder equivalently.

# C. CRM Implementation Details

## C.1. Algorithm for Two Attribute Case

---

**Algorithm 2** Compositional Risk Minimization (CRM) for 2 Attribute Case

---

**Input:** training set $\mathcal{D}^{\text{train}}$ with examples $(x, y, a)$, where $y$ is the class to predict and $a$ is an attribute spuriously correlated with $y$

**Output:** classifier parameters $\theta, W, B^\star$.

- Let $L, B \in \mathbb{R}^{d_y \times d_a}$ be the log prior and the bias terms.
- Define logits: $F_{L,B}(x) := -((W \cdot \phi(x;\theta))_{1:d_y} + (W \cdot \phi(x;\theta))_{d_y+1:d_y+d_a}^\top) + L - B$
- Define log probabilities: $\log p(y, a|x; \theta, W, L, B) := (F_{L,B}(x) - \text{logsumexp}(F_{L,B}(x)))_{y,a}$

**Training:**

- Estimate log prior $L^{\text{train}}$ from $\mathcal{D}^{\text{train}}$; $\hspace{3cm} L^{\text{train}}_{y,a} \leftarrow -\infty$ if $(y, a)$ absent from $\mathcal{D}^{\text{train}}$.
- Optimize $\theta, W$, and $B$ to maximize the log-likelihood over $\mathcal{D}^{\text{train}}$:
  $\theta, W, B \leftarrow \arg\max_{\theta, W, B} \sum_{(x,y,a) \in \mathcal{D}^{\text{train}}} \log p(y, a|x; \theta, W, L^{\text{train}}, B)$
- Extrapolate bias: $B^\star \leftarrow \log \left( \frac{1}{n} \sum_{x \in \mathcal{D}^{\text{train}}} \exp(F_{0,0}(x) - \text{logsumexp}(F_{L^{\text{train}},B}(x))) \right)$

**Inference on test point $x$:**

- Compute group probabilities, using $B^\star$, and $L^{\text{unif}} = \log \frac{1}{d_y d_a}$ aiming for shift to uniform prior:
  $q(y, a|x) \leftarrow \exp(\log p(y, a|x; \theta, W, L^{\text{unif}}, B^\star))$
- Marginalize over $a$ to get class probabilities: $q(y|x) \leftarrow \sum_a q(y, a|x)$

---

## C.2. PyTorch Implementation for Two Attribute Case

```python
# Pseudo-code for Compositional Risk Minimization (CRM)
import torch
import torchvision
from torch import nn
from torch.nn import functional as F
import load_dataloaders
import compute_metric

class CRM(nn.Module):
    """ Architcture of CRM Layer
    Args:
        d_phi: Input feature dimension
        d_y: Total categories for class labels y
        d_a: Total cateogies for spurious attributes a
    """
    def __init__(self,
                 d_phi: int,
                 d_y: int,
                 d_a: int,
                ):
        super(CRM, self).__init__()
        self.d_phi= d_phi
        self.d_y= d_y
        self.d_a= d_a

        self.linear_layer= nn.Linear(d_phi, d_y+d_a)
        self.bias= nn.Parameter(torch.zeros( (self.d_y, self.d_a) )).requires_grad
()
```

```
29
30      def forward(self, x):
31          return self.linear_layer(x)
32
33  def CRM_logits(x, phi, crm_layer, log_prior):
34      """ Computes the logit as per the  additive energy classifier
35      Inputs:
36          x: Batch of input images, Expected Shape (batch size, 3, 224, 224)
37          phi: Representation network
38          crm_layer: CRM layer with linear layer and bias
39          log_prior: Expected Shape (d_y, d_a)
40
41      Returns:
42          Logits for all groups of shape (batch_size, d_y, d_a)
43      """
44      d_y= crm_layer.d_y
45      d_a= crm_layer.d_a
46
47      #Obtain features via the representation network
48      features = phi(x)
49      #E_x is a vector of size d_y+d_a
50      E_x = crm_layer.linear_layer(features)
51      #Energy components for y, vector of size d_y
52      E_1 = E_x[0:d_y]
53      #Energy components for a, vector of size d_a
54      E_2 = E_x[d_y:d_y+d_a]
55      # Energy for all groups
56      logits = -(E_1.unsqueeze(1) + E_2.unsqueeze(0))
57      # Integrate log prior and bias
58      logits += log_prior - crm_layer.bias
59      return logits
60
61  def estimate_priors(dataset):
62      """Return counts of different groups in the dataset"""
63      d_y= dataset.num_y
64      d_a= dataset.num_a
65      counts = torch.zeros((d_y, d_a))
66      for x, y, a in dataset:
67          counts[y,a] += 1
68      priors = counts / len(dataset)
69      return priors
70
71  def CRM_extrapolate_bias(trainset, phi, crm_layer, log_prior):
72      """Compute extrapolated bias (B^{\star}) as per equation (11)
73      Inputs:
74          trainset: Train Dataloader
75          phi: Representation Network
76          crm_layer: CRM layer with linear layer and bias
77          log_prior: Expected Shape (d_y, d_a)
78
79      Returns:
80          Updated CRM layer with extrapolated bias of shape (d_y, d_a)
81      """
82      d_y= trainset.num_y
83      d_a= trainset.num_a
84
85      #Compute logits on all samples in trainset
86      logits=[]
87      for x, y, a in trainset:
88          logits.append( CRM_logits(x, phi, crm_layer, log_prior) )
89      logits= torch.cat(logits, dim=0)
90
91      #Compute extrapolated bias (B^{\star})
92      energy_tr= logits - log_prior + crm_layer.bias
93      log_probs= torch.sum( torch.exp(logits).view(-1), dim=1 )
```

```
94     extrapolated_bias= torch.log( torch.mean( torch.exp(-energy_tr) / log_probs ,
       dim=0) )
95
96     crm_layer.extrapolaed_bias= extrapolated_bias
97     return crm_layer
98
99  def CRM_test(testset, phi, crm_layer):
100     """Module for evaluating CRM
101     Inputs:
102         testset: Test Dataloader
103         phi: Representation Network
104         crm_layer: CRM layer with linear layer and bias
105
106     Returns:
107         List of values corresponing to evaluation of metric on each test batch
108     """
109
110     d_y= testset.num_y
111     d_a= testset.num_a
112
113     #Set test prior to be uniform
114     log_prior=  torch.log(1./(d_y+d_a)) * torch.ones()
115
116     final_res=[]
117     for x, y, a in testset:
118         #Forward Pass
119         logit = CRM_logits(x, phi, crm_layer, log_prior)
120         #Marginalize over attribute a to get class label predictions
121         y_pred= logit.sum(2)
122         #Compute metric of choices
123         res = compute_metric(y_pred, y)
124         final_res.append(res)
125
126     return final_res
127
128 def CRM_train(trainset, phi, crm_layer):
129     """Module for training CRM
130     Inputs:
131         trainset: Train Dataloader
132         phi: Representation Network
133         crm_layer: CRM layer with linear layer and bias
134
135     Returns:
136         Learned models phi and crmlayer
137     """
138
139     #Initalize Optimizer
140     opt= torch.optim.SGD(
141             [phi.parameters()+crm_layer.parameters()],
142             lr=0.001,
143             )
144
145     priors = estimate_priors(trainset)
146     #Since some group counts can be 0, so we clamp them to a tiny value to avoud -
       inf with log
147     log_prior = priors.clamp(min=1e-10).log()
148
149     #CRM Step 1 training loop
150     for epoch in range(1000):
151         for x, y, a in trainset:
152             #Forward Pass
153             opt.zero_grad()
154             logit = CRM_logits(x, phi, crm_layer, log_prior)
155             #Compute group label
156             g=trainset.d_a * y + a
```

```
157              #Compute cross entropy loss
158              loss = F.cross_entropy(logit.view(-1), g.long(), reduction='mean')
159              #Optimize parameters
160              loss.backward()
161              opt.step()
162
163      return phi, crm_layer, log_prior
164
165  def example_main():
166
167      #Load training and test dataloaders
168      trainset, testset = load_dataloaders()
169      #Attributes z = (y,a)
170      d_y = trainset.num_y
171      d_a = trainset.num_a
172
173      #Pretrained representation network
174      phi= torchvision.models.resnet.resnet50(pretrained=True)
175      #Feature Dimension
176      d_phi = phi.feat_dim
177      #CRM Layer
178      crm_layer= CRM(d_phi, d_y, d_a)
179
180      #Train
181      phi, crm_layer, log_prior= CRM_train(trainset, phi, crm_layer)
182
183      #This is where the magic is happening: extrapolating to biases of unseen
         combinations
184      crm_layer = CRM_extrapolate_bias(trainset, phi, crm_layer, log_prior)
185
186      #Test
187      CRM_test(testset, phi, crm_layer)
```

# D. Proofs

**Remark on proofs** We want to emphasize that the proofs developed here are quite different from related works on compositionality (Dong & Ma, 2022; Wiedemer et al., 2023). The foundation of proofs is built on a new mathematical object, discrete affine hull. The proof of Theorem 2 cleverly exploits properties of softmax and discrete affine hulls to show how we can learn the correct distribution without involving the intractable partition function in learning. The proof of Theorem 3, uses fundamental ideas from randomized algorithms to arrive at the probabilistic extrapolation guarantees.

We start with a basic lemma.

**Lemma 1.** *If* $z' \in \mathsf{DAff}(\mathcal{Z}^{\mathsf{train}})$, *i.e.,* $\sigma(z') = \sum_{z \in \mathcal{Z}^{\mathsf{train}}} \alpha_z \sigma(z)$, *where* $\langle 1, \alpha_z \rangle = 1$, *then* $\langle \sigma(z'), E(x) \rangle = \sum_{z \in \mathcal{Z}^{\mathsf{train}}} \alpha_z \langle \sigma(z), E(x) \rangle$.

*Proof.* $\langle \sigma(z'), E(x) \rangle = \langle \sum_{z \in \mathcal{Z}^{\mathsf{train}}} \alpha_z \sigma(z), E(x) \rangle = \sum_{z \in \mathcal{Z}^{\mathsf{train}}} \alpha_z \langle \sigma(z), E(x) \rangle$.

$\square$

## D.1. Proof for Theorem 1: Extrapolation of Conditional Density

**Theorem 1.** *If the true and learned distribution* ($p(\cdot|z)$ *and* $\hat{p}(\cdot|z)$) *are AED, then* $\hat{p}(\cdot|z) = p(\cdot|z), \forall z \in \mathcal{Z}^{\mathsf{train}} \implies \hat{p}(\cdot|z') = p(\cdot|z'), \forall z' \in \mathsf{DAff}(\mathcal{Z}^{\mathsf{train}})$.

*Proof.* We start by expanding the expressions for true and estimated log densities below

$$
\begin{aligned}
- \log \left[ p(x|z) \right] &= \langle \sigma(z), \mathbb{E}(x) \rangle + \log(\mathbb{Z}(z)), \\
- \log \left[ \hat{p}(x|z) \right] &= \langle \sigma(z), \hat{E}(x) \rangle + \log(\hat{\mathbb{Z}}(z)).
\end{aligned}
\tag{16}
$$

We equate these densities for the training attributes $z \in \mathcal{Z}^{\mathsf{train}}$. For a fixed $z \in \mathcal{Z}^{\mathsf{train}}$, we obtain that for all $x \in \mathbb{R}^n$

$$
\langle \sigma(z), \hat{E}(x) \rangle = \langle \sigma(z), E(x) \rangle + C(z),
\tag{17}
$$

where $C(z) = \log\left(\mathbb{Z}(z)/\hat{\mathbb{Z}}(z)\right)$. Since $z' \in \mathsf{DAff}(\mathcal{Z}^{\mathsf{train}})$, we can write $z' = \sum_{z \in \mathcal{Z}^{\mathsf{train}}} \alpha_z z$, $\langle 1, \alpha_z \rangle = 1$. From Lemma 1, we know that $\langle \sigma(z'), E(x) \rangle = \sum_{z \in \mathcal{Z}^{\mathsf{train}}} \alpha_z \langle \sigma(z), \hat{E}(x) \rangle$.

We use this decomposition and equation 17 to arrive at the key identity below. For all $x \in \mathbb{R}^n$

$$
\begin{aligned}
\langle \sigma(z'), \hat{E}(x) \rangle &= \sum_{z \in \mathcal{Z}^{\mathsf{train}}} \alpha_z \langle \sigma(z), \hat{E}(x) \rangle \\
&= \sum_{z \in \mathcal{Z}^{\mathsf{train}}} \alpha_z (\langle \sigma(z), E(x) \rangle + C(z)) \\
&= \left( \sum_{z \in \mathcal{Z}^{\mathsf{train}}} \alpha_z (\langle \sigma(z), E(x) \rangle \right) + \left( \sum_{z \in \mathcal{Z}^{\mathsf{train}}} \alpha_z C(z) \right) \\
&= \langle \sigma(z'), E(x) \rangle + \sum_{z \in \mathcal{Z}^{\mathsf{train}}} \alpha_z C(z)
\end{aligned}
\tag{18}
$$

From this we can infer that

$$
\begin{aligned}
\hat{p}(x|z') &= \frac{1}{\hat{\mathbb{Z}}(z')} \exp\left( -\langle \sigma(z'), \hat{E}(x) \rangle \right) \\
&= \frac{1}{\hat{\mathbb{Z}}(z')} \exp\left( -\langle \sigma(z'), E(x) \rangle - \sum_{z \in \mathcal{Z}^{\mathsf{train}}} \alpha_z C(z) \right)
\end{aligned}
\tag{19}
$$

We now use the fact that density integrates to one for continuous random variables (or alternatively the probability sums to one for discrete random variables). Thus

$$\int \hat{p}(x|z')dx = 1$$

$$\int \frac{1}{\hat{\mathbb{Z}}(z')} \exp\Big( -\langle \sigma(z'), E(x)\rangle - \sum_{z\in\mathcal{Z}^{\text{train}}} \alpha_z C(z)\Big)dx = 1$$

$$\frac{1}{\hat{\mathbb{Z}}(z')} \exp\Big( -\sum_{z\in\mathcal{Z}^{\text{train}}} \alpha_z C(z)\Big) \int \exp\Big( -\langle \sigma(z'), E(x)\rangle\Big)dx = 1$$

$$\frac{1}{\hat{\mathbb{Z}}(z')} \exp\Big( -\sum_{z\in\mathcal{Z}^{\text{train}}} \alpha_z C(z)\Big)\mathbb{Z}(z') = 1$$

$$\frac{1}{\hat{\mathbb{Z}}(z')} \exp\Big( -\sum_{z\in\mathcal{Z}^{\text{train}}} \alpha_z C(z)\Big) = \frac{1}{\mathbb{Z}(z')} \tag{20}$$

We substitute equation 20 into equation 19 to obtain

$$\hat{p}(x|z') = \frac{1}{\mathbb{Z}(z')} \exp\Big( -\langle \sigma(z'), E(x)\rangle\Big) = p(x|z'), \forall x \in \mathbb{R}^n \tag{21}$$

$\square$

### D.2. Proof for Theorem 2: Extrapolation of CRM

**Theorem 2.** *Consider the setting where $p(.|z)$ follows AED $\forall z \in \mathcal{Z}^\times$, the test distribution $q$ satisfies compositional shift characterization and $\mathcal{Z}^{\text{test}} \subseteq \text{DAff}(\mathcal{Z}^{\text{train}})$. If $\hat{p}(z|x) = p(z|x), \forall z \in \mathcal{Z}^{\text{train}}, \forall x \in \mathbb{R}^n$ and $\hat{q}(z) = q(z), \forall z \in \mathcal{Z}^{\text{test}}$, then the output of CRM (equation 10) matches the test distribution, i.e., $\hat{q}(z|x) = q(z|x), \forall z \in \mathcal{Z}^{\text{test}}, \forall x \in \mathbb{R}^n$.*

*Proof.* Since $q$ follows compositional shifts,

$$\log q(x|z) = \log p(x|z) = -\langle \sigma(z), E(x)\rangle - \log \mathbb{Z}(z) \tag{22}$$

We can write it as $-\langle \sigma(z), E(x)\rangle = \log p(x|z) + \log \mathbb{Z}(z)$.

Consider $z' \in \text{DAff}(\mathcal{Z}^{\text{train}})$. We can express $z'$ as $\sigma(z') = \sum_{z\in\mathcal{Z}^{\text{train}}} \alpha_z \sigma(z)$, where $\langle 1, \alpha_z\rangle = 1$.

We use equation 22 and show that the partition function at $z'$ can be expressed as affine combination of partition of the individual points and a correction term. We obtain the following condition. $\forall z' \in \mathcal{Z}^{\text{test}}$, where recall $\mathcal{Z}^{\text{test}} \subseteq \text{DAff}(\mathcal{Z}^{\text{train}})$,

$$
\begin{aligned}
\log\big(\mathbb{Z}(z')\big) &= \log \int \exp\big( -\langle \sigma(z'), E(x)\rangle\big)dx \\
&= \log \int \exp\Big( -\sum_{z\in\mathcal{Z}^{\text{train}}} \alpha_z \langle \sigma(z), E(x)\rangle\Big)dx \\
&= \log \int \exp\Big( \sum_{z\in\mathcal{Z}^{\text{train}}} \alpha_z \big( \log p(x|z) + \log \mathbb{Z}(z)\big)\Big)dx \\
&= \sum_{z\in\mathcal{Z}^{\text{train}}} \alpha_z \log \mathbb{Z}(z) + \log \int \exp\Big( \sum_{z\in\mathcal{Z}^{\text{train}}} \alpha_z \log p(x|z)\Big)dx
\end{aligned}
\tag{23}
$$

Denote the latter term in the above expression as

$$R(\{\alpha_z\}_{z\in\mathcal{Z}^{\text{train}}}) = \log \int \exp\Big( \sum_{z\in\mathcal{Z}^{\text{train}}} \alpha_z \log p(x|z)\Big)dx \tag{24}$$

We now simplify $\log\big(q(x|z')\big)$ using the property of partition function from equation 23 below. $\forall z' \in \mathcal{Z}^{\text{test}}$,

$$
\begin{aligned}
\log\big(q(x|z')\big) &= -\langle \sigma(z'), E(x) \rangle - \log \mathbb{Z}(z') \\
&= \sum_{z \in \mathcal{Z}^{\text{train}}} \alpha_z \Big( \log p(x|z) + \log \mathbb{Z}(z) \Big) - \log \mathbb{Z}(z') \\
&= \sum_{z \in \mathcal{Z}^{\text{train}}} \alpha_z \log p(x|z) + \sum_{z \in \mathcal{Z}^{\text{train}}} \alpha_z \log \mathbb{Z}(z) - \sum_{z \in \mathcal{Z}^{\text{train}}} \alpha_z \log \mathbb{Z}(z) - R\big(\{\alpha_z\}_{z \in \mathcal{Z}^{\text{train}}}\big) \\
&= \sum_{z \in \mathcal{Z}^{\text{train}}} \alpha_z \log p(x|z) - R\big(\{\alpha_z\}_{z \in \mathcal{Z}^{\text{train}}}\big)
\end{aligned}
\tag{25}
$$

We now simplify the first term in the above expression, i.e., $\sum_{z \in \mathcal{Z}^{\text{train}}} \alpha_z \log p(x|z)$, in terms of $p(z|x)$.

$$
\begin{aligned}
\sum_{z \in \mathcal{Z}^{\text{train}}} \alpha_z \big( \log \big( p(x|z) \big) &= \sum_{z \in \mathcal{Z}^{\text{train}}} \alpha_z \log \left( \frac{p(z|x)p(x)}{p(z)} \right) \\
&= \sum_{z \in \mathcal{Z}^{\text{train}}} \alpha_z \Big( \log p(z|x) - \log p(z) \Big) + \log p(x)
\end{aligned}
\tag{26}
$$

Similarly, $R(\{\alpha_z\}_{z \in \mathcal{Z}^{\text{train}}})$ can be phrased in terms of $p(z|x)$ as follows.

$$
\begin{aligned}
R\big(\{\alpha_z\}_{z \in \mathcal{Z}^{\text{train}}}\big) &= \log \int \exp \Big( \sum_{z \in \mathcal{Z}^{\text{train}}} \alpha_z \log p(x|z) \Big) dx \\
&= - \sum_{z \in \mathcal{Z}^{\text{train}}} \alpha_z \log p(z) + \log \Big( \mathbb{E}_{x \sim p(x)} \Big[ \exp \Big( \sum_{z \in \mathcal{Z}^{\text{train}}} \alpha_z \log p(z|x) \Big) \Big] \Big) \\
&= - \sum_{z \in \mathcal{Z}^{\text{train}}} \alpha_z \log p(z) + S\big(\{\alpha_z\}_{z \in \mathcal{Z}^{\text{train}}}\big),
\end{aligned}
\tag{27}
$$

where $S\big(\{\alpha_z\}_{z \in \mathcal{Z}^{\text{train}}}\big) = \log \Big( \mathbb{E}_{x \sim p(x)} \Big[ \exp \Big( \sum_{z \in \mathcal{Z}^{\text{train}}} \alpha_z \log p(z|x) \Big) \Big] \Big)$ and $\mathbb{E}_{x \sim p(x)}$ is the expectation w.r.t distribution $p(x)$. We use equation 26, equation 27 to simplify equation 25 as follows. $\forall z' \in \mathcal{Z}^{\text{test}}$,

$$
\begin{aligned}
\log q(x|z') &= \sum_{z \in \mathcal{Z}^{\text{train}}} \alpha_z \log p(z|x) - S\big(\{\alpha_z\}_{z \in \mathcal{Z}^{\text{train}}}\big) + \log p(x) \\
\log \left( \frac{q(z'|x)q(x)}{q(z')} \right) &= \sum_{z \in \mathcal{Z}^{\text{train}}} \alpha_z \log p(z|x) - S\big(\{\alpha_z\}_{z \in \mathcal{Z}^{\text{train}}}\big) + \log p(x) \\
\log \big( q(z'|x) \big) &= \sum_{z \in \mathcal{Z}^{\text{train}}} \alpha_z \Big( q(z') + \log p(z|x) \Big) - S\big(\{\alpha_z\}_{z \in \mathcal{Z}^{\text{train}}}\big) + \log \left( \frac{p(x)}{q(x)} \right)
\end{aligned}
\tag{28}
$$

We use translation invariance of softmax to obtain

$$
\begin{aligned}
q(z'|x) &= \mathsf{Softmax}\Big( \log q(z') + \sum_{z \in \mathcal{Z}^{\text{train}}} \alpha_z \log p(z|x) - S\big(\{\alpha_z\}_{z \in \mathcal{Z}^{\text{train}}}\big) \Big) \\
q(z'|x) &= \mathsf{Softmax}\Big( \log q(z') + \sum_{z \in \mathcal{Z}^{\text{train}}} \alpha_z \log p(z|x) - \log \Big( \mathbb{E}_{x \sim p(x)} \Big[ \exp \Big( \sum_{z \in \mathcal{Z}^{\text{train}}} \alpha_z \log p(z|x) \Big) \Big] \Big) \Big)
\end{aligned}
\tag{29}
$$

To avoid cumbersome notation, we took the liberty to show only one input to softmax, other inputs bear the same parametrization, they are computed at other $z$'s. From the above equation it is clear that if the learner knows the marginal distribution over the groups at test time, i.e., $q(z)$ and estimates $p(z|x)$ for all $z$'s in the training distribution's support, i.e., $\mathcal{Z}^{\text{train}}$, then the learner can successfully extrapolate to $q(z'|x)$.

Let us now use the *additive energy classifier* of the form we defined in equation 7 and whose energy $\hat{E}$ and bias $\hat{B}$ we optimized (equation 9) to match $p(z|x)$, so that:

$$p(z|x) = \frac{\exp\Big(-\langle\sigma(z),\hat{E}(x)\rangle + \log\hat{p}(z) - \hat{B}(z)\Big)}{\sum_{\tilde{z}\in\mathcal{Z}^{\text{train}}}\exp\Big(-\langle\sigma(\tilde{z}),\hat{E}(x)\rangle + \log\hat{p}(\tilde{z}) - \hat{B}(\tilde{z})\Big)},$$

Consequently

$$\sum_{z\in\mathcal{Z}^{\text{train}}}\alpha_z \log p(z|x)$$

$$= \Big(\sum_{z\in\mathcal{Z}^{\text{train}}}\alpha_z\Big(-\langle\sigma(z),\hat{E}(x)\rangle + \log p(z) - \hat{B}(z)\Big)\Big) - \log\Big(\sum_{\tilde{z}\in\mathcal{Z}^{\text{train}}}\exp\Big(-\langle\sigma(\tilde{z}),\hat{E}(x)\rangle + \log p(\tilde{z}) - \hat{B}(\tilde{z})\Big)\Big)$$

(30)

where we used the property that $\langle 1, \alpha_z\rangle = 1$.

Let us use this to simplify the last term of equation 29:

$$\log\Big(\mathbb{E}_{x\sim p(x)}\Big[\exp\Big(\sum_{z\in\mathcal{Z}^{\text{train}}}\alpha_z\log p(z|x)\Big)\Big]\Big)$$

$$= \log\Big(\mathbb{E}_{x\sim p(x)}\Big[\frac{\exp\Big(\sum_{z\in\mathcal{Z}^{\text{train}}}\alpha_z\Big(-\langle\sigma(z),\hat{E}(x)\rangle + \log p(z) - \hat{B}(z)\Big)\Big)}{\Big(\sum_{\tilde{z}\in\mathcal{Z}^{\text{train}}}\exp\Big(-\langle\sigma(\tilde{z}),\hat{E}(x)\rangle + \log p(\tilde{z}) - \hat{B}(\tilde{z})\Big)\Big)}\Big]\Big)$$

$$= \log\Big(\mathbb{E}_{x\sim p(x)}\Big[\frac{\exp\Big(\sum_{z\in\mathcal{Z}^{\text{train}}}\alpha_z\Big(-\langle\sigma(z),\hat{E}(x)\rangle\Big)\Big)}{\Big(\sum_{\tilde{z}\in\mathcal{Z}^{\text{train}}}\exp\Big(-\langle\sigma(\tilde{z}),\hat{E}(x)\rangle + \log p(\tilde{z}) - \hat{B}(\tilde{z})\Big)\Big)}\Big]\exp\Big(\sum_{z\in\mathcal{Z}^{\text{train}}}\alpha_z\Big(\log p(z) - \hat{B}(z)\Big)\Big)\Big)$$

(31)

$$= \log\Big(\mathbb{E}_{x\sim p(x)}\Big[\frac{\exp\Big(\sum_{z\in\mathcal{Z}^{\text{train}}}\alpha_z\Big(-\langle\sigma(z),\hat{E}(x)\rangle\Big)\Big)}{\Big(\sum_{\tilde{z}\in\mathcal{Z}^{\text{train}}}\exp\Big(-\langle\sigma(\tilde{z}),\hat{E}(x)\rangle + \log p(\tilde{z}) - \hat{B}(\tilde{z})\Big)\Big)}\Big]\Big) + \sum_{z\in\mathcal{Z}^{\text{train}}}\alpha_z\Big(\log p(z) - \hat{B}(z)\Big)$$

$$= \log\Big(\mathbb{E}_{x\sim p(x)}\Big[\frac{\exp\Big(-\langle\sigma(z'),\hat{E}(x)\rangle\Big)}{\Big(\sum_{\tilde{z}\in\mathcal{Z}^{\text{train}}}\exp\Big(-\langle\sigma(\tilde{z}),\hat{E}(x)\rangle + \log p(\tilde{z}) - \hat{B}(\tilde{z})\Big)\Big)}\Big]\Big) + \sum_{z\in\mathcal{Z}^{\text{train}}}\alpha_z\Big(\log p(z) - \hat{B}(z)\Big)$$

$$= B^{\star}(z') + \sum_{z\in\mathcal{Z}^{\text{train}}}\alpha_z\Big(\log p(z) - \hat{B}(z)\Big)$$

where we used Lemma 1, and $B^{\star}$ is as defined in equation 11.

Let us also define $c(x) = \log\Big(\sum_{\tilde{z}\in\mathcal{Z}^{\text{train}}}\exp\Big(-\langle\sigma(\tilde{z}),\hat{E}(x)\rangle + \log p(\tilde{z}) - \hat{B}(\tilde{z})\Big)\Big)$ so that we can reexpress equation 30 as:

$$\sum_{z\in\mathcal{Z}^{\text{train}}}\alpha_z\log p(z|x) = \Big(\sum_{z\in\mathcal{Z}^{\text{train}}}\alpha_z\Big(-\langle\sigma(z),\hat{E}(x)\rangle + \log p(z) - \hat{B}(z)\Big)\Big) - c(x)$$

(32)

Subtracting equation 31 from equation 32 we get:

$$\sum_{z \in \mathcal{Z}^{\text{train}}} \alpha_z \log p(z|x) - \log \left( \mathbb{E}_{x \sim p(x)} \Big[ \exp \Big( \sum_{z \in \mathcal{Z}^{\text{train}}} \alpha_z \log p(z|x) \Big) \Big] \right)$$

$$= \sum_{z \in \mathcal{Z}^{\text{train}}} \alpha_z \Big( - \langle \sigma(z), \hat{E}(x) \rangle + \log p(z) - \hat{B}(z) \Big) - c(x)$$

$$- B^{\star}(z') - \sum_{z \in \mathcal{Z}^{\text{train}}} \alpha_z \Big( \log p(z) - \hat{B}(z) \Big)$$

$$= \sum_{z \in \mathcal{Z}^{\text{train}}} \alpha_z \Big( - \langle \sigma(z), \hat{E}(x) \rangle \Big) - c(x) - B^{\star}(z')$$

$$= - \langle \sigma(z'), \hat{E}(x) \rangle - c(x) - B^{\star}(z') \tag{33}$$

Substituting this inside equation 29 yields

$$q(z'|x) = \mathsf{Softmax}\Big( \log q(z') - \langle \sigma(z'), \hat{E}(x) \rangle - c(x) - B^{\star}(z') \Big)$$
$$= \mathsf{Softmax}\Big( - \langle \sigma(z'), \hat{E}(x) \rangle + \log q(z') - B^{\star}(z') \Big) \tag{34}$$

where we removed the $c(x)$ term as softmax is invariant to addition of terms that do not depend on $z'$.

If $\hat{q}(z') = q(z'), \forall z' \in \mathcal{Z}^{\text{test}}$, then the expression in RHS corresponds to $\hat{q}(z'|x)$, as we had defined it in equation 10, before stating our theorem. Thus $q(z'|x) = \hat{q}(z'|x)$. This completes the proof.

$\square$

### D.3. Growth of Discrete Affine Hull for $2$ Attribute Case

Before presenting the proof of Theorem 3, we specifically deal with the case of $m = 2$ attributes and present a proof guided with visual intuitions. We believe this proof is easier to understand and helps build intuition for a reader to better grasp the proof in the next section for the general case. We first establish some basic lemmas. In the first lemma below, we consider a setting with two attributes, where each attribute takes two possible values, i.e., $m = 2$ and $d = 2$. In this setting there are four possible one-hot vectors $z^1, z^2, z^3, z^4$. We first show that each $z^i$ can be expressed as an affine combination of the remaining three.

**Lemma 2.** *If $m = 2, d = 2$, then there are four possible concatenated one-hot vectors $z$ denoted $z^1, z^2, z^3, z^4$. Each $z^i$ can be expressed as an affine combination of the remaining.*

*Proof.* Below we explicitly show how each $z^i$ can be expressed in terms of other $z^j$'s.

$$(+1) \cdot \begin{bmatrix} 0 \\ 1 \\ 0 \\ 1 \end{bmatrix} + (-1) \cdot \begin{bmatrix} 0 \\ 1 \\ 1 \\ 0 \end{bmatrix} + (+1) \cdot \begin{bmatrix} 1 \\ 0 \\ 1 \\ 0 \end{bmatrix} = \begin{bmatrix} 1 \\ 0 \\ 0 \\ 1 \end{bmatrix} \tag{35}$$

$$(-1) \cdot \begin{bmatrix} 0 \\ 1 \\ 0 \\ 1 \end{bmatrix} + (+1) \cdot \begin{bmatrix} 0 \\ 1 \\ 1 \\ 0 \end{bmatrix} + (+1) \cdot \begin{bmatrix} 1 \\ 0 \\ 0 \\ 1 \end{bmatrix} = \begin{bmatrix} 1 \\ 0 \\ 1 \\ 0 \end{bmatrix} \tag{36}$$

$$(+1) \cdot \begin{bmatrix} 1 \\ 0 \\ 1 \\ 0 \end{bmatrix} + (+1) \cdot \begin{bmatrix} 0 \\ 1 \\ 0 \\ 1 \end{bmatrix} + (-1) \cdot \begin{bmatrix} 1 \\ 0 \\ 0 \\ 1 \end{bmatrix} = \begin{bmatrix} 0 \\ 1 \\ 1 \\ 0 \end{bmatrix} \tag{37}$$

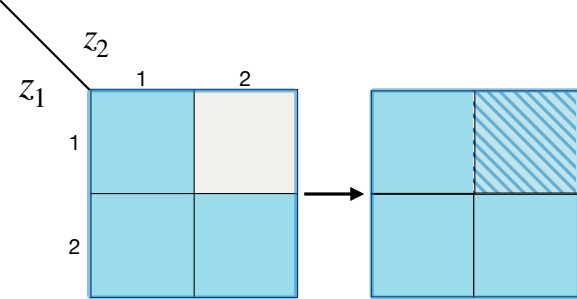

*Figure 3.* Setting of two attributes and two possible values per attribute. Illustration of extrapolation from three groups to the remaining fourth group. Three dark colored groups indicate the observed groups and the light colored shaded group indicates the group that is the affine combination of the three observed groups.

$$(-1) \cdot \begin{bmatrix} 1 \\ 0 \\ 1 \\ 0 \end{bmatrix} \quad + \quad (+1) \cdot \begin{bmatrix} 0 \\ 1 \\ 1 \\ 0 \end{bmatrix} \quad + \quad (+1) \cdot \begin{bmatrix} 1 \\ 0 \\ 0 \\ 1 \end{bmatrix} \quad = \quad \begin{bmatrix} 0 \\ 1 \\ 0 \\ 1 \end{bmatrix} \tag{38}$$

$\square$

We illustrate the setting of Lemma 2 in Figure 3. We now understand an implication of Lemma 2. Let us consider the setting where $m = 2$ and $d > 2$. Consider a subset of four groups $\{(i, j), (i', j), (i, j'), (i', j')\}$. Under one-hot concatenations these groups are denoted as $z^1 = [\underbrace{0, \cdots, 1_i, \cdots 0}_{\text{first attribute}}, \underbrace{0, \cdots, 1_j, \cdots 0}_{\text{second attribute}}]$, $z^2 = [0, \cdots, 1_{i'}, \cdots 0, 0, \cdots, 1_j, \cdots 0]$,

$z^3 = [0, \cdots, 1_i, \cdots 0, 0, \cdots, 1_{j'}, \cdots 0]$, and $z^4 = [0, \cdots, 1_{i'}, \cdots 0, 0, \cdots, 1_{j'}, \cdots 0]$. Observe that using Lemma 2, we get $z^4 = z^2 + z^3 - z^1$. Similarly, we can express every other $z^i$ in terms of rest of $z^j$'s in the the set $\{(i, j), (i', j), (i, j'), (i', j')\}$.

In the setting when $m = 2$ and $d \geq 2$, the total number of possible values $z$ takes is $d^2$. Each group recall is associated with attribute vector $z = [z_1, z_2]$, where $z_1 \in \{1, \cdots, d\}$ and $z_2 \in \{1, \cdots d\}$. The set of all possible values of $z$ be visualized as $d \times d$ grid in this notation. We call this $d \times d$ grid as $G$. We will first describe a specific approach of selecting observed groups $z$ for training, which shows that with just $2d - 1$ it is possible to affine span all the possible $d^2$ groups in the grid $G$. We leverage the insights from this approach and show that with a randomized approach of selecting groups, we can continue to affine span $d^2$ groups with $\mathcal{O}(d \log(d))$ groups.

Denote the set of observed groups as $N$. Suppose that their affine hull contains all the points in a subgrid $S \subseteq G$ of size $m \times n$. Let the subgrid $S = \{x_1, \cdots, x_m\} \times \{y_1, \cdots, y_n\}$. Without loss of generality, we can permute the points and make the subgrid contiguous as follows $S = \{1, \cdots, m\} \times \{1, \cdots, n\}$. Next, we add a new point $g = (g_x, g_y) \in G$ but $g \notin S$. We argue that if $g_x \in \{1, \cdots, m\}$, then the affine hull of $N \cup \{g\}$ contains a larger subgrid of size $m \times (n + 1)$. Similarly, we want to argue that if $g_y \in \{1, \cdots, n\}$, then the affine hull of $N \cup \{g\}$ contains a larger subgrid of size $(m + 1) \times n$. Define $C_x$ as the Cartesian product of $\{g_x\}$ with $\{1, \cdots, n\}$, i.e., $C_x = \{(g_x, 1), (g_x, 2), \cdots, (g_x, n)\}$. Define $C_y$ as the Cartesian product of $\{1, \cdots, m\}$ with $\{g_y\}$, i.e., $C_y = \{(1, g_y), (2, g_y), \cdots, (m, g_y)\}$.

**Theorem 4.** *Suppose the affine hull of the observed set $N$ contains a subgrid $S$ of size $m \times n$. If the new point $g = (g_x, g_y)$ shares the $x$-coordinate with a point in $S$, and $g \notin S$, then the the affine hull of $N \cup \{g\}$ contains $S \cup C_y$.*

*Proof.* We write the set of observed groups $N$ as $N = \{z^{\theta_j}\}_j$. The affine hull of $N$ contains $S = \{1, \cdots, m\} \times \{1, \cdots, n\}$. We observe a new group $g \notin S$, which shares its $x$ coordinate with one of the points in $S$. Without loss of generality let this point be $g = (1, n + 1)$ (if this were not the case, then we can always permute the columns and rows to achieve such a configuration). Consider the triplet $- (z^1, z^2, z^3) = ((1, n), (2, n), (1, n + 1))$. Observe that $z^1, z^2, z^3, z^4$ form a $2 \times 2$ subgrid, where $z^4 = (2, n + 1)$. We use Lemma 2 to infer that the fourth point $z^4 = (2, n + 1)$ on this $2 \times 2$ subgrid can be obtained as an affine combination of this triplet, i.e., $z^4 = \alpha z^1 + \beta z^2 + \gamma z^3$. Since $z^1, z^2$ are in the affine hull of $N$, they can be written as an affine combination of seen points in $N$ as follows $z^1 = \sum_{k \in N} a_k z^{\theta_k}$, $z^2 = \sum_{k \in N} b_k z^{\theta_k}$. As a result,

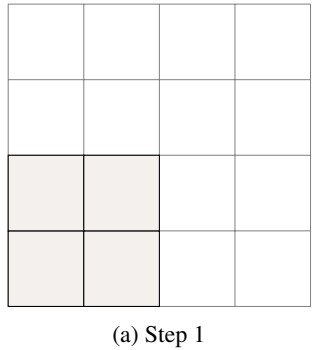 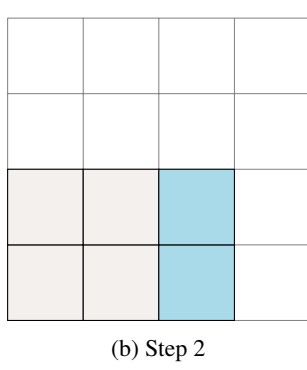 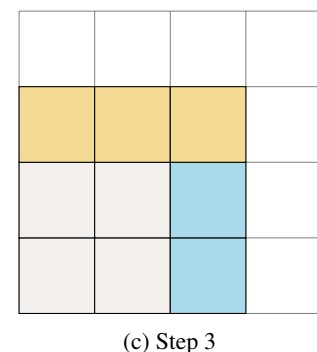

(a) Step 1          (b) Step 2          (c) Step 3

*Figure 4.* Illustration of steps of the deterministic sampling procedure for a $4 \times 4$ grid. (a) shows the base set, (b) add a group in blue and the affine hull extends to include all the blue cells, (c) add a group in yellow and the affine hull extends to include all yellow cells.

we obtain

$$
\begin{aligned}
z^4 = \alpha z^1 + \beta z^2 + \gamma z^3 &= \alpha \Big( \sum a_k z^{\theta_k} \Big) + \beta \Big( \sum b_k z^{\theta_k} \Big) + \gamma z^3 \\
&= \sum_{k \in N} \Big( \alpha a_k + \beta b_k \Big) z^{\theta_k} + \gamma z^3
\end{aligned}
\tag{39}
$$

Observe that $\sum_k (\alpha a_k + \beta b_k) = (\alpha \sum_k a_k + \beta \sum_k b_k) = \alpha + \beta$. Since $\alpha + \beta + \gamma = 1$, $z^4$ is an affine combination of points in $N \cup \{g\}$. Thus we have shown the claim for the point $(2, n+1)$. We can repeat this claim for point $(3, n+1)$ and so on until we reach $(m, n+1)$ beyond which there would be no points in $S$ that are expressed as affine combination of $N$. We can make this argument formal through induction. We have already shown the base case above. Suppose all the points $(j, n+1)$ in $j \leq i < m$ are in the affine hull of $N \cup \{g\}$. Consider the point $z^4 = (i+1, n+1)$. Construct the triplet $(z^1, z^2, z^3) = \big((i, n), (i, n+1), (i+1, n)\big)$. Again from Lemma 2, it follows that $z^4 = \alpha z^1 + \beta z^2 + \gamma z^3$. We substitute $z^1, z^2$ and $z^3$ with their corresponding affine combinations. $z^4 = \alpha \sum_{k \in N \cup \{g\}} a_k z^{\theta_k} + \beta \sum_{k \in N \cup \{g\}} b_k z^{\theta_k} + \gamma \sum_{k \in N \cup \{g\}} c_k z^{\theta_k}$. Since $\sum_{k \in N \cup \{g\}} \alpha a_k + \beta b_k + \gamma c_k = 1$, it follows that $z^4$ is an affine combination of $z^1, z^2$ and $z^3$. This completes the proof.

$\square$

We now describe a simple deterministic procedure that helps us understand how many groups we need to see before we are guaranteed that the affine hull of seen points span the whole grid $G = \{1, \cdots, d\} \times \{1, \cdots, d\}$.

- We start with a base set of three points – $B = \{(1,1), (1,2), (2,1)\}$. From Lemma 2, the affine hull contains $(2, 2)$.

- For each $i \in \{2, \cdots, d-1\}$ add the points $(1, i+1), (i+1, 1)$ to the set $B$. From Theorem 4, it follows that affine hull of $B \cup \{(1, i+1)\} \cup \{(i+1, 1)\}$ contains $(i+1 \times i+1)$ subgrid $\{1, \cdots, i+1\} \times \{1, \cdots, i+1\}$ (here we apply Theorem 4 in two steps once for the addition of $(1, i+1)$ and then for the addition of $(i+1, 1)$.

At the end of the above procedure $B$ contains $2d - 1$ points and its affine hull contains the grid $G$. We illustrate this procedure in Figure 4.

We now discuss a randomized procedure that also allows us to span the entire grid $G$ with $\mathcal{O}(d \log(d))$ groups. The idea of the procedure is to start with a base set of groups and construct their affine hull $S$. Then we wait to sample a group $g$ that is outside this affine hull. If this sampled group shares the $x$ coordinate with affine hull of $B$ denoted as $S$, then we expand the subgrid by one along with $y$ coordinate. Similarly, we also wait for a point that shares a $y$ coordinate and then we expand the subgrid by one along the $x$ coordinate.

We use $S_x$ to denote the distinct set of $x$-coordinates that appear in $S$ and same goes for $S_y$. We write $g = (g_x, g_y)$. The procedure goes as follows.

Set $S = \emptyset, B = \emptyset$ and $\mathsf{Flag} = x$.

- Sample a group $g$ from $G$ uniform at random. Update $B = B \cup \{g\}, S = S \cup \{g\}$.

- While $S \neq G$, sample a group $g$ from $G$ uniform at random.
  - If Flag $== x$, $g_x \in S_x$, $g \notin S$, then update $B = B \cup \{g\}$, $S = S \cup (S_x \times \{g_y\})$ and Flag $= y$.
  - If Flag $== y$, $g_y \in S_y$, $g \notin S$, then update $B = B \cup \{g\}$, $S = S \cup (\{g_x\} \times S_y)$ and Flag $= x$.

In the above procedure, in every step in the while loop a group $g$ is sampled. Whenever the Flag flips from $x$ to $y$, then following Theorem 4, the updated set $S$ belongs to the affine hull of $B$. We can say the same when Flag flips from $y$ to $x$. In the next theorem, we will show that the while loop terminates after $8cd \log(d)$ steps with a high probability and the affine hull of $B$ contains the entire grid $G$. We follow this strategy. We count the time it takes for Flag to flip from $x$ to $y$ (from $y$ to $x$) as it grows the size of $S$ from a $k \times k$ subgrid to $k \times (k+1)$ ($k \times (k+1)$ subgrid to $(k+1) \times (k+1)$) subgrid.

**Theorem 5.** *Suppose we sample the groups based on the randomized procedure described above. If the number of sampled groups is greater than $8cd \log(d)$, then $G \subseteq \mathsf{DAff}(B)$ with a probability greater than equal to $1 - \frac{1}{c}$.*

*Proof.* We take the first group $g$ that is sampled. Without loss of generality, we say this group is $(1, 1)$.

Suppose the Flag is set to $x$. Define an event $A_1^k$: newly sampled $g = (g_x, g_y)$ shares $x$-coordinate with a point in $S$ (size $k \times k$), $g \notin S$. Under these conditions Flag flips from $x$ to $y$. To compute the probability of this event let us count the number of scenarios in which this happens. If $g_x$ takes one of the $k$ values in $S_x$ and $g_y$ takes one of the remaining $(d - k)$, then the event happens. As a result, the probability of this event is $P(A_1^k) = \frac{(k)(d-k)}{d^2}$.

Suppose the Flag is set to $y$. Define an event $A_2^k$: newly sampled $g = (g_x, g_y)$ shares $y$-coordinate with a point in $S$ (size $k \times (k+1)$) and $g \notin S$. Under these conditions Flag flips from $y$ to $x$. The probability of this event is $P(A_2^k) = \frac{(k+1)(d-k)}{d^2}$.

Define $T_1^k$ as the number of groups that need to be sampled before $A_1^k$ occurs. Define $T_2^k$ as the number of groups that need to be sampled before $A_2^k$ occurs. Observe that after $T_1^k + T_2^k$ number of sampled groups the size of the current subgrid $S$, which is in the affine hull of $B$, grows to $(k+1) \times (k+1)$.

Define $T_{\mathsf{sum}} = \sum_{k=1}^{d-1}(T_1^k + T_2^k)$. $T_{\mathsf{sum}}$ is the total number of groups sampled before the affine span of the observed groups $B$ contains the grid $G$.

We compute

$$\mathbb{E}[T_{\mathsf{sum}}] = \sum_{k=1}^{d-1}(\mathbb{E}[T_1^k] + \mathbb{E}[T_2^k])$$

$$\sum_{k=1}^{d} \mathbb{E}[T_1^k] = \sum_{k=1}^{d-1} d^2/(k(d-k)) = 2 \sum_{k=1}^{(d-1)/2} d^2/(k(d-k)) \tag{40}$$

$$2 \sum_{k=1}^{(d-1)/2} d^2/(k(d-k)) = 2d \sum_{k=1}^{(d-1)/2} \left[\frac{1}{k} + \frac{1}{d-k}\right] \approx 4d \log((d-1)/2)$$

Similarly, we obtain a similar bound for $\sum_{k=1}^{d-1} \mathbb{E}[T_2^k]$.

$$\sum_{k=1}^{d}(\mathbb{E}[T_2^k] = \sum_{k=1}^{d-1} d^2/((k+1)(d-k)) = 2 \sum_{k=1}^{(d-1)/2} d^2/((k+1)(d-k))$$

$$2 \sum_{k=1}^{(d-1)/2} d^2/((k+1)(d-k)) \leq 2d \sum_{k=1}^{(d-1)/2} \left[\frac{1}{k+1} + \frac{1}{d-k}\right] \approx 4d \log((d-1)/2) \tag{41}$$

Overall $\mathbb{E}[T_{\mathsf{sum}}] \approx 8d \log(d/2)$. From Markov inequality, it immediately follows that $P(T_{\mathsf{sum}} \leq 8cd \log(d/2)) \geq 1 - \frac{1}{c}$. In the above approximations, we use $\sum_{i=1}^{d} \frac{1}{i} \approx \log d + \gamma$, where $\gamma$ is Euler's constant. We drop $\gamma$ as its a constant, which can always be absorbed by adapting the constant $c$.

$\square$

**Theorem 6.** *Consider the setting where $p(.|z)$ follows AED $\forall z \in \mathcal{Z}^{\times}$, $\mathcal{Z}^{\text{train}}$ comprises of $s$ attribute vectors $z$ drawn uniformly at random from $\mathcal{Z}^{\times}$, and the test distribution $q$ satisfies compositional shift characterization. If $s \geq 8cd \log d$, where $d$ is sufficiently large, $\hat{p}(z|x) = p(z|x), \forall z \in \mathcal{Z}^{\text{train}}, \forall x \in \mathbb{R}^n$, $\hat{q}(z) = q(z), \forall z \in \mathcal{Z}^{\text{test}}$, then the output of CRM (equation 10) matches the test distribution, i.e., $\hat{q}(z|x) = q(z|x)$, $\forall z \in \mathcal{Z}^{\text{test}}, \forall x \in \mathbb{R}^n$, with probability greater than $1 - \frac{1}{c}$.*

*Proof.* Suppose the support of training distribution $p(z)$ contains $s$ groups. We know that these $s$ groups are drawn uniformly at random. From Theorem 5, it is clear that if $s$ grows as $\mathcal{O}(d \log d)$, then with a high probability the entire grid of $d^2$ combinations is contained in the affine span of these observed groups. This can be equivalently stated as $\mathcal{Z}^{\times} \subseteq \mathsf{DAff}(\mathcal{Z}^{\text{train}})$ with a probability greater than equal to $1 - \frac{1}{c}$. If $\mathcal{Z}^{\times} \subseteq \mathsf{DAff}(\mathcal{Z}^{\text{train}})$, then from the assumption of compositional shifts, it follows that $\mathcal{Z}^{\text{test}} \subseteq \mathsf{DAff}(\mathcal{Z}^{\text{train}})$. We can now use Theorem 2 and arrive at our result. This completes the proof. $\square$

### D.4. Proof for Theorem 3: Growth of Discrete Affine Hull for $m$ Attribute Case

Let $G = \{1, \cdots, d\}^m$ and let us consider the groups $z = [z_1, \ldots, z_m] \in G$. $G$ contains $d^m$ points. We are given a set of groups $S \subset G$. $\mathsf{DAff}(S)$ is the discrete affine hull of $S$, where recall $\mathsf{DAff}(S) = \mathsf{Aff}(S) \cap G$ is the set of points in the affine hull of $S$ that lie on the grid $G$.

**Theorem 7.** *If the total number of groups sampled is greater than $2cd(m + 1 + \ln(d))$, then $\mathsf{DAff}(S) = G$ with probability at least $1 - \frac{1}{c}$.*

*Proof.* Let us write $s := 2cd(m + 1 + \ln(d))$. It is easy to see that $\mathsf{Aff}(G)$ is an affine vector space of dimension $m(d-1)$. Consider an infinite sequence $(z^l)_{l \in \mathbb{N}}$ of i.i.d. uniformly sampled groups in $G$, the increasing sequence of sets $S^l := \{z^1, \ldots, z^l\}$, and the corresponding increasing sequence of affine spaces $\mathsf{Aff}(S^l)$. The theorem is equivalent to stating that we have $\mathsf{Aff}(S^s) = \mathsf{Aff}(G)$ with probability at least $1 - \frac{1}{c}$.

For every $l \geq 1$, if the "newly sampled" point $z^l$ does not belong to $\mathsf{Aff}(S^{l-1})$, then necessarily $\dim \mathsf{Aff}(S^l) = \dim \mathsf{Aff}(S^{l-1}) + 1$. For $\mathsf{Aff}(S)$ to be equal to $\mathsf{Aff}(G)$, there needs to be $m(d-1) = \dim \mathsf{Aff}(G)$ such increases in dimensionality when going from $S^0 = \emptyset$ to $S^s$ (and there cannot be more increases than that). Note that as long as $\mathsf{DAff}(S^{l-1}) \neq G$, the probability that $z^l$ does not belong to $\mathsf{Aff}(S^{l-1})$ is at least $1/|G|$, hence with probability 1 we have $\mathsf{DAff}(S^l) = G$ for $l$ large enough.

For $i = 1, \ldots, m(d-1)$, define the random variable

$$T_i := |\{j \ : \ \dim \mathsf{Aff}(S^j) = i\}|$$

(as noted above, $T_i$ is well-defined and finite with probability 1) and the random index

$$t_i := \min\{j \ : \dim \mathsf{Aff}(S^j) = i\}.$$

The random variable $T_i$ counts the number of points $z^l$ sampled before a point not yet in $\mathsf{Aff}(S^{t_i})$ is sampled (thus leading to an increase in dimension). Define also

$$T_{\mathsf{sum}} := 1 + \sum_{i=1}^{m(d-1)-1} T_i.$$

Note that $T_{\mathsf{sum}} = \min\{j \ : \ \mathsf{Aff}(S^j) = \mathsf{Aff}(G)\}$. Hence we have to show that with high probability, the random variable $T_{\mathsf{sum}}$ is smaller than $s$.

The probability of a newly sampled point not belonging to $\mathsf{DAff}(S^{t_i})$ (and thus leading to an increase in dimension) is equal to $(|G| - |\mathsf{DAff}(S^{t_i})|)/|G|$, hence $T_i$ is a geometric variable of mean $|G|/(|G| - |\mathsf{DAff}(S^{t_i})|) = d^m/(d^m - |\mathsf{DAff}(S^{t_i})|)$, and

$$\mathbb{E}[T_{\mathsf{sum}}] = 1 + \sum_{i=1}^{m(d-1)-1} \mathbb{E}[T_i] = 1 + \sum_{i=1}^{m(d-1)-1} \frac{d^m}{d^m - |\mathsf{DAff}(S^{t_i})|}. \tag{42}$$

The following lemma, whose proof is given further below, is the key ingredient to bound this sum:

**Lemma 3.** *Let $A \subset G$ be such that $\mathsf{DAff}(A) = A$. If $\frac{|A|}{d^m} \geq \frac{1}{2}$, i.e. if $A$ contains more than half the points of $G$, then the following inequality holds:*

$$\dim \mathsf{Aff}(G) - \dim \mathsf{Aff}(A) \leq 2d \frac{d^m - |A|}{d^m}.$$

Intuitively, the lemma states that if the set $A$ contains most points in $G$ (whose cardinality is $d^m$), then the dimension of the affine space it spans is almost that of $\mathsf{Aff}(G)$.

Let $i^* := \min\{i \ : \ |\mathsf{DAff}(S^{t_i})|/d^m > 1/2\}$. Then Lemma 3 applies to $\mathsf{DAff}(S^t)$ for all $t \geq t_{i^*}$, and we see in particular that

$$\dim \mathsf{Aff}(G) - i^* = \dim \mathsf{Aff}(G) - \dim \mathsf{Aff}(S^{t_{i^*}}) \leq 2d \frac{d^m - |\mathsf{DAff}(S^{t_{i^*}})|}{d^m} < d,$$

hence $i^*$ must be greater than $\dim \mathsf{Aff}(G) - d = m(d-1) - d$. Thus we can split the sum as follows:

$$\mathbb{E}[T_{\mathsf{sum}}] = 1 + \sum_{i=1}^{m(d-1)-1} \mathbb{E}[T_i]$$

$$\leq 1 + \sum_{i=1}^{i^*-1} \frac{d^m}{d^m - |\mathsf{DAff}(S^{t_i})|} + \sum_{i=i^*}^{m(d-1)-1} \frac{d^m}{d^m - |\mathsf{DAff}(S^{t_i})|}$$

$$\leq 1 + \sum_{i=1}^{i^*-1} 2 + \sum_{i=i^*}^{m(d-1)-1} \frac{2d}{\dim \mathsf{Aff}(G) - \dim \mathsf{Aff}(S^{t_i})}$$

$$\leq 1 + 2(i^* - 1) + \sum_{i=m(d-1)-d}^{m(d-1)-1} \frac{2d}{m(d-1) - \dim \mathsf{Aff}(S^{t_i})}$$

$$\leq 2md + \sum_{j=1}^{d} \frac{2d}{j} \approx 2d(m + 1 + \ln(d)),$$

where we apply both Lemma 3 and the definition of $i^*$ to get the third line.

From Markov's inequality, we then see that $P(T_{\mathsf{sum}} \leq c\mathbb{E}[T_{\mathsf{sum}}]) \geq 1 - \frac{1}{c} \implies P(T_{\mathsf{sum}} \leq s = 2cd(m+1+\ln(d))) \geq 1 - \frac{1}{c}$.

$\square$

*Proof of Lemma 3 .* We prove the lemma by induction on $m \in \mathbb{N}$. If $m = 1$, then $\dim \mathsf{Aff}(A) = |A| - 1$ and the statement reduces to the almost trivial inequality

$$\dim \mathsf{Aff}(G) - \dim \mathsf{Aff}(A) = d - |A| \leq 2d \frac{d - |A|}{d}.$$

Let us now consider $m \in \mathbb{N}$ and assume that the lemma has been proved for $m - 1$. Let $c := (d^m - |A|)/d^m$ be the ratio of missing points, and let us define the subsets

$$G^k = \{z \in G \ : \ z_m = k\}$$

for $k \in [d]$. Consider the set of indices

$$I := \{k \in [d] \ : \ A \cap G^k \neq \emptyset\}.$$

As

$$A \subset \bigcup_{k \in I} G^k,$$

the average number of points of $A$ within each set $G^k$ (for $k \in I$) must be $|A|/|I|$, and in particular there exists $k' \in I$ such that $|A \cap G^{k'}| \geq |A|/|I|$. Up to reordering the elements $\{1, \ldots, d\}$, we can assume without loss of generality that $k' = 1$. Consider now the sets $G^1$ and $A^1 := A \cap G^1$. There is a trivial isomorphism between the one hot encoding representations

of $G^1$ and that of the set $[d]^{m-1}$, and the set $A^1$ does verify $\mathsf{Aff}(A^1) \cap G^1 = A^1$. Moreover, $\frac{|A^1|}{d^{m-1}} \geq \frac{|A|}{|I|d^{m-1}} = \frac{d}{|I|}\frac{|A|}{d^m} \geq \frac{1}{2}$ by assumption. Hence we can apply our recursive hypothesis to the sets $A^1$ and $G^1$ to conclude that

$$\dim \mathsf{Aff}(G^1) - \dim \mathsf{Aff}(A^1) \leq 2d \frac{d^{m-1} - |A^1|}{d^{m-1}} \leq 2d\frac{d^m - |A|d/|I|}{d^m}. \tag{43}$$

As $A^1 \subset A$, we can write $\{a_1, \ldots, a_l\} = A \backslash A^1$ for some $l$, and we see that

$$
\begin{aligned}
\dim \mathsf{Aff}(G^1) - \dim \mathsf{Aff}(A^1) &\geq \dim \mathsf{Aff}(G^1 \cup \{a_1\}) - \dim \mathsf{Aff}(A^1 \cup \{a_1\}) \\
&\geq \dim \mathsf{Aff}(G^1 \cup \{a_1, a_2\}) - \dim \mathsf{Aff}(A^1 \cup \{a_1, a_2\}) \\
&\geq \ldots \\
&\geq \dim \mathsf{Aff}(G^1 \cup \{a_1, \ldots, a_l\}) \\
&\quad - \dim \mathsf{Aff}(A^1 \cup \{a_1, \ldots, a_l\}) \\
&= \dim \mathsf{Aff}(G^1 \cup A) - \dim \mathsf{Aff}(A),
\end{aligned}
$$

where each successive inequality is true because the newly added point $a_{s+1}$ either belongs to both $\mathsf{Aff}(G^1 \cup \{a_1, \ldots, a_s\})$ and $\mathsf{Aff}(A^1 \cup \{a_1, \ldots, a_s\})$, to neither of them, or only to $\mathsf{Aff}(A^1 \cup \{a_1, \ldots, a_s\})$. Hence we find that

$$\dim \mathsf{Aff}(G^1 \cup A) - \dim \mathsf{Aff}(A) \leq \dim \mathsf{Aff}(G^1) - \dim \mathsf{Aff}(A^1). \tag{44}$$

Note also that $\bigcup_{k \in I} G^k \subset \mathsf{Aff}(G^1 \cup A)$ (in fact, the inclusion is an equality). Indeed, it is easy to see that for any $z \in G^k$, we have $G^k \subset \mathsf{Aff}(G^1 \cup \{z\})$: with our vectorization of $G$, the set $G^i$ is composed of all vectors of the shape $[u_1, \ldots, u_{m-1}, e_i^d]$, where the $u_j \in \mathbb{R}^d$ are one-hot encodings and $e_i^d = [0, \ldots, 1, \ldots, 0]$ is a one-hot encoding of $i \in [d]$, i.e. $d-1$ zeroes and a one in position $i$. Let us write $z \in G^k$ as $[v_1, \ldots, v_{m-1}, e_k^d]$, and consider $\tilde{z} := [v_1, \ldots, v_{m-1}, e_1^d] \in G^1$. Note that $z - \tilde{z} = [0, \ldots, 0, e_k^d - e_1^d]$. Then any $z' = [u_1, \ldots, u_{m-1}, e_k^d] \in G^k$ can be written as $[u_1, \ldots, u_{m-1}, e_1^d] + z - \tilde{z} \in \mathsf{Aff}(G^1 \cup \{z\})$, which shows that $G^k \subset \mathsf{Aff}(G^1 \cup \{z\})$. As $A \cap G^k \neq \emptyset$ for any $k \in I$ (by definition), this means that $G^k \subset \mathsf{Aff}(G^1 \cup A)$ for all $k \in I$. Hence we get

$$\dim \mathsf{Aff}\left(\bigcup_{k \in I} G^k\right) - \dim \mathsf{Aff}(A) \leq 2d\frac{d^m - |A|d/|I|}{d^m} \tag{45}$$

by combining equation 43 and equation 44. Furthermore, the same argument as above shows that if $z^j \in G^j$ for $j \in [d]\backslash I$, then $G^j \subset \mathsf{Aff}(G^1 \cup \{z^j\})$, which means that

$$G = \mathsf{Aff}\left(\left(\bigcup_{k \in I} G^k\right) \cup \{z^j\}_{j \in [d]\backslash I}\right).$$

This means that adding $d - |I|$ vectors to $\mathsf{Aff}\left(\bigcup_{k \in I} G^k\right)$ is enough for the resulting set to affinely generate $G$; this is equivalent to saying that

$$\dim \mathsf{Aff}(G) - \dim \mathsf{Aff}\left(\bigcup_{k \in I} G^k\right) \leq d - |I|. \tag{46}$$

By combining equation 45 and equation 46, we find that

$$
\begin{aligned}
\dim \mathsf{Aff}(G) - \dim \mathsf{Aff}(A) &\leq 2d\frac{d^m - |A|d/|I|}{d^m} + d - |I| \\
&= 2d\frac{d^m - |A|}{d^m} + 2d\frac{|A|}{d^m}(1 - d/|I|) + d - |I|.
\end{aligned}
$$

Now we only need to show that $2d\frac{|A|}{d^m}(1 - d/|I|) + d - |I| \leq 0$ to complete the recurrence and prove the lemma. Remember that we have assumed that $\frac{|A|}{d^m} \geq 1/2$. Note also that as $A = \bigsqcup_{k \in I}(G^k \cap A) \leq |I|d^{m-1}$, we have $|A|/d^m \leq |I|/d$. Then the desired inequality is equivalent (by setting $a := \frac{|A|}{d^m}$ and $c := |I|/d$) to showing that for any $a \in [1/2, 1]$ and any $c \in [a, 1]$, we have

$$2a(1 - 1/c) + 1 - c \leq 0,$$

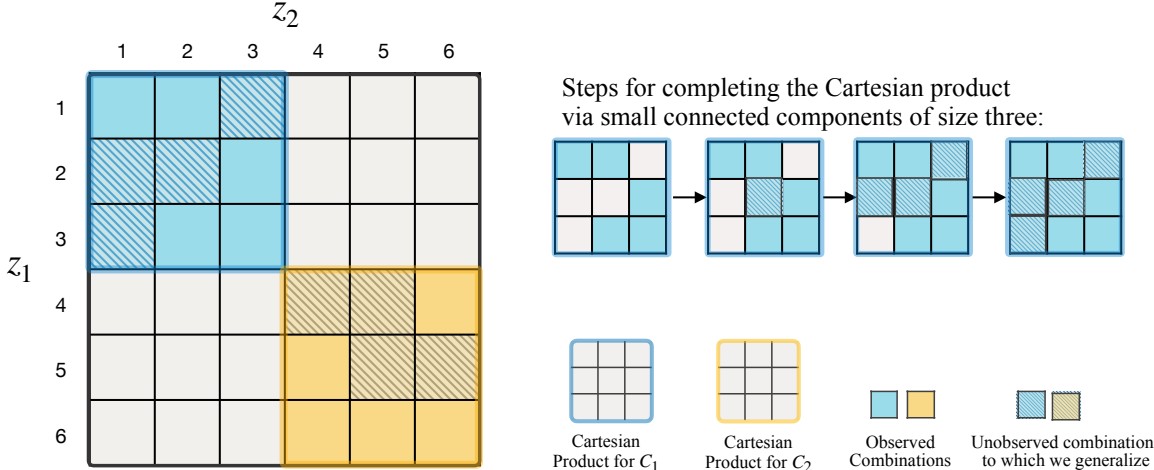

**Figure 5.** Illustration of the discrete affine hull. Each cell in the $6 \times 6$ grid represents an attribute combination, where observed combinations are solid-colored. The elements in blue form one connected component, $C_1$, and the elements in yellow form another connected component, $C_2$. Extrapolation is possible for unobserved combinations, represented by the crosshatched cells, as long as the test distribution samples from the Cartesian products of the connected components. The steps for completing the Cartesian product visually shows the intuition behind the extrapolation process.

which is a simple exercise (one sees that the expression is an increasing function of $c$ on $[a, 1]$ by deriving with respect to $c$, and that it is equal to 0 when $c = 1$).

$\square$

**Theorem 8.** *Consider the setting where $p(.|z)$ follows AED $\forall z \in \mathcal{Z}^{\times}$, $\mathcal{Z}^{\text{train}}$ comprises of $s$ attribute vectors $z$ drawn uniformly at random from $\mathcal{Z}^{\times}$, and the test distribution $q$ satisfies compositional shift characterization. If $s \geq 2d(m + 1 + \ln(d))$, $\hat{p}(z|x) = p(z|x), \forall z \in \mathcal{Z}^{\text{train}}, \forall x \in \mathbb{R}^n$, $\hat{q}(z) = q(z), \forall z \in \mathcal{Z}^{\text{test}}$, then the output of CRM (equation 10) matches the test distribution, i.e., $\hat{q}(z|x) = q(z|x), \forall z \in \mathcal{Z}^{\text{test}}, \forall x \in \mathbb{R}^n$, with probability greater than $1 - \frac{1}{c}$.*

*Proof.* Firstly, we can use Theorem 7 to conclude that $\text{DAff}(\mathcal{Z}^{\text{train}}) = \mathcal{Z}^{\times}$ with probability at least $1 - 1/c$. Owing to compositional shifts $\mathcal{Z}_{\text{test}} \subseteq \text{DAff}(\mathcal{Z}^{\text{train}})$. We can now use Theorem 2 to arrive at the result. $\square$

### D.5. Discrete Affine Hull: A Closer Look

In the next result, we aim to give a characterization of discrete affine hull that helps us give a two-dimensional visualization of $\text{DAff}(\mathcal{Z}^{\text{train}})$. Before we even state the result, we illustrate discrete affine hull of a $6 \times 6$ grid. Consider the $6 \times 6$ grid shown in Figure 5. The attribute combinations corresponding to the observed groups are shown as solid colored cells (blue and yellow). The light shaded elements (blue and yellow) denote the set of groups that belong to the affine hull of the solid colored groups. We now build the characterization that helps explain this visualization.

We introduce a graph on the attribute vectors observed. Each vertex corresponds to the attribute vector, i.e., $[z_1, z_2]$. There is an edge between two vertices if the Hamming distance between the attribute vectors is one. A connected component is a subgraph in which all vertices are connected, i.e., between every pair in the subgraph there exists a path. Let us start by making an observation about the connected components in this graph.

We consider a partition of observed groups into $K$ maximally connected components, $\{C_1, \cdots, C_K\}$. Define $C_{ij}$ as the set of values the $j^{th}$ component takes in the $i^{th}$ connected component. Observe that $C_{ij} \cap C_{lj} = \emptyset$ for $i \neq l$. Suppose this was not that case and $C_{ij} \cap C_{lj} \neq \emptyset$. In such a case, there exists a point in $C_i$ and another point in $C_l$ that share the $j^{th}$ component. As a result, the two points are connected by an edge and hence that would connect $C_i$ and $C_j$. This contradicts the fact that $C_i$ and $C_j$ are maximally connected, i.e., we cannot add another vertex to the graph while maintaining that there is a path between any two points in the component. In what follows, we will show that the afine hull of $C_j$ is $C_{j1} \times C_{j2}$,

which is the Cartesian product extension of set $C_j$. Next, we give some definitions and make a simple observation that allows us to think of sets $C_{j1} \times C_{j2}$ as subgrids, which are easier to visualize.

**Definition 1.** *Contiguous connected component: For each coordinate $j \in \{1, 2\}$, consider the smallest value and the largest value assumed by it in the connected component $C$ and call it $\min_j$ and $\max_j$. We say that the connected component $C$ is contigous if each value in the set $\{\min_j, \min_j +1, \cdots, \max_j -1, \max_j\}$ is assumed by some point in $C$ for all $j \in \{1, 2\}$.*

**Smallest subgrid containing a contigous connected component** $C$**:** The range of values assumed by $j^{th}$ coordinate in $C$, where $j \in \{1, 2\}$, are $\{\min_j, \cdots, \max_j\}$. The subgrid $\{\min_1, \cdots \max_1\} \times \{\min_2, \cdots \max_2\}$ is the smallest subgrid containing $C$. Observe that this subgrid is the smallest grid containing $C$ because if we drop any column or row, then some point taking that value in $C$ will not be in the subgrid anymore.

The groups observed at training time can be divided into $K$ maximally connected components $\{C_1, \cdots, C_K\}$. We argue that without any loss of generality each of these components are contiguous. Suppose some of the components in $\{C_1, \cdots, C_K\}$ are not contiguous. We relabel the first coordinate as $\pi(c_{i1}^r) = \sum_{j < i} |C_{j1}| + r$, where $c_{i1}^r$ is the $r^{th}$ point in $C_{i1}$. We can similarly relabel the second coordinate as well. Under the relabeled coordinates, each component is maximally connected and contiguous. Also, under this relabeling the Cartesian products $C_{j1} \times C_{j2}$ correspond to the smallest subgrid containing $C_j$. Let us go back to the setting of Figure 5. The sets of observed groups shown in solid blue and solid yellow form two connected components $C_1$ and $C_2$ respectively. Their Cartesian product extensions are shown as well in the Figure 5. Since the connected components were contiguous the Cartesian product extensions correspond to smallest subgrids containing the respective connected component.

**Theorem 9.** *Given the partition of training support as $\mathcal{Z}^{\mathsf{train}} = \{C_1, \cdots, C_K\}$, we have:*

- *The affine span of a contiguous connected component $C$ is the smallest subgrid that contains that connected component $C$.*

- *The affine span of the union over disjoint contiguous connected components is the union of the smallest subgrids that contain the respective connected components.*

*Proof.* $C$ denotes the connected component under consideration and the smallest subgrid containing it is $S$. Denote the affine span of $C$ as $A$. We first show that the subgrid $S \subseteq A$.

We start with a target point $t = (t_1, t_2)$ inside $S$. We want show that the one-hot concatention of this point $t$ can be expressed as an affine combination of the points in $C$.

Firstly, if $t$ is already in $C$, then the point is trivially in the affine span. If that is not the case, then let us proceed to more involved cases. Consider the shortest path joining a point of the form $(t_1, s_2) \in C$, where $s_2 \neq t_2$, and a point of the form $(s_1, t_2) \in C$, where $s_1 \neq t_1$. If such points do not exist, then $t$ cannot be in $S$, which is a contradiction.

We assign a weight of $(+1)$ to the concatenation of one-hot encodings of the point $(t_1, s_2)$. We then traverse the path until we encounter a point where $s_2$ changes, note that such a point has to occur because of existence of $(s_1, t_2)$ on the path. We call this point $v = (\tilde{s}_1', s_2')$. The point before $v$ on the path is $w = (\tilde{s}_1', s_2)$. We assign a weight of $(-1)$ to $w$. We summarize the path seen so far below. We also write the weights assigned to the points

$$
\begin{aligned}
s &= (t_1, s_2) && (+1) \\
u &= (s_1', s_2) \\
&\;\;\vdots && \\
w &= (\tilde{s}_1', s_2) && (-1) \\
v &= (\tilde{s}_1', s_2')
\end{aligned}
\tag{47}
$$

After $w$, we have a weight of $+1$ assigned to $t_1$, $-1$ assigned to $\tilde{s}_1'$ (note that $\tilde{s}_1'$ cannot be $t_1$, this follows from the fact that we are on shortest path between points of the form $(t_1, s_2)$ and $(s_1, t_2)$). We call this state $S_1$. After $w$, we wait for a point

on the path where $\tilde{s}_1^{'}$ changes or we reach the terminal state $(s_1, t_2)$. The latter can happen if $\tilde{s}_1^{'} = s_1$. In the latter case, we assign a weight $(+1)$ to the terminal state and thus the final weights are $(+1)$ for $t_1$ and $t_2$ and zero for everything else. This leads to the desired affine combination. We call this state $T_1$, corresponding to terminal state.

Now suppose we were in a situation where we reach a point $q = (s_1^+, \tilde{s}_2^{'})$. The point before $q$ is $r = (\tilde{s}_1^{'}, \tilde{s}_2^{'})$. We assign a weight of $(+1)$ to $r$. We summarize the path seen after encountering $w$ below.

$$
\begin{aligned}
v &= (\tilde{s}_1^{'}, s_2^{'}) \\
&\;\;\vdots \\
r &= (\tilde{s}_1^{'}, \tilde{s}_2^{'}) \qquad\qquad (+1) \\
q &= (s_1^+, \tilde{s}_2^{'})
\end{aligned}
\tag{48}
$$

After $r$, we have a weight of $+1$ assigned to $t_1$ and a weight of $+1$ assigned to $\tilde{s}_2^{'}$. We call this state $S_2$. After $r$, we wait for a point where $\tilde{s}_2^{'}$ changes. It could be that $\tilde{s}_2^{'}$ changes to $t_2$. The state before it is say $u = (s_1, \tilde{s}_2^{'})$ and last state $e = (s_1, t_2)$. Assign a weight of $-1$ to $u$ and assign a weight of $+1$ to $e$. Thus we achieve the target as affine combination of points on the path. We call this state $T_2$, corresponding to the terminal state.

Now let us consider the other possibility that the terminal state has not been reached. We call such a point $m = (\tilde{s}_1^+, \tilde{s}_2^+)$. The point that occurs before this point is $l = (\tilde{s}_1^+, \tilde{s}_2^{'})$. We assign a weight of $(-1)$ to $l$. We summarize the path taken below.

$$
\begin{aligned}
q &= (s_1^+, \tilde{s}_2^{'}) \\
&\;\;\vdots \\
l &= (\tilde{s}_1^+, \tilde{s}_2^{'}) \qquad\qquad (-1) \\
m &= (\tilde{s}_1^+, \tilde{s}_2^+)
\end{aligned}
\tag{49}
$$

After $l$, $t_1$ is assigned a weight of $+1$ and $\tilde{s}_1^+$ is assigned a weight of $-1$. We reach the state $S_1$ again. From this point on, the same steps repeat. We keep cycling between $S_1$ and $S_2$ until we reach the terminal state from either $S_1$ or $S_2$ at which point we achieve the desired affine combination. The cycling of states only goes on for a finite number of steps as the entire path we are concerned with has a finite length. We show the process in Figure 6. Thus $S \subseteq A$.

We now make an observation about the set $A$, which is the affine hull of set $C$. Suppose the first coordinate takes values between $\{\min_1, \cdots, \max_1\}$. The corresponding one-hot encodings of the first coordinate are written as $\{\mathrm{onehot}(\min_1), \cdots, \mathrm{onehot}(\max_1)\}$. Now consider a value $c$ which is not in $\{\min_1, \cdots, \max_1\}$. We claim that no affine combination of vectors in $\{\mathrm{onehot}(\min_1), \cdots, \mathrm{onehot}(\max_1)\}$ can lead to $\mathrm{onehot}(c)$. We justify this claim as follows. Observe that no vector in $\{\mathrm{onehot}(\min_1), \cdots, \mathrm{onehot}(\max_1)\}$ has a non-zero entry in the same coordinate where $\mathrm{onehot}(c)$ is also non-zero. Hence, any affine combination of vectors in $\{\mathrm{onehot}(\min_1), \cdots, \mathrm{onehot}(\max_1)\}$ will always have a zero weight in the entry where $\mathrm{onehot}(c)$ is non-zero. It is now clear that the first component of affine hull of $A$ is always between $\{\min_1, \cdots, \max_1\}$. Similarly, the second component of affine hull of $A$ is always between $\{\min_2, \cdots, \max_2\}$. Therefore, $A \subseteq S$. As a result, $A = S$. Another way to say this is that $\mathsf{DAff}(C_j) = C_{j1} \times C_{j2}$.

We now move to the second part of the theorem. We have already shown that $\mathsf{DAff}(C_j) = C_{j1} \times C_{j2}$. We now want to show that

$$
\mathsf{DAff}\Big( \bigcup_{j=1}^{K} C_j \Big) = \bigcup_{j=1}^{K} \Big( C_{j1} \times C_{j2} \Big)
$$

Observe that $\mathsf{DAff}(A) \subseteq \mathsf{DAff}(A \cup B)$ and $\mathsf{DAff}(B) \subseteq \mathsf{DAff}(A \cup B)$, which implies $\mathsf{DAff}(A) \cup \mathsf{DAff}(B) \subseteq \mathsf{DAff}(A \cup B)$. Therefore, from the first part and this observation it follows that $\bigcup_{j=1}^{K} \Big( C_{j1} \times C_{j2} \Big) \subseteq \mathsf{DAff}\big( \bigcup_{j=1}^{K} C_j \big)$. We now show $\mathsf{DAff}\big( \bigcup_{j=1}^{K} C_j \big) \subseteq \bigcup_{j=1}^{K} \Big( C_{j1} \times C_{j2} \Big)$.

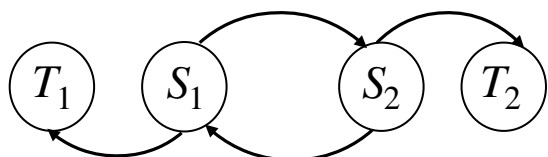

*Figure 6.* Illustration of state transition in proof of Theorem 9.

Take the $K$ maximally connected components $\{C_1, \cdots, C_K\}$ and let the set of respective smallest subgrids containing them be $\{S_1, \cdots, S_K\}$. Define a point $z'$ as the affine combination of points across these components as $z' = \sum_{i=1}^{K} \sum_{j=1}^{N_i} \alpha_{ij} z_{ij}$, where $z_{ij}$ is the $j^{th}$ point in $C_i$, which contains $N_i$ points. We can also write $z'$ as $z' = \sum_{i=1}^{K} \left( \sum_{j=1}^{N_i} \alpha_{ij} \right) \sum_{j=1}^{N_i} \frac{\alpha_{ij}}{\sum_{j=1}^{N_i} \alpha_{ij}} z_{ij}$. Define $z_i' = \sum_{j=1}^{N_i} \frac{\alpha_{ij}}{\sum_{j=1}^{N_i} \alpha_{ij}} z_{ij}$. Observe that $z_i'$ is in the affine combination of points in $C_i$ and hence $z_i'$ is a point in $S_i$. Let $\tilde{\alpha}_i = \sum_{j=1}^{N_i} \alpha_{ij}$. In this notation, we can see $z'$ is an affine combination of $z_i'$'s denoted as $\sum_{i=1}^{K} \tilde{\alpha}_i z_i'$. In this representation, there is at most one point per $S_i$ in the affine combination. There are two cases to consider. In the first case, exactly one component $\tilde{\alpha}_i$ is non-zero and rest all components are zero. In the second case, at least two components $\tilde{\alpha}_i$'s are non-zero. In this setting, we can only keep the non-zero $\tilde{\alpha}_i$'s in the sum denoted as $\sum_i \tilde{\alpha}_i z_i'$. Suppose $z_i' = (e_p, e_q)$ (without loss of generality), where $e_p$ is one-hot vector that is one on the $p^{th}$ coordinate. Observe that no other point in the sum $\sum_i \tilde{\alpha}_i z_i'$ will have a non-zero contribution on the $p^{th}$ coordinate. As a result, in the final vector the $p^{th}$ coordinate of the first attribute will take the value $0 < \tilde{\alpha}_i < 1$. This point is not a valid point in the set of all possible one-hot concatenations $\mathcal{Z}$ and hence it does not belong to the affine hull $\mathsf{DAff}\left( \bigcup_{j=1}^{K} C_j \right)$. Thus we are left with the first case. Observe that in the first case, we will always generate a point in one of the $\mathsf{DAff}(C_j)$, where $j \in \{1, \cdots, K\}$. Thus $\mathsf{DAff}\left( \bigcup_{j=1}^{K} C_j \right) \subseteq \bigcup_{j=1}^{K} \mathsf{DAff}(C_j)$, which implies $\mathsf{DAff}\left( \bigcup_{j=1}^{K} C_j \right) \subseteq \bigcup_{j=1}^{K} C_{j1} \times C_{j2}$. This completes the proof.

$\square$

### D.6. No Extrapolation beyond Discrete Affine Hull: Proof for Theorem 10

In this section, we rely on the characterization of discrete affine hulls shown in the previous section in Theorem 9. Suppose we learn an additive energy model to estimate $\hat{p}(x|z)$ and estimate the density $p(x|z)$ for all training groups using maximum likelihood. In this case, we know that $\hat{p}(x|z) = p(x|z)$ for all $z \in \mathsf{DAff}(\mathcal{Z}^{\mathsf{train}})$. In the next theorem, we show that such densities that satisfy $\hat{p}(x|z) = p(x|z)$ for all $z \in \mathsf{DAff}(\mathcal{Z}^{\mathsf{train}})$ may not match the true density outside the affine hull. In the next result, we assume that $\forall z \in \mathcal{Z}^{\times}, p(\cdot|z)$ is not uniform.

**Theorem 10.** *Suppose we learn an additive energy model to estimate $\hat{p}(x|z)$ and estimate the density $p(x|z)$ for all training groups. There exist densities that maximize likelihood and exactly match the training distributions but do not extrapolate to distributions outside the affine hull of $\mathcal{Z}^{\mathsf{train}}$, i.e., $\exists z \in \mathcal{Z}^{\times}$, where $\hat{p}(\cdot|z) \neq p(\cdot|z)$.*

*Proof.* We first take $\mathcal{Z}^{\mathsf{train}}$ and partition the groups into $K$ maximally connected components denoted $\{C_1, \cdots, C_K\}$. From Theorem 9, we know that the affine hull of $\mathcal{Z}^{\mathsf{train}}$ is the union of subgrids $\{S_1, \cdots, S_K\}$, where each subgrid $S_j$ is the Cartesian product $C_{j1} \times C_{j2}$.

Let us consider all points $(\tilde{z}_1, \tilde{z}_2)$ in some subgrid $S_k$. For each such $(\tilde{z}_1, \tilde{z}_2) \in S_k$, define $\hat{E}_1(x, \tilde{z}_1) = E_1(x, \tilde{z}_1) + \alpha_k(x)$, $\hat{E}_2(x, \tilde{z}_2) = E(x, \tilde{z}_2) - \alpha_k(x)$. Note that regardless of choice of $\alpha_k$ the density, $\hat{p}(x|z) = \frac{1}{\mathcal{Z}(z)} e^{-\langle \sigma(z), \hat{E}(x) \rangle}$ matches the true density $p(x|z)$ for all groups $z$ in $\bigcup_{i=1}^{K} S_i$.

Select any group $z_{\mathsf{ref}} = (z_1, z_2)$ that is not in the union of subgrids. From the definition of $\mathcal{Z}^{\times}$, it follows that there are points of the form $(z_1, z_2')$ in one of the subgrid $S_j$ and points of the form $(z_1', z_2)$ are in some subgrid $S_r$. Let $\alpha_j(x) = -\frac{E_1(x, z_1) + E_2(x, z_2)}{2}$ and $\alpha_r(x) = \frac{E_1(x, z_1) + E_2(x, z_2)}{2}$. Observe that $\hat{E}_1(x, z_1) + \hat{E}_2(x, z_2) = E_1(x, z_1) + E_2(x, z_2) +$

$\alpha_j(x) - \alpha_r(x) = 0$. Thus this choice of $\alpha_j(x) - \alpha_r(x)$ ensures that $\hat{p}(x|z_1, z_2)$ is uniform and hence cannot match the true $p(x|z_1, z_2)$.

This completes the proof.

$\square$

Based on the above proof, we now argue that there exist solutions to CRM that do not extrapolate outside the affine hull. Let us consider solutions to CRM denoted $\hat{E}, \hat{B}$, which satisfies the property that $\langle \sigma(z), \hat{E}(x) \rangle = \langle \sigma(z), E(x) \rangle$, $\hat{B}(z) = B(z) \forall z \in \mathcal{Z}^{\text{train}}$. Following the proof above, we can choose $\hat{E}'s$ in such a way that the sum of energies at a certain reference point outside the affine hull is zero and at all points inside the affine hull the sum of energies achieve a perfect match. For the group $z_{\text{ref}} = (z_1, z_2)$ not in the affine hull of $\mathcal{Z}^{\text{train}}$, we set $\hat{E}_1(x, z_1) + \hat{E}_2(x, z_2) = \langle \sigma(z), \hat{E}(x) \rangle = 0$.

Suppose $\hat{q}(z|x) = q(z|x), \forall z \in \mathsf{DAff}(\mathcal{Z}^{\text{train}}) \bigcup \{z_{\text{ref}}\}$. We now compute the likelihood ratio at $z_{\text{ref}}$ and a point $z \in \mathcal{Z}^{\text{train}}$. We obtain

$$
\begin{aligned}
&\frac{\hat{q}(z_{\text{ref}}|x)}{\hat{q}(z|x)} = \frac{q(z_{\text{ref}}|x)}{q(z|x)} \implies \\
&-\log\left(\frac{\hat{q}(z_{\text{ref}}|x)}{\hat{q}(z|x)}\right) = -\log\left(\frac{q(z_{\text{ref}}|x)}{q(z|x)}\right) \implies \\
&\langle \sigma(z_{\text{ref}}), \hat{E}(x) \rangle - \langle \sigma(z), \hat{E}(x) \rangle = \langle \sigma(z_{\text{ref}}), E(x) \rangle - \langle \sigma(z), E(x) \rangle - (\theta(z) - \theta(z_{\text{ref}}))
\end{aligned}
\tag{50}
$$

where $\theta(z)$ corresponds to collection of all terms that only depend on $z$. We already know that $\langle \sigma(z), \hat{E}(x) \rangle = \langle \sigma(z), E(x) \rangle$ and $\langle \sigma(z_{\text{ref}}), \hat{E}(x) \rangle = 0$. Substituting these into the above expression we obtain

$$
\langle \sigma(z_{\text{ref}}), E(x) \rangle = \theta(z) - \theta(z_{\text{ref}})
\tag{51}
$$

From the above condition, it follows that $q(x|z_{\text{ref}})$ is uniform. This implies that $p(x|z_{\text{ref}})$ is also uniform, which contradicts the condition that $p(x|z_{\text{ref}})$ is not uniform. Therefore, $\hat{q}(z|x) = q(z|x), \forall z \in \mathcal{Z}^{\text{train}} \bigcup \{z_{\text{ref}}\}$ cannot be true.

### D.7. Extrapolation of Discrete Additive Functions via Discrete Affine Hulls

Define a real-valued function $f(z_1, \cdots, z_m) = \sum_{j=1}^{m} f_j(z_j)$, where each $z_j \in \{1, \cdots, d\}, \forall j \in \{1, \cdots, m\}$. Define $\boldsymbol{f}_j = [f_j(1), \cdots, f_j(d)]$ and $\boldsymbol{f} = [\boldsymbol{f}_1, \cdots, \boldsymbol{f}_m]$. We can re-express the function $f$ in terms of one-hot encoding notation. $f(z_1, \cdots, z_m) = \langle \sigma(z), \boldsymbol{f} \rangle$. Given data $\{(z^j, y^j)\}_{j=1}^{s}$, where $y^j = f(z^j)$. Denote $S = \{z^j\}_{j=1}^{s}$. Given a new point $\tilde{z}$, which is not in the training data, we seek to predict the label $\tilde{y}$. We can exploit the additive structure of the function to predict $\tilde{y}$. Suppose $\tilde{z} \in \mathsf{DAff}(S)$. From this $\sigma(\tilde{z}) = \sum_{j=1}^{s} \alpha_j \sigma(z^j)$, where $\sum_j \alpha_j = 1$. We can express $\tilde{y}$ in terms of the seen data as follows. Observe that $\tilde{y} = \langle \sigma(\tilde{z}), \boldsymbol{f} \rangle = \langle \sum_{j=1}^{s} \alpha_j \sigma(z^j), \boldsymbol{f} \rangle = \sum_{j=1}^{n} \alpha_j \langle \sigma(z^j), \boldsymbol{f} \rangle = \sum_{j=1}^{n} \alpha_j y^j$. From this, it follows that we can perfectly predict the labels for points in the discrete affine hull. Thus from Theorem 7, it follows that if points in $S$ are sampled uniformly at random, then $O(md + d \log d)$ suffice to extrapolate to the entire grid of $d^m$ points and achieves CPE. In the work of (Dong & Ma, 2022), the authors also studied such discrete functions. However, their analysis does not propose the crucial object discrete affine hull, which gives a sharp characterization of extrapolation, and also do not provide sharp bounds on the number of samples needed for CPE. In (Dong & Ma, 2022), show that non-trivial extrapolation is achievable provided the bipartite graph induced the probability distribution over the seen data is connected.

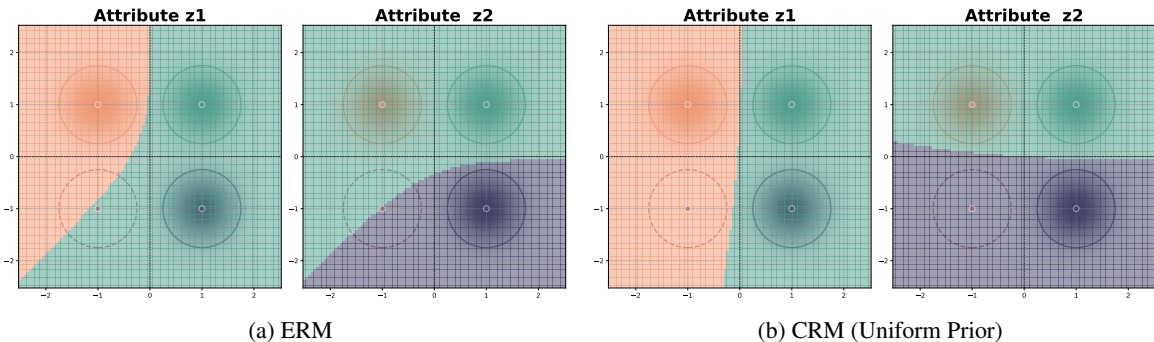

(a) ERM                   (b) CRM (Uniform Prior)

*Figure 7.* **Failure of ERM to generalize to unseen test group.** We consider the same task as in Figure 2, where we model a distribution with four Gaussian components. The group $(-1, -1)$ (pink dashed) has zero prior probability in the training distribution. Subfigure (a) shows the decision boundary obtained by an ERM-trained binary classifier for predicting the attribute $z_1$ (attribute on x-axis), or attribute $z_2$ (attribute on y-axis) respectively. ERM fails to learn the decision boundary y that would generalize well on samples from the missing group $(-1, -1)$,]. Subfigure b) shows the classifiers obtained using CRM with uniform test priors (and where we marginalize the predicted group probabilities to get $q(z_1|x)$ and $q(z_2|x)$). By contrast to ERM, CRM extrapolates and yields decision boundaries that generalize well on samples from the unseen group $(-1, -1)$. Decision regions were obtained from finite-data simulations, leading to minor imperfections.

## E. Additional Details on CRM's Adaptation to Test Distribution

### E.1. Derivation of Bayes Optimal Classifier

**Three group setting.** (Figure 2 left, train prior) Suppose the data is drawn from three groups $\{+1, +1), (-1, +1), (+1, -1)\}$, which are sampled with equal probability.

The Bayes optimal classifier predicts $(+1, +1)$ if

$$e^{-\|x-(1,1)\|^2} > e^{-\|x-(1,-1)\|^2} \implies \|x-(1,1)\|^2 < \|x-(1,-1)\|^2 \implies -2(x_1+x_2) < -2(x_1-x_2) \implies x_2 > 0$$
$$e^{-\|x-(1,1)\|^2} > e^{-\|x-(-1,1)\|^2} \implies \|x-(1,1)\|^2 < \|x-(-1,1)\|^2 \implies -2(x_1+x_2) < -2(-x_1+x_2) \implies x_1 > 0$$
$$\tag{52}$$

The Bayes optimal classifier predicts $(+1, -1)$ if

$$e^{-\|x-(1,-1)\|^2} > e^{-\|x-(1,1)\|^2} \implies \|x-(1,1)\|^2 > \|x-(1,-1)\|^2 \implies -2(x_1+x_2) > -2(x_1-x_2) \implies x_2 < 0$$
$$e^{-\|x-(1,-1)\|^2} > e^{-\|x-(-1,1)\|^2} \implies \|x-(1,1)\|^2 < \|x-(-1,1)\|^2 \implies -2(x_1-x_2) < -2(-x_1+x_2) \implies x_1 > x_2$$
$$\tag{53}$$

From same calculation it follows that the Bayes optimal classifier predicts $(-1, +1)$ if $x_1 < 0$ and $x_2 > x_1$.

**Four group setting.** (Figure 2 right, uniform prior) Suppose the data is drawn from four groups $\{+1, +1), (-1, +1), (+1, -1), (-1, -1)\}$, which are sampled with equal probability.

The Bayes optimal classifier for predicting the groups can be obtained using exactly same calculations as above.

The Bayes optimal classifier predicts: $(+1, +1)$ if $x_1 > 0$ and $x_2 > 0$, $(-1, 1)$ if $x_1 < 0$ and $x_2 > 0$, $(-1, -1)$ if $x_1 < 0$ and $x_2 < 0$, and $(-1, 1)$ if $x_1 < 0$ and $x_2 > 0$.

### E.2. Comparison with ERM

We also train ERM on the same training datasets where we observe data sampled uniformly from the groups $(+1, +1), (-1, +1), (+1, -1)$, but we don't observe data from the group $(-1, -1)$. Figure 7a shows the decision boundary of ERM for predicting the attribute $z_1$ (attribute on x-axis) and attribute $z_2$ (attribute on y-axis). Note that ERM fails to generalize to the novel group at test time (bottom left quadrant), while CRM with uniform test prior (Figure 7b) can adapt to the test distribution and extrapolate to the missing group.

# F. Experiments Setup

## F.1. Dataset Details

**Waterbirds (Wah et al., 2011).** The task is to classify land birds ($y = 0$) from water birds ($y = 1$), where the spurious attributes are land background ($a = 0$) and water background ($a = 1$). Hence, we have a total of 4 groups $z = (y, a)$ in the dataset.

**CelebA (Liu et al., 2015).** The task is to classify blond hair ($y = 1$) from non-blond hair ($y = 0$), where the spurious attribute is gender, female ($a = 0$) and male ($a = 1$). Hence, we have a total of 4 groups $z = (y, a)$ in the dataset.

**MetaShift (Liang & Zou, 2022).** The task is to classify cats ($y = 0$) from dogs ($y = 1$), where the spurious attribute is background, indoor ($a = 0$) and outdoor ($a = 1$). Hence, we have a total of 4 groups $z = (y, a)$ in the dataset.

**MultiNLI (Williams et al., 2017).** The task is to classify the relationship between the premise and hypothesis in a text document into one of the 3 classes: netural ($y = 0$), contradiction ($y = 1$), and entailment ($y = 2$). The spurious attribute are words like negation (binary attribute $a$), which are correlated with the contradiction class. Hence, we have a total of 6 groups $z = (y, a)$ in the dataset.

**CivilComments (Borkan et al., 2019).** The task is to classify whether a text document contains toxic language ($y = 0$) versus it doesn't contain toxic language ($y = 1$), where the spurious attribute $a$ corresponds to 8 different demographic identities (Male, Female, LGBTQ, Christian, Muslim, Other Religions, Black, and White). Hence, we have a total of 16 groups $z = (y, a)$ in the dataset.

**NICO++ (Zhang et al., 2023).** This is a a large scale (60 classes with 6 spurious attributes) domain generalization benchmark, and we follow the procedure in Yang et al. (2023b) where all the groups with less than 75 samples were dropped from training. This leaves us with 337 groups during training, however, the validation set still has samples from all the 360 groups. Hence, we additionally discard these groups from the validation set as well to design the compositional shift version.

| Dataset | Total Classes | Total Groups | Train Size | Val Size | Test Size |
|---------|---------------|--------------|------------|----------|-----------|
| Waterbirds | 2 | 4 | 4795 | 1199 | 5794 |
| CelebA | 2 | 4 | 162770 | 19867 | 19962 |
| MetaShift | 2 | 4 | 2276 | 349 | 874 |
| MultiNLI | 3 | 6 | 206175 | 82462 | 123712 |
| CivilComments | 2 | 16 | 148304 | 24278 | 71854 |
| NICO++ | 60 | 360 | 62657 | 8726 | 17483 |

*Table 3.* Statitics for the different benchmarks used in our experiments.

## F.2. Metric Details

Given the test distributions $z = (y, a) \sim q(z)$ and $x \sim q(x|z)$, lets denote the corresponding class predictions as $\hat{y} = \hat{M}(x)$ as per the method $\hat{M}$. Then average accuracy is defined as follows:

$$\text{Average Acc} := \mathbb{E}_{(y,a)} \mathbb{E}_{x \sim q(x|z)} \big[ \mathbb{1}[y == \hat{M}(x)] \big]$$

Hence, this denotes the mean accuracy with groups drawn as per the test distribution $q(z)$. However, if certain (majority) groups have a higher probability of being sampled than others (minority groups) as per the distribution $q(z|x)$, then the average accuracy metric is more sensitive to mis-classifications in majority groups as compared to the minority groups. Hence, a method can achieve high average accuracy even though its accuracy for the minority groups might be low.

Therefore, we use the worst group accuracy metric, defined as follows.

$$\text{Worst Group Acc} := min_{(y,a) \in \mathcal{Z}^{\text{test}}} \mathbb{E}_{x \sim q(x|z)} \big[ \mathbb{1}[y == \hat{M}(x)] \big]$$

Essentially we compute the accuracy for each group $(y, a) \sim q(z)$ as $\mathbb{E}_{x \sim q(x|z)} \big[ \mathbb{1}[y == \hat{M}(x)] \big]$ and then report the worst performance over all the groups. This metrics has been widely used for evaluating methods for subpopulation shifts (Sagawa

et al., 2019; Yang et al., 2023b).

Similarly, we define the group balanced accuracy (Tsirigotis et al., 2024) as follows, where we compute the average of all per-group accuracy $\mathbb{E}_{x \sim q(x|z)}\big[\mathbb{1}[y == \hat{M}(x)]\big]$.

$$\text{Group Balanced Acc} := \frac{1}{|\mathcal{Z}^{\text{test}}|} \sum_{(y,a) \in \mathcal{Z}^{\text{test}}} \mathbb{E}_{x \sim q(x|z)}\big[\mathbb{1}[y == \hat{M}(x)]\big]$$

### F.3. Method Details

For all the methods we have a *pre-trained* representation network backbone with linear classifier heads. We use ResNet-50 (He et al., 2016) for the vision datasets (Waterbirds, CelebA, MetaShift, NICO++) and BERT (Devlin et al., 2018) for the text datasets (MultiNLI, CivilComments). The parameters of both the representation network and linear classifier are updated with the same learning rate, and do not employ any special fine-tuning strategy for the representation network. For vision datasets we use the SGD optimizer (default values for momemtum 0.9), while for the text datasets we use the AdamW optimizer (Paszke et al., 2017) (default values for beta (0.9, 0.999) ).

**Hyperparameter Selection.** We rely on the group balanced accuracy on the validation set to determine the optimal hyperparameters. We specify the grids for each hyperparameter in Table 4, and train each method with 5 randomly drawn hyperparameters. The grid sizes for hyperparameter selection were designed following Pezeshki et al. (2023).

| Dataset | Learning Rate | Weight Decay | Batch Size | Total Epochs |
|---------|---------------|--------------|------------|--------------|
| Waterbirds | $10^{\text{Uniform}(-5,-3)}$ | $10^{\text{Uniform}(-6,-3)}$ | $2^{\text{Uniform}(5,7)}$ | 5000 |
| CelebA | $10^{\text{Uniform}(-5,-3)}$ | $10^{\text{Uniform}(-6,-3)}$ | $2^{\text{Uniform}(5,7)}$ | 10000 |
| MetaShift | $10^{\text{Uniform}(-5,-3)}$ | $10^{\text{Uniform}(-6,-3)}$ | $2^{\text{Uniform}(5,7)}$ | 5000 |
| MulitNLI | $10^{\text{Uniform}(-6,-4)}$ | $10^{\text{Uniform}(-6,-3)}$ | $2^{\text{Uniform}(4,6)}$ | 10000 |
| CivilComments | $10^{\text{Uniform}(-6,-4)}$ | $10^{\text{Uniform}(-6,-3)}$ | $2^{\text{Uniform}(4,6)}$ | 10000 |
| NICO++ | $10^{\text{Uniform}(-5,-3)}$ | $10^{\text{Uniform}(-6,-3)}$ | $2^{\text{Uniform}(5,7)}$ | 10000 |

*Table 4.* Details about the grids for hyperparameter selection. The choices for grid sizes were taken from Pezeshki et al. (2023).

# G. Additional Results

## G.1. Results Averaged over all the Compositional Shift Scenarios

We provide complete results on benchmarking CRM for compositional shifts on the datasets Waterbirds (Wah et al., 2011), CelebA (Liu et al., 2015), MetaShift (Liang & Zou, 2022), MultiNLI (Williams et al., 2017), CivilComments (Borkan et al., 2019), and NICO++ dataset (Zhang et al., 2023). We compare CRM against 7 baselines; ERM, Group Distributionally Robust Optimization (GroupDRO) (Sagawa et al., 2019), Logit Correction (LC) (Liu et al., 2022b), supervised logit adjustment (sLA) (Tsirigotis et al., 2024), Invariant Risk Minimization (IRM) (Arjovsky et al., 2019), Risk Extrapolation (VREx) (Krueger et al., 2021), and Mixup (Zhang et al., 2017).

Table 5 provides full results for the same, with groups balanced accuracy, standard across 3 random seeds, and baselines IRM, VREx, and Mixup, that were excluded from Table 1 in the main paper. We find that CRM is either competitive or outperforms all the baselines w.r.t. both group balanced accuracy and the worst group accuracy. Also, CRM outperforms the baselines IRM, VREx, and Mixup by a wide margin w.r.t WGA. Hence, we had decided to drop these baselines from the Table 1 in the main paper to show succinct comparison with strong baselines.

Further, in scenarios where CRM is competitive with baselines on WGA (Waterbirds, MultiNLI, CivilComments), please note that here we aggregate performance over multiple compositional shift scenarios. When we analyze the worst case compositional shift scenario, we find that CRM outperforms all the baselines. Please check the next subsection (Appendix G.2) for detailed results in each compositional shift scenario.

## G.2. Detailed Results for all the Compositional Shift Scenarios

We now present resutls for all the compositional shift scenarios associated with each benchmark; Waterbirds (Table 6), CelebA (Table 7), MetaShift (Table 8), MultiNLI (Table 9), and CivilComments (Table 10, Table 11). Here we do not aggregate over the multiple compositional shift scenarios of a benchmark, and provide a more detailed analysis with results for each scenario. For each method, we further highlight the worst case scenario for it, i.e, the scenario with the lowest worst group accuracy amongst all the compositional shift scenarios. This helps us easily compare the performance of methods for the respective worst case compositional shift scenario, as opposed to the average over all scenarios in Table 1. An interesting finding is that CRM outperforms all the baselines in the respective worst case compositional shift scenarios by a wide margin, especially w.r.t worst-group accuracy.

| Dataset | Method | Average Acc | Balanced Acc | Worst Group Acc | Worst Group Acc (No Groups Dropped) |
|---|---|---|---|---|---|
| Waterbirds | ERM | 77.9 (0.1) | 75.3 (0.1) | 43.0 (0.2) | 62.3 (1.2) |
| | G-DRO | 77.9 (0.9) | 78.8 (0.7) | 42.3 (2.6) | 87.3 (0.3) |
| | LC | 88.3 (0.9) | 86.9 (0.6) | 75.5 (1.8) | 88.7 (0.3) |
| | sLA | 89.3 (0.4) | 87.5 (0.4) | 77.3 (1.4) | 89.7 (0.3) |
| | IRM | 73.6 (0.8) | 70.4 (0.3) | 28.7 (2.2) | 72.3 (1.2) |
| | VREx | 81.0 (0.6) | 80.0 (0.5) | 45.6 (1.1) | 84.3 (0.7) |
| | Mixup | 81.6 (0.1) | 79.9 (0.1) | 52.2 (0.4) | 69.7 (0.9) |
| | CRM | 87.1 (0.7) | 87.8 (0.1) | 78.7 (1.0) | 86.0 (0.6) |
| CelebA | ERM | 85.8 (0.3) | 75.6 (0.1) | 39.0 (0.3) | 52.0 (1.0) |
| | G-DRO | 89.2 (0.5) | 86.8 (0.1) | 67.8 (0.8) | 91.0 (0.6) |
| | LC | 91.1 (0.2) | 83.5 (0.0) | 57.4 (0.5) | 90.0 (0.6) |
| | sLA | 90.9 (0.2) | 83.6 (0.3) | 57.4 (1.3) | 86.7 (1.9) |
| | IRM | 80.4 (1.3) | 76.7 (1.1) | 40.1 (2.4) | 67.7 (3.5) |
| | VREx | 86.2 (0.3) | 82.8 (0.5) | 49.2 (2.1) | 89.0 (0.6) |
| | Mixup | 84.9 (0.2) | 77.9 (0.2) | 42.8 (0.9) | 62.0 (1.0) |
| | CRM | 91.1 (0.2) | 89.2 (0.0) | 81.8 (0.5) | 89.0 (0.6) |
| MetaShift | ERM | 85.7 (0.4) | 81.7 (0.3) | 60.5 (0.5) | 63.0 (0.0) |
| | G-DRO | 86.0 (0.3) | 82.6 (0.2) | 63.8 (1.1) | 80.7 (1.3) |
| | LC | 88.5 (0.0) | 85.0 (0.0) | 68.2 (0.5) | 80.0 (1.2) |
| | sLA | 88.4 (0.1) | 84.0 (0.0) | 63.0 (0.5) | 80.0 (1.2) |
| | IRM | 83.7 (0.3) | 80.3 (0.4) | 55.8 (1.0) | 69.3 (2.4) |
| | VREx | 84.9 (0.4) | 81.7 (0.3) | 59.9 (0.2) | 75.3 (2.2) |
| | Mixup | 86.8 (0.0) | 82.8 (0.1) | 62.8 (0.7) | 68.3 (2.7) |
| | CRM | 87.6 (0.3) | 84.7 (0.2) | 73.4 (0.4) | 74.7 (1.5) |
| MultiNLI | ERM | 68.4 (2.1) | 68.1 (1.9) | 7.5 (1.3) | 68.0 (1.7) |
| | G-DRO | 70.4 (0.2) | 73.7 (0.2) | 34.3 (0.2) | 57.0 (2.3) |
| | LC | 75.9 (0.1) | 77.3 (0.2) | 54.3 (1.0) | 74.3 (1.2) |
| | sLA | 76.4 (0.3) | 77.4 (0.2) | 55.0 (1.5) | 71.7 (0.3) |
| | IRM | 65.7 (0.1) | 63.7 (0.4) | 8.1 (0.8) | 54.3 (2.4) |
| | VREx | 69.0 (0.0) | 68.8 (0.2) | 4.1 (0.3) | 69.7 (0.3) |
| | Mixup | 70.2 (0.1) | 69.7 (0.1) | 14.6 (1.0) | 63.7 (2.9) |
| | CRM | 74.3 (0.3) | 76.1 (0.3) | 58.7 (1.4) | 74.7 (1.3) |
| CivilComments | ERM | 80.4 (0.2) | 78.4 (0.0) | 55.9 (0.2) | 61.0 (2.5) |
| | G-DRO | 80.1 (0.1) | 78.9 (0.0) | 61.6 (0.5) | 64.7 (1.5) |
| | LC | 80.7 (0.1) | 79.0 (0.0) | 65.7 (0.5) | 67.3 (0.3) |
| | sLA | 80.6 (0.1) | 79.1 (0.0) | 65.6 (0.2) | 66.3 (0.9) |
| | IRM | 79.7 (0.2) | 78.0 (0.0) | 53.5 (0.5) | 60.3 (1.5) |
| | VREx | 79.8 (0.1) | 78.7 (0.1) | 57.5 (0.4) | 63.3 (1.5) |
| | Mixup | 80.1 (0.1) | 78.2 (0.0) | 55.4 (0.6) | 61.3 (1.5) |
| | CRM | 83.7 (0.1) | 78.4 (0.0) | 67.9 (0.5) | 70.0 (0.6) |
| NICO++ | ERM | 85.0 (0.0) | 85.0 (0.0) | 35.3 (2.3) | 35.3 (2.3) |
| | G-DRO | 84.0 (0.0) | 83.7 (0.3) | 36.7 (0.7) | 33.7 (1.2) |
| | LC | 85.0 (0.0) | 85.0 (0.0) | 35.3 (2.3) | 35.3 (2.3) |
| | sLA | 85.0 (0.0) | 85.0 (0.0) | 33.0 (0.0) | 35.3 (2.3) |
| | IRM | 64.0 (0.6) | 62.7 (0.3) | 0.0 (0.0) | 0.0 (0.0) |
| | VREx | 86.0 (0.0) | 86.0 (0.0) | 37.3 (4.3) | 38.0 (5.0) |
| | Mixup | 85.0 (0.0) | 84.7 (0.3) | 33.0 (0.0) | 33.0 (0.0) |
| | CRM | 84.7 (0.3) | 84.7 (0.3) | 40.3 (4.3) | 39.0 (3.2) |

*Table 5.* **Robustness under compositional shift.** We compare the proposed Compositional Risk Minimization (CRM) method with 7 baselines. We report various metrics, averaged as a group is dropped from training and validation sets. Last column is WGA under the dataset's standard subpopulation shift benchmark, i.e. with no group dropped. All methods have a harder time to generalize when groups are absent from training, but CRM appears consistently more robust (standard error based on 3 random seeds).

| Discarded Group $(y, a)$ | Method | Average Acc | Balanced Acc | Worst Group Acc |
|---|---|---|---|---|
| (0, 0) | ERM | 74.0 (0.0) | 82.3 (0.3) | 67.0 (0.0) |
| | G-DRO | 77.3 (0.7) | 83.0 (0.6) | 59.7 (1.9) |
| | LC | 85.7 (0.3) | 88.7 (0.3) | 82.0 (0.6) |
| | sLA | 86.0 (0.0) | 89.0 (0.0) | 82.3 (0.3) |
| | IRM | 72.7 (2.0) | 81.3 (1.2) | 58.7 (3.2) |
| | VREx | 80.0 (1.5) | 85.7 (0.9) | 69.3 (2.4) |
| | Mixup | 87.3 (0.3) | 89.3 (0.3) | 84.3 (0.3) |
| | CRM | 86.7 (0.9) | 88.7 (0.3) | 83.0 (1.5) |
| (0, 1) | ERM | 67.3 (0.3) | 71.7 (0.3) | 28.0 (1.2) |
| | G-DRO | 58.3 (3.2) | 70.7 (2.0) | 11.7 (4.6) |
| | LC | 82.7 (3.2) | 86.0 (1.7) | 72.0 (5.8) |
| | sLA | 86.3 (1.7) | 88.0 (1.0) | 78.7 (3.3) |
| | IRM | 55.7 (2.2) | 67.7 (0.9) | 7.7 (3.7) |
| | VREx | 66.0 (1.0) | 74.0 (0.6) | 22.7 (1.9) |
| | Mixup | 65.7 (0.3) | 73.0 (0.0) | 19.7 (0.3) |
| | CRM | 86.0 (2.1) | 86.7 (0.7) | 73.0 (4.2) |
| (1, 0) | ERM | 84.0 (0.0) | 78.0 (0.0) | 38.3 (0.3) |
| | G-DRO | 90.0 (0.0) | 86.0 (0.6) | 67.0 (3.6) |
| | LC | 93.0 (0.0) | 89.0 (0.6) | 79.0 (1.2) |
| | sLA | 93.0 (0.0) | 89.0 (0.6) | 79.3 (1.5) |
| | IRM | 87.7 (0.3) | 81.3 (0.7) | 48.0 (4.0) |
| | VREx | 89.7 (0.3) | 82.7 (0.3) | 50.3 (1.7) |
| | Mixup | 84.3 (0.3) | 80.7 (0.3) | 51.7 (2.0) |
| | CRM | 86.7 (0.3) | 89.0 (0.0) | 83.7 (0.3) |
| (1, 1) | ERM | 86.3 (0.3) | 69.3 (0.3) | 38.7 (0.7) |
| | G-DRO | 86.0 (0.6) | 75.7 (2.2) | 31.0 (9.2) |
| | LC | 92.0 (0.0) | 84.0 (0.6) | 69.0 (1.5) |
| | sLA | 92.0 (0.0) | 84.0 (0.6) | 69.0 (1.5) |
| | IRM | 78.3 (0.3) | 51.3 (0.9) | 0.3 (0.3) |
| | VREx | 88.3 (0.7) | 77.7 (1.8) | 40.0 (5.5) |
| | Mixup | 89.0 (0.0) | 76.7 (0.3) | 53.0 (0.6) |
| | CRM | 89.0 (0.6) | 86.7 (0.7) | 75.0 (3.2) |

*Table 6.* Results for the various compositional shift scenarios for the **Waterbirds** benchmark. For each metric, report the mean (standard error) over 3 random seeds on the test dataset. We highlight the worst case compositional shift scenario for each method, i.e, the scenario with the lowest worst group accuracy amongst all the compositional shift scenarios. CRM outperforms all the baselines in the respective worst case compositional shift scenarios.

| Discarded Group $(y, a)$ | Method | Average Acc | Balanced Acc | Worst Group Acc |
|---|---|---|---|---|
| $(0, 0)$ | ERM | 68.7 (0.3) | 74.0 (0.0) | 37.7 (0.3) |
| | G-DRO | 85.0 (0.6) | 88.0 (0.0) | 75.0 (1.2) |
| | LC | 88.0 (0.0) | 90.3 (0.3) | 82.3 (0.3) |
| | sLA | 87.7 (0.3) | 90.3 (0.3) | 82.3 (0.7) |
| | IRM | 65.0 (2.5) | 72.0 (0.6) | 31.3 (4.7) |
| | VREx | 82.7 (0.3) | 87.7 (0.3) | 70.7 (0.9) |
| | Mixup | 64.7 (0.3) | 72.0 (0.0) | 29.3 (0.7) |
| | CRM | 91.7 (0.3) | 89.3 (0.3) | 81.0 (2.0) |
| $(0, 1)$ | ERM | 91.3 (0.9) | 91.0 (0.6) | 86.7 (1.3) |
| | G-DRO | 85.0 (1.5) | 88.7 (0.7) | 72.7 (3.7) |
| | LC | 93.0 (0.6) | 87.7 (0.9) | 71.0 (1.7) |
| | sLA | 92.7 (0.3) | 88.0 (0.0) | 71.3 (0.9) |
| | IRM | 77.0 (2.5) | 83.7 (1.3) | 53.0 (4.5) |
| | VREx | 78.7 (0.9) | 85.0 (0.6) | 55.7 (1.9) |
| | Mixup | 92.0 (0.6) | 91.3 (0.3) | 87.3 (0.9) |
| | CRM | 88.3 (0.9) | 91.0 (0.6) | 85.0 (2.0) |
| $(1, 0)$ | ERM | 87.0 (0.0) | 59.3 (0.3) | 4.0 (0.0) |
| | G-DRO | 91.7 (0.3) | 86.3 (0.7) | 71.7 (0.9) |
| | LC | 88.3 (0.3) | 70.7 (0.7) | 21.0 (2.1) |
| | sLA | 88.3 (0.3) | 71.0 (0.6) | 21.3 (1.9) |
| | IRM | 84.7 (1.5) | 72.7 (3.9) | 47.3 (10.3) |
| | VREx | 88.3 (0.7) | 79.3 (1.8) | 40.3 (7.9) |
| | Mixup | 88.0 (0.0) | 67.7 (0.3) | 17.3 (1.7) |
| | CRM | 93.0 (0.0) | 85.7 (0.3) | 73.3 (1.8) |
| $(1, 1)$ | ERM | 96.0 (0.0) | 78.0 (0.6) | 27.7 (2.0) |
| | G-DRO | 95.0 (0.0) | 84.3 (0.3) | 51.7 (1.2) |
| | LC | 95.0 (0.0) | 85.3 (0.3) | 55.3 (1.9) |
| | sLA | 95.0 (0.0) | 85.0 (0.6) | 54.7 (2.3) |
| | IRM | 95.0 (0.0) | 78.3 (0.7) | 28.7 (3.0) |
| | VREx | 95.0 (0.0) | 79.0 (0.0) | 30.3 (0.9) |
| | Mixup | 95.0 (0.0) | 80.7 (0.3) | 37.0 (1.2) |
| | CRM | 91.3 (0.3) | 91.0 (0.0) | 88.0 (0.6) |

*Table 7.* Results for the various compositional shift scenarios for the **CelebA** benchmark. For each metric, report the mean (standard error) over 3 random seeds on the test dataset. We highlight the worst case compositional shift scenario for each method, i.e, the scenario with the lowest worst group accuracy amongst all the compositional shift scenarios. CRM outperforms all the baselines in the respective worst case compositional shift scenarios.

| Discarded Group $(y, a)$ | Method | Average Acc | Balanced Acc | Worst Group Acc |
|---|---|---|---|---|
| | ERM | 84.3 (0.3) | 84.0 (0.6) | 80.3 (0.9) |
| | G-DRO | 84.0 (0.6) | 83.3 (0.7) | 78.0 (0.6) |
| | LC | 89.0 (0.0) | 85.7 (0.3) | 74.3 (1.8) |
| $(0, 0)$ | sLA | 90.0 (0.0) | 85.0 (0.0) | 67.3 (1.9) |
| | IRM | 81.0 (0.6) | 81.7 (0.3) | 70.0 (1.5) |
| | VREx | 83.7 (0.9) | 83.0 (0.6) | 76.7 (1.9) |
| | Mixup | 85.0 (0.0) | 84.0 (0.0) | 77.0 (0.6) |
| | CRM | 87.3 (0.3) | 84.3 (0.3) | 73.3 (0.7) |
| | ERM | 85.0 (0.0) | 79.0 (0.0) | 49.0 (0.0) |
| | G-DRO | 86.0 (1.0) | 81.7 (0.3) | 55.3 (3.2) |
| | LC | 86.0 (0.0) | 84.0 (0.0) | 63.7 (0.3) |
| $(0, 1)$ | sLA | 86.0 (0.0) | 84.0 (0.0) | 64.0 (0.6) |
| | IRM | 83.7 (0.3) | 78.7 (0.9) | 44.7 (1.9) |
| | VREx | 84.0 (0.0) | 81.0 (0.0) | 48.3 (0.7) |
| | Mixup | 86.0 (0.0) | 80.0 (0.0) | 52.7 (0.3) |
| | CRM | 88.3 (0.3) | 85.7 (0.3) | 78.0 (1.0) |
| | ERM | 90.0 (0.0) | 82.0 (0.0) | 48.3 (0.3) |
| | G-DRO | 90.3 (0.3) | 82.7 (0.9) | 52.7 (2.3) |
| | LC | 90.0 (0.0) | 84.3 (0.3) | 62.0 (0.0) |
| $(1, 0)$ | sLA | 88.7 (0.3) | 81.0 (0.0) | 46.7 (0.7) |
| | IRM | 90.0 (0.0) | 81.7 (0.3) | 49.3 (1.7) |
| | VREx | 90.0 (0.0) | 82.0 (0.0) | 48.3 (0.3) |
| | Mixup | 90.3 (0.3) | 82.7 (0.3) | 52.3 (1.7) |
| | CRM | 87.0 (1.2) | 83.3 (0.7) | 70.0 (1.0) |
| | ERM | 83.3 (1.2) | 81.7 (0.9) | 64.3 (1.2) |
| | G-DRO | 83.7 (0.9) | 82.7 (0.9) | 69.3 (2.0) |
| | LC | 89.0 (0.0) | 86.0 (0.0) | 72.7 (0.7) |
| $(1, 1)$ | sLA | 89.0 (0.0) | 86.0 (0.0) | 74.0 (0.0) |
| | IRM | 80.0 (0.6) | 79.0 (0.6) | 59.0 (1.0) |
| | VREx | 82.0 (0.6) | 81.0 (0.6) | 66.3 (0.9) |
| | Mixup | 85.7 (0.3) | 84.3 (0.3) | 69.3 (0.9) |
| | CRM | 87.7 (0.3) | 85.3 (0.3) | 72.3 (1.7) |

*Table 8.* Results for the various compositional shift scenarios for the **MetaShift** benchmark. For each metric, report the mean (standard error) over 3 random seeds on the test dataset. We highlight the worst case compositional shift scenario for each method, i.e, the scenario with the lowest worst group accuracy amongst all the compositional shift scenarios. CRM outperforms all the baselines in the respective worst case compositional shift scenarios.

| Discarded Group $(y, a)$ | Method | Average Acc | Balanced Acc | Worst Group Acc |
|---|---|---|---|---|
| (0, 0) | ERM | 62.7 (0.3) | 66.7 (0.3) | 0.7 (0.3) |
| | G-DRO | 63.3 (0.3) | 68.0 (0.0) | 1.7 (0.7) |
| | LC | 68.0 (0.0) | 72.0 (0.0) | 20.0 (0.0) |
| | sLA | 67.7 (0.3) | 72.0 (0.0) | 19.7 (1.5) |
| | IRM | 61.3 (0.7) | 64.3 (0.3) | 4.3 (0.9) |
| | VREx | 63.3 (0.3) | 68.7 (0.3) | 7.0 (1.0) |
| | Mixup | 63.0 (0.0) | 64.3 (0.3) | 1.0 (0.0) |
| | CRM | 64.7 (0.9) | 70.7 (0.9) | 31.0 (5.6) |
| (0, 1) | ERM | 77.7 (0.3) | 71.7 (0.3) | 14.0 (1.0) |
| | G-DRO | 80.7 (0.7) | 80.7 (0.7) | 74.0 (1.0) |
| | LC | 81.0 (0.0) | 81.0 (0.0) | 75.3 (0.3) |
| | sLA | 81.3 (0.3) | 80.7 (0.3) | 69.0 (0.6) |
| | IRM | 74.0 (0.0) | 68.7 (0.3) | 10.0 (1.5) |
| | VREx | 76.0 (0.0) | 69.7 (0.3) | 4.0 (0.6) |
| | Mixup | 78.3 (0.3) | 74.3 (0.3) | 34.0 (2.0) |
| | CRM | 80.0 (0.6) | 78.0 (1.2) | 62.3 (8.2) |
| (1, 0) | ERM | 58.0 (0.0) | 67.0 (0.0) | 0.0 (0.0) |
| | G-DRO | 57.7 (0.3) | 67.7 (0.3) | 0.0 (0.0) |
| | LC | 70.7 (0.9) | 74.3 (0.3) | 47.3 (4.3) |
| | sLA | 73.3 (2.7) | 76.3 (1.7) | 58.3 (9.7) |
| | IRM | 49.0 (0.6) | 55.3 (1.5) | 0.7 (0.3) |
| | VREx | 55.3 (0.3) | 65.7 (0.3) | 1.0 (0.0) |
| | Mixup | 57.0 (0.0) | 66.0 (0.0) | 0.0 (0.0) |
| | CRM | 69.5 (0.5) | 74.0 (0.0) | 63.5 (0.5) |
| (1, 1) | ERM | 82.0 (0.2) | 73.0 (0.2) | 20.0 (1.2) |
| | G-DRO | 80.3 (0.3) | 79.3 (0.3) | 72.7 (0.9) |
| | LC | 81.7 (0.3) | 81.3 (0.3) | 74.3 (1.5) |
| | sLA | 82.0 (0.0) | 81.0 (0.0) | 75.3 (0.7) |
| | IRM | 76.3 (0.3) | 67.7 (1.2) | 23.0 (7.0) |
| | VREx | 79.7 (0.3) | 69.0 (0.0) | 3.0 (1.5) |
| | Mixup | 80.7 (0.3) | 72.3 (0.3) | 28.3 (3.8) |
| | CRM | 81.3 (0.3) | 80.7 (0.3) | 71.3 (1.8) |
| (2, 0) | ERM | 62.0 (0.0) | 68.3 (0.3) | 0.0 (0.0) |
| | G-DRO | 60.0 (0.0) | 67.7 (0.3) | 0.0 (0.0) |
| | LC | 72.3 (0.3) | 74.7 (0.3) | 48.7 (0.7) |
| | sLA | 72.7 (0.7) | 74.3 (0.3) | 48.3 (0.9) |
| | IRM | 57.0 (0.6) | 57.3 (0.3) | 0.0 (0.0) |
| | VREx | 59.7 (0.3) | 67.7 (0.3) | 0.0 (0.0) |
| | Mixup | 61.0 (0.0) | 66.3 (0.3) | 0.0 (0.0) |
| | CRM | 68.7 (0.3) | 72.7 (0.3) | 50.0 (0.6) |
| (2, 1) | ERM | 81.3 (0.3) | 74.3 (0.3) | 17.3 (2.4) |
| | G-DRO | 80.7 (0.3) | 79.0 (0.0) | 57.3 (2.2) |
| | LC | 82.0 (0.0) | 80.7 (0.3) | 60.0 (1.2) |
| | sLA | 81.7 (0.3) | 80.3 (0.3) | 59.3 (0.9) |
| | IRM | 76.3 (0.3) | 69.0 (0.0) | 10.7 (0.9) |
| | VREx | 80.0 (0.0) | 72.3 (0.3) | 9.7 (1.2) |
| | Mixup | 81.0 (0.0) | 75.0 (0.6) | 24.3 (3.0) |
| | CRM | 81.3 (0.3) | 80.0 (0.6) | 72.7 (0.9) |

*Table 9.* Results for the various compositional shift scenarios for the **MultiNLI** benchmark. For each metric, report the mean (standard error) over 3 random seeds on the test dataset. We highlight the worst case compositional shift scenario for each method, i.e, the scenario with the lowest worst group accuracy amongst all the compositional shift scenarios. CRM outperforms all the baselines in the respective worst case compositional shift scenarios.

| Discarded Group $(y, a)$ | Method | Average Acc | Balanced Acc | Worst Group Acc |
|---|---|---|---|---|
| $(0, 0)$ | ERM | 79.0 (0.6) | 78.7 (0.3) | 61.3 (1.5) |
| | G-DRO | 79.3 (1.2) | 79.0 (0.0) | 64.7 (3.0) |
| | LC | 79.7 (0.3) | 79.0 (0.0) | 64.3 (0.9) |
| | sLA | 79.7 (0.3) | 79.3 (0.3) | 66.7 (1.8) |
| | IRM | 77.7 (0.3) | 78.0 (0.0) | 60.3 (0.3) |
| | VREx | 79.3 (0.3) | 79.0 (0.0) | 65.0 (0.0) |
| | Mixup | 77.7 (0.3) | 78.3 (0.3) | 58.7 (1.9) |
| | CRM | 84.0 (0.0) | 78.7 (0.3) | 67.0 (2.5) |
| $(0, 1)$ | ERM | 78.0 (0.6) | 78.3 (0.3) | 64.3 (1.2) |
| | G-DRO | 78.0 (0.6) | 78.7 (0.3) | 64.3 (1.5) |
| | LC | 79.3 (0.3) | 79.0 (0.0) | 64.3 (0.9) |
| | sLA | 79.7 (0.3) | 79.0 (0.0) | 65.3 (0.3) |
| | IRM | 77.3 (0.7) | 78.0 (0.0) | 62.3 (2.6) |
| | VREx | 77.3 (0.7) | 79.0 (0.0) | 66.0 (1.2) |
| | Mixup | 78.0 (0.6) | 78.3 (0.3) | 62.3 (2.4) |
| | CRM | 83.3 (0.7) | 78.7 (0.3) | 71.0 (1.5) |
| $(0, 2)$ | ERM | 78.3 (0.3) | 77.7 (0.3) | 38.0 (1.0) |
| | G-DRO | 79.0 (0.6) | 78.3 (0.3) | 43.7 (0.3) |
| | LC | 79.0 (0.6) | 79.0 (0.0) | 53.7 (2.3) |
| | sLA | 79.3 (0.3) | 79.0 (0.0) | 55.0 (2.1) |
| | IRM | 78.3 (0.3) | 77.7 (0.3) | 34.7 (3.2) |
| | VREx | 78.3 (0.7) | 77.7 (0.3) | 30.7 (2.2) |
| | Mixup | 79.7 (0.9) | 77.7 (0.3) | 40.0 (0.6) |
| | CRM | 83.3 (0.3) | 78.7 (0.3) | 68.0 (1.0) |
| $(0, 3)$ | ERM | 80.3 (0.3) | 79.0 (0.0) | 64.3 (2.0) |
| | G-DRO | 80.0 (0.6) | 79.0 (0.0) | 67.3 (2.7) |
| | LC | 81.3 (0.3) | 79.0 (0.0) | 69.0 (1.2) |
| | sLA | 80.7 (0.7) | 79.0 (0.0) | 66.7 (2.7) |
| | IRM | 79.7 (0.7) | 79.0 (0.0) | 65.7 (2.3) |
| | VREx | 77.7 (0.3) | 79.0 (0.0) | 64.7 (0.9) |
| | Mixup | 78.7 (0.9) | 78.7 (0.3) | 62.7 (2.8) |
| | CRM | 83.7 (0.3) | 78.7 (0.3) | 69.7 (0.3) |
| $(0, 4)$ | ERM | 78.0 (0.0) | 77.7 (0.3) | 38.0 (0.6) |
| | G-DRO | 78.7 (0.9) | 78.7 (0.3) | 52.0 (3.2) |
| | LC | 79.0 (0.0) | 79.0 (0.0) | 60.7 (1.5) |
| | sLA | 78.3 (0.3) | 79.0 (0.0) | 62.0 (1.0) |
| | IRM | 76.7 (1.3) | 77.7 (0.3) | 33.0 (2.0) |
| | VREx | 77.0 (0.6) | 78.3 (0.3) | 41.7 (1.2) |
| | Mixup | 77.7 (0.7) | 77.7 (0.3) | 37.0 (4.4) |
| | CRM | 83.7 (0.3) | 79.0 (0.0) | 69.7 (1.9) |
| $(0, 5)$ | ERM | 80.0 (0.0) | 79.0 (0.0) | 61.0 (0.6) |
| | G-DRO | 80.0 (0.6) | 79.0 (0.0) | 67.3 (1.8) |
| | LC | 79.3 (0.9) | 79.0 (0.0) | 65.7 (2.3) |
| | sLA | 80.0 (0.0) | 79.7 (0.3) | 66.7 (0.3) |
| | IRM | 78.3 (0.3) | 78.3 (0.3) | 59.7 (0.9) |
| | VREx | 78.7 (0.3) | 78.7 (0.3) | 59.0 (0.6) |
| | Mixup | 79.3 (0.7) | 78.7 (0.3) | 59.3 (3.4) |
| | CRM | 84.0 (0.0) | 78.7 (0.3) | 71.0 (1.0) |
| $(0, 6)$ | ERM | 78.7 (0.3) | 78.0 (0.0) | 36.3 (1.2) |
| | G-DRO | 78.3 (0.3) | 78.3 (0.3) | 46.3 (1.2) |
| | LC | 80.7 (0.3) | 79.0 (0.0) | 58.7 (2.3) |
| | sLA | 79.7 (0.9) | 79.0 (0.0) | 57.0 (3.1) |
| | IRM | 78.0 (0.6) | 77.0 (0.0) | 28.3 (1.2) |
| | VREx | 78.7 (0.3) | 78.0 (0.0) | 37.3 (2.2) |
| | Mixup | 78.7 (0.3) | 78.0 (0.0) | 33.7 (0.3) |
| | CRM | 83.3 (0.7) | 78.7 (0.3) | 70.0 (1.0) |
| $(0, 7)$ | ERM | 79.0 (0.0) | 77.7 (0.3) | 40.0 (1.2) |
| | G-DRO | 77.7 (0.3) | 78.7 (0.3) | 49.7 (0.3) |
| | LC | 79.7 (0.3) | 79.0 (0.0) | 60.0 (2.3) |
| | sLA | 78.7 (0.3) | 79.0 (0.0) | 56.3 (1.3) |
| | IRM | 77.0 (0.6) | 77.3 (0.3) | 33.0 (2.0) |
| | VREx | 77.0 (1.5) | 78.0 (0.0) | 39.3 (4.5) |
| | Mixup | 77.7 (1.2) | 77.7 (0.3) | 40.7 (3.5) |
| | CRM | 83.3 (0.3) | 78.3 (0.3) | 64.0 (1.2) |

*Table 10.* Results for the various compositional shift scenarios for the **CivilComments** benchmark (**Part 1**). For each metric, report the mean (standard error) over 3 random seeds on the test dataset. We highlight the worst case compositional shift scenario for each method, i.e, the scenario with the lowest worst group accuracy amongst all the compositional shift scenarios. CRM outperforms all the baselines in the respective worst case compositional shift scenarios.

| Discarded Group $(y, a)$ | Method | Average Acc | Balanced Acc | Worst Group Acc |
|---|---|---|---|---|
| (1, 0) | ERM | 81.3 (0.3) | 79.0 (0.0) | 60.3 (0.3) |
| | G-DRO | 82.3 (0.7) | 79.0 (0.0) | 69.7 (1.3) |
| | LC | 81.3 (0.3) | 79.0 (0.0) | 71.0 (0.6) |
| | sLA | 81.3 (0.9) | 79.0 (0.0) | 70.0 (1.2) |
| | IRM | 81.7 (0.3) | 78.3 (0.3) | 64.0 (0.6) |
| | VREx | 82.0 (0.0) | 79.0 (0.0) | 68.3 (1.2) |
| | Mixup | 81.3 (0.3) | 78.0 (0.0) | 63.3 (1.2) |
| | CRM | 84.0 (0.0) | 78.0 (0.0) | 68.3 (0.9) |
| (1, 1) | ERM | 81.7 (0.3) | 77.7 (0.3) | 60.3 (1.2) |
| | G-DRO | 82.0 (0.6) | 79.0 (0.0) | 67.3 (0.9) |
| | LC | 80.7 (0.3) | 79.0 (0.0) | 69.3 (0.9) |
| | sLA | 81.3 (0.3) | 79.0 (0.0) | 71.0 (1.2) |
| | IRM | 81.7 (0.3) | 78.0 (0.0) | 58.0 (1.0) |
| | VREx | 82.3 (0.7) | 79.0 (0.0) | 67.3 (1.2) |
| | Mixup | 82.3 (0.7) | 78.0 (0.0) | 58.3 (0.3) |
| | CRM | 84.0 (0.0) | 78.3 (0.3) | 70.0 (0.6) |
| (1, 2) | ERM | 81.3 (0.3) | 78.7 (0.3) | 61.3 (0.7) |
| | G-DRO | 80.7 (0.3) | 79.0 (0.0) | 63.7 (2.4) |
| | LC | 82.0 (0.6) | 79.0 (0.0) | 70.0 (2.1) |
| | sLA | 82.0 (0.6) | 79.0 (0.0) | 69.7 (1.8) |
| | IRM | 81.3 (0.3) | 78.0 (0.0) | 56.3 (0.9) |
| | VREx | 81.3 (0.3) | 79.0 (0.0) | 60.0 (2.6) |
| | Mixup | 81.7 (0.3) | 78.7 (0.3) | 62.3 (1.3) |
| | CRM | 83.7 (0.3) | 78.3 (0.3) | 63.7 (3.2) |
| (1, 3) | ERM | 82.3 (0.9) | 78.0 (0.0) | 59.0 (1.5) |
| | G-DRO | 81.0 (0.6) | 79.0 (0.0) | 67.3 (2.6) |
| | LC | 82.0 (0.0) | 79.0 (0.0) | 70.0 (1.5) |
| | sLA | 82.7 (0.9) | 79.3 (0.3) | 69.0 (1.5) |
| | IRM | 82.0 (1.0) | 78.0 (0.0) | 58.7 (1.5) |
| | VREx | 81.3 (0.3) | 79.0 (0.0) | 66.7 (0.3) |
| | Mixup | 82.0 (0.6) | 78.0 (0.0) | 57.3 (1.2) |
| | CRM | 83.7 (0.3) | 78.0 (0.0) | 71.0 (1.5) |
| (1, 4) | ERM | 82.3 (0.3) | 78.7 (0.3) | 58.3 (1.8) |
| | G-DRO | 80.3 (0.3) | 79.0 (0.0) | 68.0 (0.6) |
| | LC | 82.0 (0.0) | 79.3 (0.3) | 70.7 (0.3) |
| | sLA | 82.0 (0.6) | 79.3 (0.3) | 70.0 (0.6) |
| | IRM | 81.0 (0.0) | 78.0 (0.0) | 54.3 (0.9) |
| | VREx | 81.0 (0.6) | 79.0 (0.0) | 59.7 (0.3) |
| | Mixup | 82.0 (0.6) | 78.0 (0.0) | 57.3 (1.2) |
| | CRM | 83.7 (0.3) | 78.3 (0.3) | 60.0 (1.5) |
| (1, 5) | ERM | 82.0 (0.0) | 78.7 (0.3) | 63.7 (0.3) |
| | G-DRO | 81.7 (0.3) | 79.0 (0.0) | 64.7 (1.3) |
| | LC | 81.3 (0.3) | 79.3 (0.3) | 68.3 (0.7) |
| | sLA | 82.0 (0.6) | 79.0 (0.0) | 71.3 (0.9) |
| | IRM | 81.7 (0.3) | 78.0 (0.0) | 62.0 (1.2) |
| | VREx | 81.3 (0.3) | 79.0 (0.0) | 65.0 (2.1) |
| | Mixup | 82.0 (1.0) | 78.3 (0.3) | 61.7 (0.3) |
| | CRM | 83.7 (0.3) | 78.3 (0.3) | 70.0 (1.0) |
| (1, 6) | ERM | 82.0 (0.6) | 79.0 (0.0) | 65.3 (2.4) |
| | G-DRO | 81.0 (0.0) | 79.3 (0.3) | 66.0 (1.2) |
| | LC | 81.7 (0.7) | 79.0 (0.0) | 69.7 (2.3) |
| | sLA | 80.7 (0.3) | 79.0 (0.0) | 66.7 (0.3) |
| | IRM | 81.3 (0.3) | 79.0 (0.0) | 63.3 (0.7) |
| | VREx | 82.0 (0.0) | 79.0 (0.0) | 65.3 (1.9) |
| | Mixup | 81.3 (0.9) | 79.0 (0.0) | 64.7 (1.7) |
| | CRM | 84.0 (0.0) | 78.3 (0.3) | 70.0 (1.5) |
| (1, 7) | ERM | 82.0 (1.2) | 78.7 (0.3) | 63.3 (1.8) |
| | G-DRO | 81.0 (0.0) | 79.0 (0.0) | 64.3 (0.3) |
| | LC | 81.7 (0.3) | 79.0 (0.0) | 66.0 (1.5) |
| | sLA | 82.3 (0.3) | 79.0 (0.0) | 67.0 (1.5) |
| | IRM | 81.3 (0.3) | 78.3 (0.3) | 61.7 (0.7) |
| | VREx | 81.0 (0.6) | 79.0 (0.0) | 63.3 (1.5) |
| | Mixup | 82.3 (0.3) | 79.0 (0.0) | 66.3 (1.2) |
| | CRM | 84.3 (0.3) | 77.0 (0.0) | 63.0 (1.2) |

*Table 11.* Results for the various compositional shift scenarios for the **CivilComments** benchmark (**Part 2**). For each metric, report the mean (standard error) over 3 random seeds on the test dataset. We highlight the worst case compositional shift scenario for each method, i.e, the scenario with the lowest worst group accuracy amongst all the compositional shift scenarios. CRM outperforms all the baselines in the respective worst case compositional shift scenarios.

## G.3. CelebA Multiple Suprious Attributes

We augment the CelebA dataset (Liu et al., 2015) with three more binary spurious attribute $(a_2, a_3, a_4)$, described as follows:

- $a_2$: Determines whether the person is wearing eyeglasses or not

- $a_3$: Determines whether the person is wearing hat or not

- $a_4$: Determines whether the person is wearing earring or not

Hence, we have a total of $2^5 = 32$ groups with three binary attributes $(y, a_1, a_2, a_3, a_4)$; with $y$ denoting blond hair and $a_1$ denoting the gender, same as in our prior experiments with CelebA. Since CRM models each attribute with a different energy component, we incorporate additional energy layer for new attributes as compared to our prior experiments with two attributes. However, all the baselines would treat the two spurious attributes $(a_1, a_2, a_3, a_4)$ as a single "meta" spurious attribute $a'$ that takes 16 possible values, and aim to predict $y$. Table 12 presents the results for the multi-attribute CelebA dataset, where we generate multiple benchmarks with compositional shift by dropping one of the 32 groups from the training & validation dataset (similar to the setup for our prior experiments). We find that CRM outperforms all the baselines w.r.t worst group accuracy and balanced accuracy, hence, remains superior for the case of multiple attributes as well.

| Method | Average Acc | Balanced Acc | Worst Group Acc | Worst Group Acc (No Groups Dropped) |
|--------|-------------|--------------|-----------------|-------------------------------------|
| ERM    | 94.8 (0.1)  | 83.5 (0.1)   | 0.9 (0.5)       | 0.0 (0.0)   |
| G-DRO  | 93.2 (0.0)  | 89.1 (0.0)   | 11.2 (1.5)      | 0.0 (0.0)   |
| LC     | 93.1 (0.1)  | 91.7 (0.3)   | 34.3 (5.9)      | 52.0 (26.1) |
| sLA    | 93.4 (0.1)  | 91.7 (0.0)   | 33.0 (1.3)      | 26.3 (26.3) |
| IRM    | 90.2 (0.6)  | 85.6 (0.2)   | 26.6 (0.4)      | 40.0 (2.0)  |
| VREx   | 90.0 (0.3)  | 89.1 (0.1)   | 10.6 (1.3)      | 25.0 (25.0) |
| Mixup  | 94.9 (0.0)  | 87.0 (0.0)   | 2.7 (0.5)       | 14.7 (14.7) |
| CRM    | 91.0 (0.1)  | 90.1 (0.1)   | 63.2 (2.6)      | 73.7 (2.9)  |

*Table 12.* **CelebA with Multiple Spurious Attributes.** We compare CRM to baselines on CelebA dataset with 5 attributes. Similar to the prior setup (Table 1), we report the Average Accuracy, Group Balanced Accuracy, and Worst Group Accuracy (WGA), averaged as a group is dropped from the training and validation sets. Last column is WGA under the standard subpopulation shift scenario where no groups were dropped. CRM is the best approach w.r.t the worst group accuracy.

## G.4. Results for Ablations with CRM

In the implementation of CRM in Algorithm C, we have the following two choices; 1) we use the extrapolated bias $B^\star$ (equation 11); 2) we set $\hat{q}(z)$ as the uniform distribution, i.e, $\hat{q}(z = (y, a)) = \frac{1}{d_y \times d_a}$. We now conduct ablation studies by varying these components as follows.

- *Bias $B^\star$ + Emp Prior:* We still use the extrapolated bias $B^\star$ but instead of uniform $\hat{q}(z)$, we use test dataset to obtain the counts of each group, denoted as the empirical prior. Note that this approach assumes the knowledge of test distribution of groups, hence we expect this to improve the average accuracy but not the necessarily the worst group accuracy.

- *Bias $\hat{B}$ + Unf Prior:* We still use the uniform prior for $\hat{q}(z)$ but instead of the extrapolated bias $B^\star$, we use the learned bias $\hat{B}$ (equation 9). This ablation helps us to understand whether extrapolated bias $B^\star$ are crucial for CRM to generalize to compositional shifts.

- *Bias $\hat{B}$ + Emp Prior:* Here we change both aspects of CRM as we use the learned bias $\hat{B}$ and empirical prior from the test dataset for $\hat{q}(z)$.

Table 13 presents the results of the ablation study. We find that extrapolated bias is crucial for CRM as the worst group accuracy with learned bias is much worse! Further, using empirical prior instead of the uniform prior leads to improvement in average accuracy at the cost of worst group accuracy.

| Dataset | Ablation | Average Acc | Balanced Acc | Worst Group Acc |
|---|---|---|---|---|
| Waterbirds | CRM | 87.1 (0.7) | 87.8 (0.1) | 78.7 (1.6) |
| | Bias $B^\star$ + Emp Prior | 91.6 (0.2) | 87.4 (0.3) | 75.2 (1.3) |
| | Bias $\hat{B}$ + Unf Prior | 81.2 (0.6) | 82.7 (0.2) | 55.7 (1.0) |
| | Bias $\hat{B}$ + Emp Prior | 84.3 (0.6) | 81.6 (0.3) | 51.3 (1.0) |
| CelebA | CRM | 91.1 (0.2) | 89.2 (0.3) | 81.8 (1.2) |
| | Bias $B^\star$ + Emp Prior | 94.3 (0.1) | 75.8 (0.4) | 34.1 (1.0) |
| | Bias $\hat{B}$ + Unf Prior | 83.6 (0.1) | 84.7 (0.2) | 58.9 (0.4) |
| | Bias $\hat{B}$ + Emp Prior | 90.9 (0.1) | 77.2 (0.3) | 35.4 (0.7) |
| MetaShift | CRM | 87.6 (0.2) | 84.7 (0.1) | 73.4 (0.7) |
| | Bias $B^\star$ + Emp Prior | 89.2 (0.2) | 84.0 (0.4) | 65.1 (1.4) |
| | Bias $\hat{B}$ + Unf Prior | 87.2 (0.3) | 82.9 (0.4) | 58.7 (0.6) |
| | Bias $\hat{B}$ + Emp Prior | 88.1 (0.1) | 82.1 (0.1) | 56.1 (0.4) |
| MultiNLI | CRM | 74.3 (0.3) | 76.1 (0.3) | 58.7 (1.4) |
| | Bias $B^\star$ + Emp Prior | 74.7 (0.3) | 72.3 (0.4) | 41.4 (1.5) |
| | Bias $\hat{B}$ + Unf Prior | 72.5 (0.6) | 74.0 (0.4) | 30.4 (2.6) |
| | Bias $\hat{B}$ + Emp Prior | 73.2 (0.5) | 70.8 (0.1) | 22.2 (0.9) |
| CivilComments | CRM | 83.7 (0.1) | 78.4 (0.0) | 67.9 (0.5) |
| | Bias $B^\star$ + Emp Prior | 87.0 (0.1) | 74.0 (0.2) | 48.1 (0.6) |
| | Bias $\hat{B}$ + Unf Prior | 76.9 (0.3) | 77.8 (0.1) | 52.4 (0.7) |
| | Bias $\hat{B}$ + Emp Prior | 83.5 (0.2) | 77.9 (0.1) | 62.3 (0.8) |
| NICO++ | CRM | 84.7 (0.3) | 84.7 (0.3) | 40.3 (4.3) |
| | Bias $B^\star$ + Emp Prior | 85.0 (0.0) | 85.0 (0.0) | 41.0 (4.9) |
| | Bias $\hat{B}$ + Unf Prior | 85.0 (0.0) | 85.0 (0.0) | 31.0 (1.0) |
| | Bias $\hat{B}$ + Emp Prior | 85.0 (0.0) | 85.0 (0.0) | 27.7 (3.9) |

*Table 13.* **Ablation study with CRM.** We consider the average performance over the different compositional shift scenarios for each benchmark, and report the mean (standard error) over 3 random seeds on the test dataset. CRM corresponds to the usual implementation with extrapolated bias $B^\star$ and uniform prior for $\hat{q}(z)$. CRM obtains better worst group accuracy than all the ablations, highlighting the importance of both extrapolated bias and uniform prior! Extrapolated bias is critical for generalization to compositional shifts as the performance with learned bias is much worse.

## G.5. Results for the Original Benchmarks

We present results for the original benchmarks $(\mathcal{D}_{\text{train}}, \mathcal{D}_{\text{val}}, \mathcal{D}_{\text{train}})$ in Table 14, which corresponds to the standard subpopulation shift case for these benchmarks. For Waterbirds, CelebA, MetaShift, and MultiNLI, subpopulation shift implies all the groups $z = (y, a)$ are present in both the train and test dataset ($\mathcal{Z}^{\text{train}} = \mathcal{Z}^{\text{test}} = \mathcal{Z}^\times$), however, the groups sizes change from train to test, inducing a spurious correaltion between class labels $y$ and attributes $a$. For the NICO++ dataset, we have a total of 360 groups in the test dataset but only 337 of them are present in the train dataset. But still this is not a compositional shift as the validation dataset contains all the 360 groups. We find that CRM is still competitive to the baselines for the standard subpopulation shift scenario of each benchmark!

| Dataset | Method | Average Acc | Balanced Acc | Worst Group Acc |
|---|---|---|---|---|
| Waterbirds | ERM | 87.3 (0.3) | 84.0 (0.0) | 62.3 (1.2) |
| | G-DRO | 91.7 (0.3) | 91.0 (0.0) | 87.3 (0.3) |
| | LC | 92.0 (0.0) | 91.0 (0.0) | 88.7 (0.3) |
| | sLA | 92.3 (0.3) | 91.0 (0.0) | 89.7 (0.3) |
| | IRM | 87.3 (0.3) | 86.0 (0.0) | 72.3 (1.2) |
| | VREx | 92.0 (0.0) | 90.7 (0.3) | 84.3 (0.7) |
| | Mixup | 85.0 (0.0) | 86.0 (0.0) | 69.7 (0.9) |
| | CRM | 91.3 (0.9) | 91.0 (0.0) | 86.0 (0.6) |
| CelebA | ERM | 95.7 (0.3) | 84.0 (0.0) | 52.0 (1.0) |
| | G-DRO | 92.0 (0.6) | 93.0 (0.0) | 91.0 (0.6) |
| | LC | 92.0 (0.6) | 92.0 (0.0) | 90.0 (0.6) |
| | sLA | 92.3 (0.3) | 91.7 (0.3) | 86.7 (1.9) |
| | IRM | 87.0 (2.5) | 85.3 (1.2) | 67.7 (3.5) |
| | VREx | 92.0 (0.0) | 92.0 (0.0) | 89.0 (0.6) |
| | Mixup | 95.0 (0.0) | 86.7 (0.3) | 62.0 (1.0) |
| | CRM | 93.0 (0.0) | 92.0 (0.0) | 89.0 (0.6) |
| MetaShift | ERM | 90.0 (0.0) | 84.0 (0.0) | 63.0 (0.0) |
| | G-DRO | 90.3 (0.3) | 88.3 (0.3) | 80.7 (1.3) |
| | LC | 89.7 (0.3) | 87.7 (0.3) | 80.0 (1.2) |
| | sLA | 90.0 (0.6) | 87.7 (0.3) | 80.0 (1.2) |
| | IRM | 90.7 (0.3) | 85.3 (0.9) | 69.3 (2.4) |
| | VREx | 90.0 (0.0) | 86.7 (0.3) | 75.3 (2.2) |
| | Mixup | 90.7 (0.3) | 85.3 (0.7) | 68.3 (2.7) |
| | CRM | 88.3 (0.7) | 85.7 (0.3) | 74.7 (1.5) |
| MultiNLI | ERM | 81.7 (0.3) | 80.7 (0.3) | 68.0 (1.7) |
| | G-DRO | 80.7 (0.3) | 78.0 (0.0) | 57.0 (2.3) |
| | LC | 82.0 (0.0) | 82.0 (0.0) | 74.3 (1.2) |
| | sLA | 82.0 (0.0) | 82.0 (0.0) | 71.7 (0.3) |
| | IRM | 75.7 (0.3) | 74.3 (0.3) | 54.3 (2.4) |
| | VREx | 79.0 (0.0) | 79.0 (0.0) | 69.7 (0.3) |
| | Mixup | 81.3 (0.3) | 80.0 (0.0) | 63.7 (2.9) |
| | CRM | 81.7 (0.3) | 81.7 (0.3) | 74.7 (1.3) |
| CivilComments | ERM | 80.3 (0.3) | 79.0 (0.0) | 61.0 (2.5) |
| | G-DRO | 79.7 (0.3) | 79.0 (0.0) | 64.7 (1.5) |
| | LC | 80.7 (0.3) | 79.7 (0.3) | 67.3 (0.3) |
| | sLA | 80.3 (0.3) | 79.0 (0.0) | 66.3 (0.9) |
| | IRM | 80.3 (0.7) | 79.0 (0.0) | 60.3 (1.5) |
| | VREx | 80.3 (0.7) | 79.0 (0.0) | 63.3 (1.5) |
| | Mixup | 80.0 (0.6) | 79.0 (0.0) | 61.3 (1.5) |
| | CRM | 83.3 (0.3) | 78.0 (0.0) | 70.0 (0.6) |
| NICO++ | ERM | 85.3 (0.3) | 85.0 (0.0) | 35.3 (2.3) |
| | G-DRO | 83.7 (0.3) | 83.3 (0.3) | 33.7 (1.2) |
| | LC | 85.0 (0.0) | 85.0 (0.0) | 35.3 (2.3) |
| | sLA | 85.0 (0.0) | 85.0 (0.0) | 35.3 (2.3) |
| | IRM | 63.7 (0.3) | 62.7 (0.3) | 0.0 (0.0) |
| | VREx | 86.0 (0.0) | 86.0 (0.0) | 38.0 (5.0) |
| | Mixup | 85.0 (0.0) | 84.7 (0.3) | 33.0 (0.0) |
| | CRM | 85.0 (0.0) | 84.7 (0.3) | 39.0 (3.2) |

*Table 14.* Results for the standard subpopulation shift case for each benchmark. Here we do not transform the datasets for compositional shifts, hence all the groups are present in both the train and the test dataset (except the NICO++ benchmark). CRM is still competitive with the baselines for this scenario where no groups were discarded additionally.

## G.6. CRM's Analysis with Varying Group Size

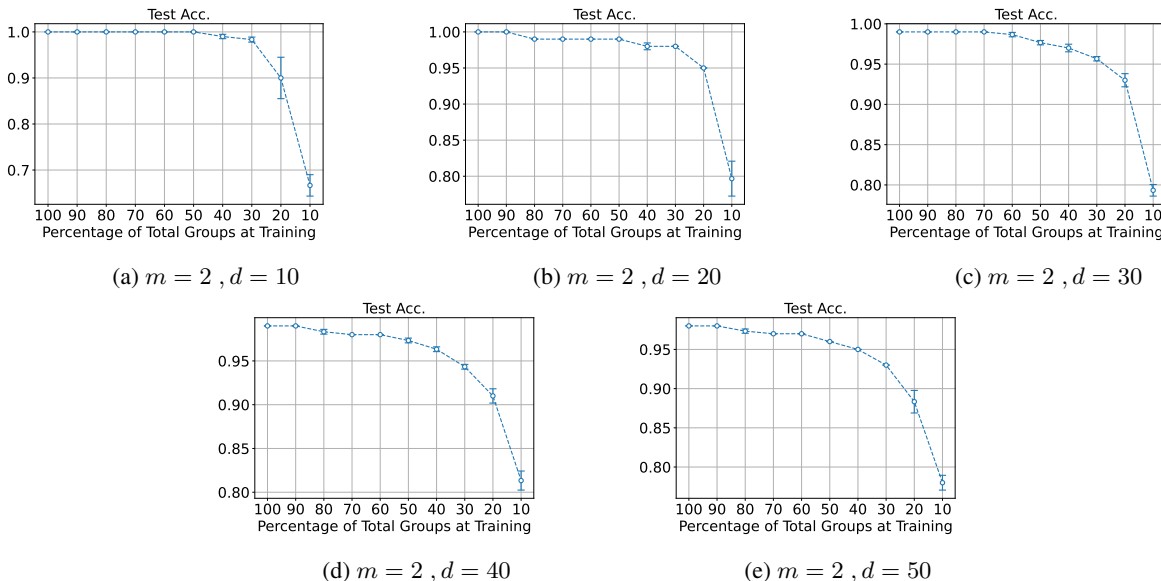

(a) $m = 2$ , $d = 10$      (b) $m = 2$ , $d = 20$      (c) $m = 2$ , $d = 30$

(d) $m = 2$ , $d = 40$      (e) $m = 2$ , $d = 50$

*Figure 8.* **Varying Group Size Analysis** ($m = 2$ attributes). We analyze the rate of growth of total groups required to achieve cartesian-product extrapolation. For each scenario, we evaluate CRM's generalization capabilities as we discard more groups from the training dataset. X-axis denotes the percentage of total groups available for training, and y-axis denotes the test average accuracy (mean & standard error over 3 random seeds) obtained by CRM. We find that observing at least $20\%$ of total train groups is sufficient for good generalization.

**Setup.** We conduct experiments to understand the rate of growth of total groups required in order to achieve Cartesian-Product extrapolation, as we vary the total number of attributes ($m$) and the total number of categories ($d$) for each attribute. Given attributes $z = (z_1, z_2, \cdots, z_m)$, we sample data sample data from the following (additive) energy function.

$$E(x, z) = \sum_{i=1}^{m} ||x - \mu(z_i)||^2$$

where $x, \mu(z_i) \in \mathbb{R}^n$ for all $i \in \{1, \cdots, m\}$. Note that the energy function can be rewritten as follows:

$$E(x, z) = \frac{1}{2}(x - \mu(z))^T \Sigma^{-1}(x - \mu(z)) + C(z_1, z_2)$$

with $\mu(z) = \frac{1}{m}\sum_{i=1}^{m} \mu(z_i)$ and $\Sigma^{-1} = 2mI_n$. Hence, the resulting distribution is essentially a multi-variate gaussian distribution $p(x|z) = \frac{1}{\mathbb{Z}(z)} \exp\left(-E(x, z)\right) = N\left(x|\mu(z), \Sigma\right)$.

To generate data from a particular configuration $(d, m, n)$ we first sample $d * m$ orthogonal vectors to get mean vectors for the different realizations of each attribute, i.e, $\{\mu(z_i = k) \mid i \in [1, m] \ \& \ k \in [1, d]\}$. Then we sample $x$ from the resulting normal distribution $x \sim N\left(\mu(z), \Sigma\right)$ to create a dataset with uniform support over all the $d^m$ groups. We fix the data dimension as $n = 100$ and have the following two setups.

- $m = 2$ Attribute Case. We fix $m = 2$ and vary $d$ in the following range, $[10, 20, 30, 40, 50]$. This results in groups with sizes $[100, 400, 900, 1600, 2500]$.

- $m > 2$ Attribute Case. We fix $d = 2$ and vary $m$ in the following range, $[7, 8, 9]$. This results in groups with sizes $[128, 256, 512]$.

For both these setups, we analyze how the performance of CRM degrades as we discard more groups from the training dataset. Note that the test dataset contains samples from all the groups and there are no group imbalances. Hence, average accuracy in itself is a good indicator of generalization performance.

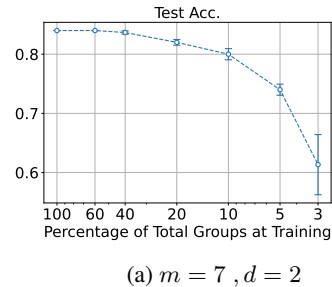 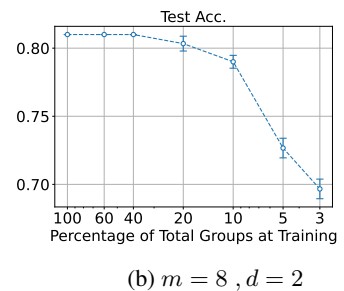 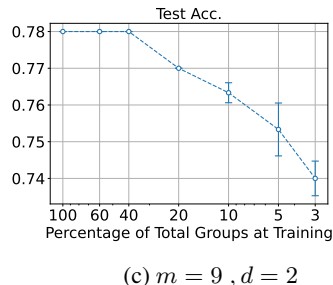

(a) $m = 7$ , $d = 2$          (b) $m = 8$ , $d = 2$          (c) $m = 9$ , $d = 2$

*Figure 9.* **Varying Group Size Analysis** ($m > 2$ attributes). We analyze the rate of growth of total groups required to achieve cartesian-product extrapolation. For each scenario, we evaluate CRM's generalization capabilities as we discard more groups from the training dataset. X-axis (log scale) denotes the percentage of total groups available for training, and y-axis denotes the test average accuracy (mean & standard error over 3 random seeds) obtained by CRM. We find that observing at least $10\%$ of total train groups is sufficient for good generalization.

**Results.** Figure 8 and Figure 9 presents the results for the $m = 2$ and $m > 2$ attribute case respectively. For $m = 2$ attribute case, we find that CRM trained with $20\%$ of the total groups $(0.2d^2)$ still shows good generalization for predicting $z_1$ ($\sim 90\%$ test accuracy), and the drop in test accuracy as compared to the oracle case of no groups dropped is within $10\%$. Similarly, for $m > 2$ attribute case, we find that CRM trained with $10\%$ of the total groups $(0.1 \times 2^m)$ still shows good generalization for predicting $z_1$, and the drop in test accuracy as compared to the oracle case is within $10\%$.

