# OpenReview forum: "Compositional Risk Minimization"
_ICML.cc/2025/Conference — ICML 2025 poster_

### Official Review · Reviewer_C2eF · 2025-02-17

**Overall Recommendation:** 4

**Summary:**

This paper addresses compositional generalization by tackling compositional shift, where test data contains unseen combinations of attributes. The authors propose Compositional Risk Minimization (CRM), using additive energy distributions to model attributes and providing an alternative to empirical risk minimization. Their approach involves training an additive energy classifier and adjusting it for compositional shifts, with theoretical analysis showing extrapolation capabilities to affine hulls of seen attribute combinations. Experimental results on benchmark datasets demonstrate improved robustness compared to existing methods for handling subpopulation shifts.

**Claims And Evidence:**

The claims are supported by both theoretical and empirical evidence.

**Essential References Not Discussed:**

None

**Experimental Designs Or Analyses:**

Based on my review of the experimental section, the empirical results support the paper's claims.

**Methods And Evaluation Criteria:**

Yes, the evalucation criteria (Average Accuracy and Worst Group Accuracy) makes sense.

**Other Comments Or Suggestions:**

The authors use 'Compositional Risk Minimization' as the title. Could you explain the core concept of Compositional Risk Minimization using simple mathematical formulations? Please refer to the classical mathematical formulation of Empirical Risk Minimization (ERM).

**Other Strengths And Weaknesses:**

Strengths:

1. The research tackles an important problem in compositional generalization with novel insights.
2. Clear implementation details and reproducibility through provided pseudocode.

Weakness:

Limited empirical comparisons against existing baseline methods

**Questions For Authors:**

No

**Relation To Broader Scientific Literature:**

The authors provide a thorough and systematically structured overview of previous work in this research area.

**Theoretical Claims:**

The mathematical derivations and theoretical arguments seem rigorous upon initial examination.

---

> ### Author Rebuttal · Authors · 2025-03-31
>
> We thank the reviewer for their positive and insightful feedback! We now address the concerns raised by the reviewer ahead.
>
> **1. Limited empirical comparisons against existing baseline methods**
>
> We have done extensive benchmarking of CRM with 6 widely used baselines in the literature of subpopulation shifts apart from ERM. Please check Table 5 in Appendix G.1, where we compare CRM with GroupDRO, LC, sLA, IRM, VREx, and Mixup. We find that CRM outperforms all the baselines across diverse benchmarks w.r.t. the worst group accuracy.
>
> Note that in the main body of the paper (Table 1), due to space constraints where we only compared with the best performing baselines (GroupDRO, LC, sLA). Also, note that our primary comparison baseline has been with GroupDRO, as it is the most effective method for addressing subpopulation shifts, hence it was the main focus of our work. We also included baselines like LA (Logit Adjustment) that share conceptual similarities with our approach.
>
> **2. The authors use 'Compositional Risk Minimization' as the title. Could you explain the core concept of Compositional Risk Minimization using simple mathematical formulations? Please refer to the classical mathematical formulation of Empirical Risk Minimization (ERM).**
>
> The classical ERM objective can be stated as follows.
>
> $$ R_p(f) = \sum_{z \in \mathcal{Z}} p(z)R(f|z)$$
>
> where $R(f|z) = \mathbb{E}_{p(x|z)}[\ell(f(X),Y)|z]$ and $p(z)$ is the training prior probability and $\mathcal{Z}$ is the set of all $d^m$ groups. If $\ell$ is the cross-entropy loss, then output of ERM (with no capacity constraints) matches the true $p(z|x)$. Also, note that in the above summation $p(z)$ is zero on all groups that are not in the support of the training distribution.
>
> To tackle compositional distribution shifts, we want to learn predictors that minimize
>
> $$ R_q(f) = \sum_{z \in \mathcal{Z}} q(z)R(f|z)$$
>
> and not $R_p(f)$, which ERM minimizes. In the above objective $q(z)$ can be non-zero on groups $z$ that have zero probability under $p(z)$. Our results show that our approach outputs a predictor that provably minimizes risk under any compositional shift $R_q(f)$ and hence the name *compositional risk minimization*.
>
> Specifically, in Theorem 3, we showed that CRM outputs the Bayes optimal predictor and hence it provably minimizes $R_q(f)$ with a high probability, where $\ell$ is cross-entropy loss or $0/1$ loss, as long as the number of training groups grow as $O(md + dlog d)$.
>
> To clarify, our approach does not require us to explicitly compute the risk on the test distribution $R_q(f)$. For additive energy distributions, in the second step of CRM we adapt the $\hat{p}(z|x)$ to $\hat{q}(z|x)$ with the extrapolated bias $B^{\star}$, which equals the true test predictor $q(z|x)$ (Theorem 2). Hence, CRM avoids the computation of $R_q(f)$.
>
> Thanks again for this interesting question! We will add this discussion to the paper as well. We are very open to further discussion and would be happy to address any remaining concerns.

---

### Official Review · Reviewer_TU2X · 2025-02-27

**Overall Recommendation:** 4

**Summary:**

This paper proposes compositional risk minimization (CRM), an approach to compositional generalization that is based on additive energy distributions. The intuition is to train an energy-based classifer on the training set, then modify it to account for known bias between the observed training and test distributions. The authors show a number of theoretical results as well as empirical results on benchmarks for subpopulation shifts.

**Claims And Evidence:**

Yes, all claims are supported with adequate theoretical and empirical evidence.

**Essential References Not Discussed:**

No missing references as far as I am aware.

**Experimental Designs Or Analyses:**

The experimental setup described in section 5.1 appears sound.

**Methods And Evaluation Criteria:**

The method is well constructed and the benchmarks are well chosen.

**Other Comments Or Suggestions:**

Typos

- "boradcasting" in Figure 1 caption

**Other Strengths And Weaknesses:**

The paper is quite well written and presented. Experiments are conducted over a number of datasets. Overall, the authors present an interesting, fresh approach to compositional generalization.

My main concern is whether the additive energy distribution assumption is realistic (beyond the particular subpopulation-shift setting considered in the experiments). It would be great to have additional discussion on this point.

**Questions For Authors:**

What practical settings is the additive energy distribution assumption applicable to? For example, does it apply to the blue elephant on the Moon example laid out in the introduction? It's reasonable if it does not hold, but discussing the boundaries of when it holds would be good to include.

**Relation To Broader Scientific Literature:**

This work is related to prior work on compositionality with energy-based models. However, these works typically consider a generative setting; by contrast, the authors consider a discriminative setting and produce a novel set of theoretical results.

**Theoretical Claims:**

The proofs appear correct; no issues found.

---

> ### Author Rebuttal · Authors · 2025-03-31
>
> We thank the reviewer for their positive and insightful feedback! We will fix the typo in the caption of Figure 1, thanks for pointing this. We now address the concerns raised by the reviewer ahead.
>
> **My main concern is whether the additive energy distribution assumption is realistic (beyond the particular subpopulation-shift setting considered in the experiments). It would be great to have additional discussion on this point. What practical settings is the additive energy distribution assumption applicable to? For example, does it apply to the blue elephant on the Moon example laid out in the introduction? It's reasonable if it does not hold, but discussing the boundaries of when it holds would be good to include.**
>
> We believe that additive energy assumption is practical for settings where the image is aptly described by an AND operation among the attributes (and this applies to the blue elephant example). The summation of additive energies leads to a product of exponentials which act as a soft AND operation. Hence, each energy term contributes to checking one of the conditions in the AND operation.
>
> Regarding the specific image "blue elephant on the moon", let us start with a simplification of this image, "elephant on the moon". Then we have one energy term that detects elephant and the other energy term detects moon. However, let us now consider the original image "blue elephant on the moon". If we do a simple AND between detecting blue color, elephant, and the moon, then even the image "elephant on blue moon" will have the same energy (density) as the image "blue elephant on the moon", which is not desirable.
>
> But we can model this scenario using additive energy distribution by having energy components for each object-specific attribute. Hence, the final energy function becomes $E(x|o, z)= \sum_{o_i} \sum_{z_{ij}}  E_{ij}(x, o_i, z_{ij})$, where $o_i$ refers to the location of the $i^{th}$ object and $z_{ij}$ refers to the $j^{th}$ attribute for the $i^{th}$ object. This essentially allows us to "bind" the attribute information to an object and still model the overall distribution with additive energies. In the example above, we have one energy term for the object elephant with attribute blue, which gets added with the energy term for the object moon (with some default attribue value).
>
> Thanks again for this interesting question! A thorough investigation of this is a fruitful future direction. We are very open to further discussion and would be happy to address any remaining concerns.

---

### Official Review · Reviewer_ypJd · 2025-03-16

**Overall Recommendation:** 3

**Summary:**

This paper introduces a method for addressing compositional shifts in discriminative tasks. The authors propose a theoretical framework built on additive energy distributions, where each energy term represents an attribute. They introduce the discrete affine hull concept to characterize extrapolation capabilities. Their two-step algorithm first trains an additive energy classifier to predict attributes jointly, then adjusts this classifier for compositional shifts. Theoretical guarantees show that the proposed method can extrapolate to test distributions within the discrete affine hull of training distributions. Experiments on several benchmarks demonstrate the effectiveness of the proposed method.

Pros:
+ The proposed method is well-motivated and reasonable, and builds on additive energy distributions that are studied in generative compositionality.
+ The proposed algorithm is practical and easy to implement. And the authors provide detailed implementation in the appendix.
+ The extensive empirical evaluation demonstrates consistent improvements across diverse benchmarks.

Cons:
- The additive energy assumption may be too limited for many real-world situations where different factors interact in complex ways rather than simply adding together. This could reduce how useful the approach is in practice.
- The additive energy distributions were previously studied in generative compositionality, while the authors extend this framework to discrimination tasks. This is an incremental contribution and the novelty appears limited in scope.
- The paper assumes access to attribute labels during training, which might not always be available in practice.

**Claims And Evidence:**

1. The authors provide rigorous theoretical analysis with detailed proofs showing that the proposed method can generalize to novel attribute combinations within the discrete affine hull.

2. Empirical validation on several common benchmarks show the proposed method outperforms other baselines.

**Essential References Not Discussed:**

The idea that compositional generalization can only be achieved within the discrete affine hull is analogous to the assumption in the following papers, which posit that the test distribution should lie within the convex hull of the training distributions:
[1] Qiao, F., & Peng, X. Topology-aware Robust Optimization for Out-of-Distribution Generalization. ICLR 2023.
[2] Yao, H., Yang, X., Pan, X., Liu, S., Koh, P. W., & Finn, C. Improving Domain Generalization with Domain Relations. ICLR 2024.

**Experimental Designs Or Analyses:**

The experiments systematically demonstrate the advantages of the proposed method, especially on worst-group accuracy. The ablation studies clearly show the importance of the extrapolated bias term, which aligns with the theoretical framework.

**Methods And Evaluation Criteria:**

1. The proposed two-step algorithm is consistent with the theoretical framework. The proposed method is well-motivated and reasonable, and builds on additive energy distributions that are studied in generative compositionality.

2. The authors use average accuracy, group-balanced accuracy, and worst-group accuracy to evaluate performance. The evaluation is comprehensive.

**Other Comments Or Suggestions:**

None.

**Other Strengths And Weaknesses:**

Please see summary.

**Questions For Authors:**

1. The paper assumes attribute labels are available during training. Can the proposed method be adapted to settings where attribute labels are only partially available or must be inferred?

2. How does the computational complexity of the proposed method scale with the number of attributes and classes?

**Relation To Broader Scientific Literature:**

1. This work is built on additive energy distributions for generative tasks. The authors extend it to discriminative tasks.

2. The problem connects to out-of-distribution generalization and subpopulation shifts. The authors clearly articulate how compositional generalization relates to these established research areas.

**Theoretical Claims:**

The proofs appear mathematically sound. The theoretical analysis provides a sharp characterization of extrapolation, demonstrating that generalization beyond the discrete affine hull is fundamentally impossible. This establishes clear boundaries on what can be achieved in this domain.

---

> ### Author Rebuttal · Authors · 2025-03-31
>
> We thank the reviewer for their positive and insightful feedback! We are glad they appreciate the technical soundness of our work, on both the theoretical and empirical front. We now address the concerns raised by them.
>
> > Additive Energy Distribution (AED) Limitations
>
> We emphasize that the benchmarks used to evaluate CRM are both realistic and widely adopted in the subpopulation shift literature. Since CRM consistently outperforms baselines, this suggests that AED assumption is not overly restrictive and can model realistic datasets effectively.
>
> We now clarify how AED models complex interactions between attributes. Note that AED does not imply additive interactions in data space ($x= \sum_{i} g(z_i)$) as in additive decoders (Lachapelle et al. 2024). Instead, it models the AND operation between attributes, as illustrated below via examples (check Appendix B for details).
>
> i) Consider images that contain a distinct object varying in shape, size, and color. At any pixel, it is unlikely that shape, color, and size attributes interact additively, rather their interactions are complex which can't be captured via additive decoders. However, under AED, interactions are modeled via an energy component per attribute: one detecting shape, AND another detecting color, AND a third detecting size. Together, these energy terms define the distribution of images conditioned on attributes.
>
> ii) An example from a different data modality is the the CivilComments benchmark, where the attributes toxic language (class label) and demographic identity (spurious attribute) interact non-trivially in text space. However, under AED, we can model their interactions via an energy component that checks whether the language is toxic, AND another energy component checks the demographic identity.
>
> > Novelty of the work
>
> We explain the key features that set this work apart.
>
> a) *Discrete Affine Hull, a novel mathematical object:* Existing AED works in generative compositionality lack theoretical guarantees for generalization beyond training data. To address this, we introduce a novel mathematical object, the discrete affine hull, which precisely characterizes extrapolation to new distributions for both discriminative and generative tasks. For instance, our theoretical guarantee states that it is possible to generalize from $O(m*d)$ groups to $d^m$ groups in both discriminative and generative tasks.
>
> b) *Discriminative training without estimating partition function:*  Compositionality in discriminative tasks is a major problem, and our work makes key advances. One way to learn a classifier is via generative classification (lines 171-190, right column), where we first train densities $\hat{p}(x|z)$ on observed groups, estimate new densities via affine combinations, and then use Bayes rule to derive $\hat{p}(z|x)$. While this guarantees generalization, it is impractical due to the intractable gradient estimation of log partition functions (line 193-207, right column). While CRM circumvents these issues and retains the same guarantees.
>
> > Prior work regarding convex hulls
>
> We thank the reviewer for pointing us to these references, and we are happy to cite and contrast with them. However, note that the densities from a new group $q(x|z')$ in the affine hull cannot be expressed as a convex or even an affine combination of the densities. For details, see e.q. (22) in Appendix D.2, summarized below.
>
> $\log\big(q(x|z')\big)= \sum_{z\in \mathcal{Z}^{\mathsf{train}}}\alpha_z \log p(x|z) - R\big(\{\alpha_z\}_{z\in \mathcal{Z}^{\mathsf{train}}}\big)$
>
> Thus, in our setting, only energy terms are expressed as an affine combination, and our guarantees apply to distributions that are outside the convex hull of the training distributions.
>
> > Missing attribute labels scenario
>
> We believe this is an exciting future work. Existing works such as XRM (Pezeshki et al. 2024) show how one can discover the environments and then use existing domain generalization methods that require environment labels. We believe it would be exciting to extend these works to infer spurious attributes directly in combination with our approach.
>
> > Computational complexity of CRM
>
> For CRM training stage 1 (e.q. 7), the cost for each step is similar to that of training an ERM-based classifier for predicting the group $z$, which is proportional to  $| \mathcal{Z}_{\mathsf{train}} | \times m \times d$.
>
>  In the training stage 2, we compute the extrapolated bias $B^{*}(z)$ (e.q. 11), and the cost is proportional to $| \mathcal{Z}_{\mathsf{test}} | \times \text{number of training samples} \times m \times d$.
>
> Observe that in the worst case scenario the number of classes for $z$ at test time is $d^m$, making inference cost $O(d^{m+1}m)$. Any method that predicts $z$ would have to compute a probability vector of size $d^m$ and thus spend at least $O(d^m)$ per inference.
>
> Thanks again for your constructive comments, and please let us know if there are any remaining concerns.

---

### Official Review · Reviewer_uM4Q · 2025-03-21

**Overall Recommendation:** 3

**Summary:**

This paper addresses the compositional shifts, a hard type of sub-population shifts, and proposes compositional risk minimization. The method is well-motivated and some theoretical analyses are provided. Results on the sub-population shift benchmark are shown to support the proposed method.

**Claims And Evidence:**

- The compositional risk minimization method is reasonable and well-motivated.
- The formulation of the compositional shift setting provides a foundation for further research.
- Experimental results are good to support the method.

**Essential References Not Discussed:**

None

**Experimental Designs Or Analyses:**

I understand that the paper’s chosen disjoint setting adds a level of complexity. However, in real-world scenarios, it is often feasible to obtain a small number of samples for different attribute combinations (particularly with only two attributes, as in these experiments). The proposed method should also be evaluated in traditional settings where all attribute combinations have some representation. This would confirm that the method performs well without requiring group-dropping.

**Methods And Evaluation Criteria:**

The analysis and proposed algorithm are designed to handle multiple attributes, with the theoretical advantages being most relevant for this multi-attribute context. However, the experiments are limited to only 2~3 attributes. I suggest that the authors include empirical results with multiple attributes to better align with the theoretical analysis.

**Other Comments Or Suggestions:**

I would suggest the authors moving additional results into the main body.

**Other Strengths And Weaknesses:**

N/A

**Questions For Authors:**

N/A

**Relation To Broader Scientific Literature:**

This paper proposes a more efficient method for domain generalization.

**Theoretical Claims:**

Correct

---

> ### Author Rebuttal · Authors · 2025-03-31
>
> We thank the reviewer for their positive and insightful feedback! We now address the concerns raised by the reviewer ahead.
>
> **1. The analysis and proposed algorithm are designed to handle multiple attributes, with the theoretical advantages being most relevant for this multi-attribute context. However, the experiments are limited to only 2~3 attributes. I suggest that the authors include empirical results with multiple attributes to better align with the theoretical analysis.**
>
> Thanks for raising this issue, we would like to provide some clarifications regarding this. In our theoretical results (Theorem 2 & 3), the key finding is that if we observe $O(md + dlogd)$ groups at training time, then CRM would generalize to test distributions over all the $d^m$  groups. Note that the result considers the setting with multiple groups, which can arise from multiple attributes $m$ or from multiple values per attribute $d$. Hence, the theoretical advantages are not restricted to the multi-attribute scenario.
>
> In our experiments, we already consider several scenarios that go beyond few attributes and few groups. NICO++ dataset has 360 groups, CelebA (multiple spurious attribute case, Table 12, Appendix G.3) consists of 5 attributes and a total of 32 groups. In addition to this, we also provided experiments on synthetic data with varying $d$ ($m=2$ with $d$ up to 50 leading to 2500 groups) and varying $m$ ($d=2$ and $m$ up to $9$ leading a total of 512 groups) in Figure 9, Appendix G.6. In all these experiments CRM performs well thus aligning the behavior of the method with the theoretical claims.
>
> Finally, we also want to point the guarantees in Theorem 2 are applicable to settings with small number of groups as well, i.e., in the setting of datasets like Waterbirds CRM offers a non-trivial Bayes optimality guarantee when only three out of the four groups are observed at training time.
>
> **2. I understand that the paper’s chosen disjoint setting adds a level of complexity. However, in real-world scenarios, it is often feasible to obtain a small number of samples for different attribute combinations (particularly with only two attributes, as in these experiments). The proposed method should also be evaluated in traditional settings where all attribute combinations have some representation. This would confirm that the method performs well without requiring group-dropping.**
>
> In the paper, we had already carried out a comparison in the traditional setting where all attribute combinations have some representation. These were presented in the rightmost column in Table 1, "WGA (no groups dropped)", as well as in Table 14 in Appendix G.5 which contains additional metrics. CRM remains competitive with the baselines in this scenario. This confirms that the method performs well also without requiring group-dropping. We also want to point that as the number of groups grow (due to increase in $d$ or $m$), it is natural to expect that we will be disjoint setting where no samples are available from any group.
>
> Also, given the page limit, several results currently are in the supplementary material and we will move some results to the main body in the future revision. If you want us to move some specific result to the main body, please let us know.
>
> Thanks again for your constructive comments! We are open to further discussion and would be happy to address any remaining concerns.

---

### Decision · Program_Chairs · 2025-05-01

**Decision:**

Accept (poster)

**Comment:**

The paper proposes a novel attribute classification method that achieves Bayes-optimal classification for the test distribution, even when it includes attribute combinations not observed in the training data. Assuming that features follow an additive energy distribution, the authors derive a provable method that classifies, in a Bayes-optimal manner, any attribute composition lying within the discrete affine hull of the attribute compositions present in the training set.

Reviewers agree that the paper is well written, with all the claims supported by rigorous theoretical analysis and extensive experiments on benchmark datasets.